# Investigating the impact of reanalysis snow input on an observationally calibrated snow-on-sea-ice reconstruction

Alex Cabaj[1], Paul J. Kushner[2], and Alek A. Petty[3]

[1]Climate Research Division, Environment and Climate Change Canada, Toronto, ON, Canada
[2]Department of Physics, University of Toronto, Toronto, ON, Canada
[3]Earth System Science Interdisciplinary Center, University of Maryland, College Park, MD, USA

**Correspondence:** Alex Cabaj (alex.cabaj@ec.gc.ca)

**Abstract.** A key uncertainty in reanalysis-based snow-on-sea-ice reconstructions is the choice of reanalysis product used for snowfall input. Although reanalysis products have many similarities in their precipitation output over the Arctic Ocean, they nevertheless have relative biases that impact derived snow-on-sea-ice estimates. In this study, snowfall from the ERA5, JRA-55 and MERRA-2 reanalysis products is used as input to the NASA Eulerian Snow On Sea Ice Model (NESOSIM). A Markov chain Monte Carlo (MCMC) approach is used to calibrate the wind packing and blowing snow parameters in NESOSIM run with these different snowfall inputs. JRA-55 shows the largest departure from the previously-used values (Bayesian priors) when the MCMC calibration is run, and also has the largest posterior uncertainty due to parameter uncertainties. The MCMC calibration reconciles snow depths between NESOSIM run with different reanalysis snowfall inputs, but produces larger discrepancies in snow densities, due to the sensitivity of snow density in NESOSIM to parameter values and weak observational constraints on density. Regional climatologies and trends in the calibrated products are examined and compared to another reanalysis-based snow-on-sea-ice reconstruction, SnowModel-LG. NESOSIM and SnowModel-LG show close agreement in snow depth climatologies in the Central Arctic Ocean region, but differ more in peripheral seas. The models perform comparably when evaluated against IceBird airborne snow depth observations and in situ depth and density observations from MOSAiC. Trends are found to be region-dependent, and Central Arctic Ocean snow depth trends are more sensitive to the choice of reanalysis input than to the choice of model.

## 1 Introduction

Snow on Arctic sea ice plays a key role in controlling Arctic climate and ecosystem function, and is a crucial input to altimetry-derived sea-ice thickness retrieval, but is challenging to characterize consistently across the Arctic Ocean at basin scales (Webster et al., 2018). Satellite remote sensing data using, for example, depth retrievals from passive microwave data (Brucker and Markus, 2013; Rostosky et al., 2018) and altimetry-based snow depth retrievals (Lawrence et al., 2018; Kwok et al., 2020), provide basin-wide estimates of snow depth on Arctic sea ice, but are subject to significant retrieval limitations and uncertainties. Airborne (MacGregor et al., 2021; Jutila et al., 2022) and in situ (Wagner et al., 2022; Radionov et al., 1997) observation campaigns and automated snow buoys (Perovich et al., 2019; Nicolaus et al., 2017) provide generally more accurate, but also more localized observations. A complementary approach to estimate snow on Arctic sea ice on basin scales is through

reanalysis-based snow reconstructions, in which reanalysis snowfall forces a model that simulates snow processes while accounting for sea-ice concentration and drift. These reconstructions include SnowModel-LG (Liston et al., 2020), the University of Washington snow-on-sea-ice reconstruction (Blanchard-Wrigglesworth et al., 2018), and the NASA Eulerian Snow on Sea Ice Model (NESOSIM, Petty et al., 2018), which is the focus of our study. NESOSIM has been previously used with altimetry measurements from the NASA Ice, Cloud, and land Elevation Satellite 2 (ICESat-2) to produce estimates of Arctic sea ice thickness over the winter season (Petty et al., 2020, 2023b).

Not surprisingly, reanalysis snow-on-sea-ice reconstructions are strongly sensitive to snowfall input, which depends on several factors such as atmospheric process representation in reanalysis products (e.g. microphysical processes and partitioning between solid, liquid, and mixed phase precipitation), data assimilation inconsistencies, and product resolution. Reanalysis precipitation assessment for the Arctic (Behrangi et al., 2016; Boisvert et al., 2018; Barrett et al., 2020; Edel et al., 2020; Cabaj et al., 2020) is challenged by uncertainty in polar precipitation observations, especially over the Arctic Ocean. Reanalysis precipitation intercomparison work by Barrett et al. (2020) recommends that ERA5 be used to provide precipitation for sea ice thickness estimates over other contemporary reanalysis products, but acknowledges that other reanalysis products investigated in that study are of similar value for that application, given the difficulty of observational validation and bias-adjustment. Biases between reanalysis products can be reduced through calibration to satellite snowfall observations, but differences between products nevertheless persist, and satellite snowfall measurements themselves may be biased (Cabaj et al., 2020; Edel et al., 2020). This motivates the need for further calibration of snow-on-sea-ice reconstructions.

The purpose of this study is to improve consistency and characterize uncertainty amongst several reanalysis snowfall inputs for NESOSIM's snow-on-sea-ice reconstruction, using bias-adjusted snowfall input and automated calibration of NESOSIM's snow-model parameters. We also seek to assess the variability and trends in basin-wide and regional snow on Arctic sea-ice produced by NESOSIM using these newly recalibrated snow depth estimates. Our starting point is the latest version of NESOSIM, version 1.1 (v1.1; Petty et al., 2023b). In Cabaj et al. (2023), NESOSIM v1.1 free parameters for the wind packing (densification) and blowing snow (loss) processes were calibrated to snow-on-sea-ice depth and density observations using a Markov chain Monte Carlo (MCMC) approach, and uncertainty estimates for these free parameters were obtained. Several considerations motivated the development of an automated parameter calibration approach for NESOSIM. In recent upgrades to NESOSIM that included the switch to the latest ERA5 snowfall input, the free parameters were manually tuned to increase agreement with snow depths obtained from Operation IceBridge (Petty et al., 2023b). This process alluded to the potential for parameter tuning to account for forcing biases between products, as well as the potential benefits of a more automated system that could bring in different types of observations to evaluate both the snow depth and density concurrently. Optimizing NESOSIM output is motivated by its continued use as the main snow loading input to satellite derived sea ice thickness estimates from ICESat-2 (Petty et al., 2023a, b) and the need to better characterize the snow loading uncertainty contribution to the overall thickness uncertainty.

To better reconcile differences between NESOSIM run with different snowfall inputs, and to incorporate estimates of uncertainties due to the choice of model snowfall input, in this current study, we run the MCMC optimization for NESOSIM with additional reanalysis snowfall inputs, introducing MERRA-2 and JRA-55 to this study in addition to ERA5. This also

necessitates a revisiting of the CloudSat calibration for reanalysis snowfall first performed in Cabaj et al. (2020), since a longer time record and an additional reanalysis product are used in this study. We estimate resulting snow depth uncertainties due to parameter uncertainty, which helps account for uncertainties due to snow input, and examine the impact of this parameter optimization on the agreement between snow depth and density derived using these products. Then, we construct a consensus snow depth estimate that accounts for variability in reanalysis snowfall from the average of calibrated NESOSIM output for different reanalysis snow inputs, motivated by work combining land snow products (Mudryk et al., 2015) which demonstrates how multi-dataset approaches can help to reveal biases between datasets and facilitate the characterization of dataset uncertainties. We evaluate the consistency of the outputs across different snowfall forcing inputs, examining the climatologies, the interannual variability, and trends, over the 1980-2021 time period. We compare the NESOSIM output to SnowModel-LG (Liston et al., 2020), another reanalysis-based snow-on-sea-ice model. SnowModel-LG likewise includes observation-based calibration, namely an assimilation-based bias correction to precipitation to bring modelled snow depth into agreement with ground-based and remote sensing observations, including Operation IceBridge measurements (Liston et al., 2020; Stroeve et al., 2020). We further compare the NESOSIM and SnowModel-LG model outputs to independent observational datasets, namely in situ snow depth and density observations from the 2019-2020 MOSAiC observational campaign (Itkin et al., 2023; Macfarlane et al., 2023), and airborne snow depth observations from April 2017 and 2019 from the AWI IceBird campaign (Jutila et al., 2022).

## 2 Data products and models

### 2.1 Reanalysis products

Snowfall rates from the ERA5, MERRA-2, and JRA-55 reanalysis products are used as input to NESOSIM in this study. ERA-Interim is examined for reference, but not used as input to NESOSIM, since it has been superseded by ERA5. A summary of the reanalysis products used in this study is shown in Table 1, and each product is discussed in more detail in the subsections below. To format the reanalysis snowfall for use as input to NESOSIM, the snowfall rate from each reanalysis product is aggregated by day to produce daily snowfall, and then regridded to the 100 km $\times$ 100 km equal-area NESOSIM model grid. NESOSIM also uses 10-m wind input from reanalysis products, but for this study, ERA5 winds were used for all model runs.

| Reanalysis | Spatial Resolution | Time Resolution | Assimilation scheme | Reference |
|---|---|---|---|---|
| ERA-Interim | 0.75° $\times$ 0.75° | 6-hourly | 4DVar | Dee et al. (2011) |
| ERA5 | 0.25° $\times$ 0.25° | Hourly | 4DVar | Hersbach et al. (2020) |
| MERRA-2 | 0.5° $\times$ 0.625° | Hourly | 3DVar | Gelaro et al. (2017) |
| JRA-55 | 1.25° $\times$ 1.25° | 3-hourly | 4DVar | Kobayashi et al. (2015) |

**Table 1.** Reanalysis products examined in this study. Spatial resolution refers to the regular lat-lon grid used for the products in this study. To provide input to NESOSIM, all reanalysis products are regridded to the equal-area 100 km $\times$ 100 km polar grid used by the model.

### 2.1.1 ERA-Interim

The European Centre for Medium-Range Weather Forecasts (ECMWF) Re-Analysis Project ERA-Interim (Dee et al., 2011) reanalysis product is widely used in studies of Arctic snow, and is often used for precipitation input in snow models (Kwok and Cunningham, 2008; Petty et al., 2018; Blanchard-Wrigglesworth et al., 2018). It has been found to have high correlations and low biases with respect to observations of Arctic land precipitation (Lindsay et al., 2014). Sea ice concentration is represented as a fractional quantity for grid cells with concentration greater than 20%, while grid cells with less than 20% concentration

are designated as open ocean. ERA-Interim is produced using a 4DVar assimilation scheme, and it features a T255 ($\sim$ 79 km) resolution spectral dynamical core. The ERA-Interim snowfall product is provided on a N128 Gaussian grid, re-gridded to a 0.75°$\times$ 0.75° latitude/longitude grid in this study. Production of ERA-Interim has stopped as of August 2019.

### 2.1.2 ERA5

The ECMWF Reanalysis v5 (Hersbach et al., 2020), the successor to ERA-Interim, features many improvements, such as a

95 finer model resolution, an updated assimilation scheme, and an improved cloud scheme, including improvements to the representation of mixed-phase clouds and ice-phase cloud microphysics (Hersbach et al., 2020). It has been found to produce more snow than ERA-Interim, especially in the Atlantic sector (Wang et al., 2019). Like ERA-Interim, ERA5 uses a 4DVar assimilation scheme. The representation of sea ice concentration is also the same as in ERA-Interim, with fractional concentration above a 20% open ocean threshold. In this study, the ERA5 snowfall rate product is interpolated from its native N320 Gaussian

grid to a 0.25°$\times$0.25° grid. Currently, ERA5 is used as the default snowfall and 10-m wind input for NESOSIM v1.1 (Petty et al., 2023b; Cabaj et al., 2023).

### 2.1.3 MERRA-2

NASA's Modern-Era Retrospective analysis for Research and Applications, Version 2 (Gelaro et al., 2017) is produced on a cubed-sphere grid, with a finite-element dynamics scheme, and is used in this study with its native horizontal resolution

of 0.5°$\times$0.625° ($\sim$ 55 km). Unlike the other reanalysis products investigated in this study, which use a 4DVar assimilation scheme, MERRA-2 uses a 3DVar assimilation scheme, with an Incremental Analysis Update procedure which applies the analysis increment as a constant term over the assimilation window instead of only correcting the initial condition, as is done conventionally for 3DVar (Gelaro et al., 2017). Sea ice is distinguished from open ocean based on a 50% concentration threshold. MERRA-2 is known to produce more total precipitation over the Arctic compared to other reanalysis products

(Barrett et al., 2020; Boisvert et al., 2018).

### 2.1.4 JRA-55

The Japanese Meteorological Agency's Japanese 55-year Reanalysis (Kobayashi et al., 2015) is another widely-used product for Arctic snowfall estimates, and it is interpolated onto a 1.25°$\times$1.25° grid from its native TL319 ($\sim$ 55 km) spectral resolution. The product uses a 4DVar assimilation scheme. Sea ice is represented in JRA-55 with a binary classification based on a 55%

concentration threshold. JRA-55 has been previously used as a source of snowfall input for snow-on-sea-ice reconstructions, and was investigated as an input for NESOSIM version 1.0 (v1.0; Petty et al., 2018). In comparisons of total precipitation over the Arctic Ocean, JRA-55 has been found to produce less precipitation overall than other reanalysis products (Barrett et al., 2020).

## 2.2 CloudSat

CloudSat was a satellite equipped with a 94-GHz Cloud Profiling Radar (CPR) instrument which measured vertical profiles of cloud and hydrometeor reflectivity, from which near-surface snowfall rate was retrieved (Kulie and Bennartz, 2009). The satellite had an observational footprint of $1.4 \times 1.7$ km (along and across track), and a 16-day repeat cycle. The instrument was operational from 2006-2023, with an interruption in 2011 due to a battery malfunction, and a change to a lower orbit in 2018. In this study, surface snowfall rates from the 2C-SNOW-PROFILE product, version P1 R05 (Wood et al., 2013, 2014)
are used to bias-correct snowfall rates from reanalysis products by scaling the reanalysis monthly climatologies to the monthly climatology of regionally-aggregated CloudSat snowfall, following the approach in Cabaj et al. (2020). CloudSat measurements from 2006-2016 are used in this study. CloudSat's ground track had latitudinal coverage between 82°N and 82°S. Its sampling is also limited following the 2011 battery malfunction to observations taken when the instrument has line-of-sight to the sun, which may introduce low biases during the later observational period (Milani and Wood, 2021). Nevertheless, CloudSat has
been used extensively in studies of high-latitude precipitation and snowfall (Behrangi et al., 2016; Edel et al., 2020; Kulie et al., 2016; Milani et al., 2018). To mitigate ground clutter contamination of surface returns in this study, near-surface snowfall rate measurements are retrieved from the 3rd vertical bin above ocean surfaces (approximately 720 m above the surface), or the 5th vertical bin above sea ice (as determined by a climatological sea ice mask, approximately 1200 m above the surface) (Wood and L'Ecuyer, 2018). Data quality flags are applied to exclude potentially contaminated observations as described in Cabaj
et al. (2020).

## 2.3 MOSAiC

The Multidisciplinary Drifting Observatory for the Study of Arctic Climate (MOSAiC) expedition took place in September 2019-October 2020, providing high-quality in-situ observations of snow and sea ice for the duration of an entire sea ice season at a central Arctic location (Nicolaus et al., 2022). In this study, we use snow depth measurements recorded using automated
snow depth probes (MagnaProbes) (Itkin et al., 2021) and bulk snow densities calculated from snow density cutters (Macfarlane et al., 2021, 2022) as independent observational datasets for comparison with model output (Macfarlane et al., 2023). Snow depth measurements were collected over regions representative of first-year and second-year ice, over level ice and ridges. Snow density cutter measurements likewise were taken for a variety of snow conditions. To partly compensate for the highly localized and heterogeneous nature of the observations, we collocate the observed values with the nearest respective model
grid cell, and average the values by day within each grid cell. Then, we calculate monthly means of these daily-aggregated values for comparison with modelled values. Prior to aggregation, bulk snow densities are first calculated by calculating the height-weighted average of snow density samples measured in each snow density profile.

## 2.4 IceBird

The Alfred Wegener Institute (AWI) IceBird observational campaign is an airborne observational campaign for collecting measurements of sea ice thickness, and sea ice and snow properties. We make use of snow depth observations collected by airborne snow radar from April 2017 and April 2019 (Jutila et al., 2022; Jutila et al., 2024a, b) as independent observational datasets for comparison with model output. Survey tracks cover regions over the western Arctic Ocean, around the Canadian Arctic Archipelago and Beaufort Sea regions, and they encompass both first-year and multi-year ice. For comparison with model output, we calculate the average of observed values from the transects within each coincident model grid cell. Each transect spans multiple model grid cells, though some measurements coincide with grid cells considered as land by the models due to model resolution limitations.

## 2.5 NESOSIM and MCMC calibration

The NASA Eulerian Snow on Sea Ice Model (NESOSIM) produces estimates of snow depth and bulk snow density over winter Arctic (September to April) sea ice on a $100 \times 100$ km polar grid (Petty et al., 2018). The model is a 2-layer Eulerian snow-on-sea-ice model, and includes parameterized representations of snow accumulation, densification through wind packing, loss from blowing snow to the atmosphere and open ocean, and redistribution of snow due to sea ice motion. NESOSIM was initially developed to provide estimates of snow depth to enable the rapid production of Arctic sea ice thickness estimates from ICESat-2 (Petty et al., 2020, 2023b).

Several observational and reanalysis inputs are used in NESOSIM. Snowfall input for NESOSIM is provided from reanalysis products, with ERA5 being used as the default product as of v1.1, and ERA-Interim previously used as the default in v1.0. In this study, multiple reanalysis products are investigated as a source of snowfall input. Reanalysis products are also used for wind input to NESOSIM. This study uses ERA5 10 m wind as input to the model, motivated by findings that the ERA5 wind product performs relatively well with respect to observations compared to other reanalysis products in Arctic regions (Graham et al., 2019a, b). Sea ice concentration is provided by the NOAA/NSIDC Climate Data Record (CDR) product (Peng et al., 2013). Sea ice drift for the MCMC calibration is obtained from the low resolution sea ice drift product of the EUMETSAT Ocean and Sea Ice Satellite Application Facility (OSI SAF; Lavergne et al., 2010). Since the OSI SAF drift product is not available for years prior to 2009, sea ice drift from the NSIDCv4 Polar Pathfinder product (Tschudi et al., 2019) was used to generate the full 1980-2021 datasets. Aside from reanalysis products, these inputs are the same as those used in previous work using NESOSIM v1.1 (Petty et al., 2020, 2023b; Cabaj et al., 2023).

Representations of snow processes in NESOSIM are highly simplified. Since NESOSIM is a 2-layer model, bulk snow density in the model is represented as a weighted sum of the prescribed densities for old snow (350 kg/m$^3$) and new snow (200 kg/m$^3$), respectively. The old snow density represents both wind slab and depth hoar (Petty et al., 2018). These prescribed values impose maximum and minimum values on the bulk density represented by the model. Snow may be redistributed between model grid cells through the action of sea ice drift, although the representation of drift represents a large-scale average, given the model resolution. Melt processes are currently not included in NESOSIM, so the model is run from September to

April in each season, and reinitialized on September 1st based on climatological mean snow depths scaled by the number of melt days in the previous season (Petty et al., 2018).

The wind packing and blowing snow parameters in NESOSIM are free parameters, and previous work introduced an automated calibration of these parameters using an MCMC process (Cabaj et al., 2023). At each model step and grid point, if the input wind speed exceeds a prescribed threshold speed of 5 m/s, the wind packing and blowing snow processes act on the snow in NESOSIM. Wind packing controls the amount of snow transferred between layers, decreasing the snow depth and increasing the bulk snow density as snow is transferred from the upper (less dense) layer to the lower (denser) layer. The blowing snow process acts only on the upper snow layer, and decreases the snow depth in the upper layer linearly with wind speed. The blowing snow term includes an atmosphere loss and an open-water loss term, which are prescribed separately in NESOSIM v1.1 (Petty et al., 2023b). The open-water loss term accounts for sea ice concentration, with regions of lower sea ice concentration experiencing more open-water loss. For the purpose of this study, the blowing snow term parameters are treated as a single term, as was done in previous work (Cabaj et al., 2023), with the atmospheric loss factor being 0.15 times the blowing snow parameter. The blowing snow term is exclusively a loss term and does not include redistribution. When snow is lost from a grid cell via this process, it is removed, not redistributed to another grid cell. This current study will extend previous parameter calibration work by investigating the impact of using different reanalysis snowfall input products in NESOSIM.

Previous work (Cabaj et al., 2023) demonstrated a successful calibration of NESOSIM's wind packing and blowing snow parameters using an MCMC process when NESOSIM was run with ERA5 snowfall. MCMC is an algorithm applied to Bayesian problems where, given prior information of the parameters and observational constraints on the parameters, posterior parameters may be obtained which produce model output that is more closely aligned to observations, as determined by a cost function; in this case, a log-likelihood function of the difference between model output and selected aggregated observations used for the calibration, weighted by the uncertainty in the observations. Using an MCMC approach for calibrating NESOSIM allows for the automated estimation of free parameters, which were previously manually estimated via comparison to observations (Petty et al., 2018). An added benefit of this approach is that it yields posterior distributions of the parameters, which provide an estimate of parameter uncertainty subject to observational constraints. The MCMC process is iterative, and is conducted for NESOSIM following the approach in Cabaj et al. (2023). Further description of the NESOSIM MCMC calibration is provided in Appendix B.

## 2.6 SnowModel-LG

SnowModel-LG (Liston et al., 2020; Stroeve et al., 2020) is a Lagrangian snow-on-sea-ice model. Like NESOSIM, it takes snowfall input from reanalysis products. However, the representation of snow processes in SnowModel-LG is considerably more complex than NESOSIM, with the notable inclusion of snow melt, snowpack metamorphosis processes, and multiple snow layers (a maximum of 25 layers for the product used in this study). Output is provided with a daily temporal resolution and a spatial resolution of $25 \times 25$ km. SnowModel-LG output has been found to compare favourably with several observational campaigns (Stroeve et al., 2020), though agreement depends on the region and time period of comparison.

The ERA5 and MERRA-2 reanalysis products are used to provide snowfall input to SnowModel-LG. SnowModel-LG also includes an observation-based calibration, with scaling factors applied to the reanalysis snowfall based on a correction empirically derived from Operation IceBridge snow depth measurements (Liston et al., 2020; Stroeve et al., 2020). The assimilation approach used in SnowModel-LG is consistent with optimal interpolation approaches (Liston and Hiemstra, 2008).

In this study, output from SnowModel-LG run with ERA5 and MERRA-2 input (retrieved from the National Snow and Ice Data Center; Liston et al. (2021)) is used for comparison with NESOSIM. SnowModel-LG does not include the Canadian Arctic Archipelago region, so this region is not considered for the comparisons between SnowModel-LG and NESOSIM in this study. Furthermore, whereas NESOSIM is initialized in September, SnowModel-LG is initialized in August and run through the melt season. For consistency, only months during which NESOSIM and SnowModel-LG data are both available will be considered in this study.

## 3    Investigating different reanalysis snowfall products

Here, we present a comparison of the reanalysis snowfall products used in this study as input to NESOSIM. Reanalysis snowfall products are calibrated to CloudSat following the approach from Cabaj et al. (2020), but in this study, additional products are used and a longer time series is examined, as discussed below.

### 3.1    Reanalysis snowfall calibration to CloudSat

Figure 1 shows regionally-aggregated monthly-mean snowfall rates over the ocean region in the 60-82°N latitude band from reanalysis products and CloudSat, from 1980-2016, without and with scaling to the CloudSat monthly climatology (Cabaj et al., 2020). The scaling entails taking the monthly reanalysis snowfall rate for each month and multiplying it by a scaling factor, which consists of the CloudSat climatological monthly mean snowfall rate divided by the reanalysis climatological monthly mean snowfall rate for each respective month. The climatological means for this scaling are taken from 2006-2016, excluding months in 2011 where CloudSat observations are absent due to instrument malfunctions. Further details of this scaling are provided in Cabaj et al. (2020). Before the scaling is applied in Fig. 1, there is some variation between the reanalysis products, although they have similar seasonal cycles and generally coincident seasonal maxima and minima. ERA5 and MERRA-2 have relatively high snowfall compared to ERA-Interim and JRA-55. Snowfall rates from CloudSat, which are available from 2006-2016 with a gap in 2011, are comparable to the snowfall rates of the other products. MERRA-2 is known to be wetter compared to other reanalysis products over the Arctic, when total precipitation is considered (Barrett et al., 2020; Boisvert et al., 2018). This is particularly reflected in the summer months, where MERRA-2 snowfall rates are the largest relative to the other products.

As in Cabaj et al. (2020), we bias-adjust reanalysis snowfall input to climatological CloudSat snowfall for 2006-2016. CloudSat scaling improves agreement amongst the reanalyses both within and outside the 2006-2016 calibration period (Fig. 1b). Although MERRA-2's seasonal cycle in snowfall is consistently of greater amplitude than the other products prior to 2006, the agreement of MERRA-2 with the other products is considerably improved with the scaling. JRA-55, which was not

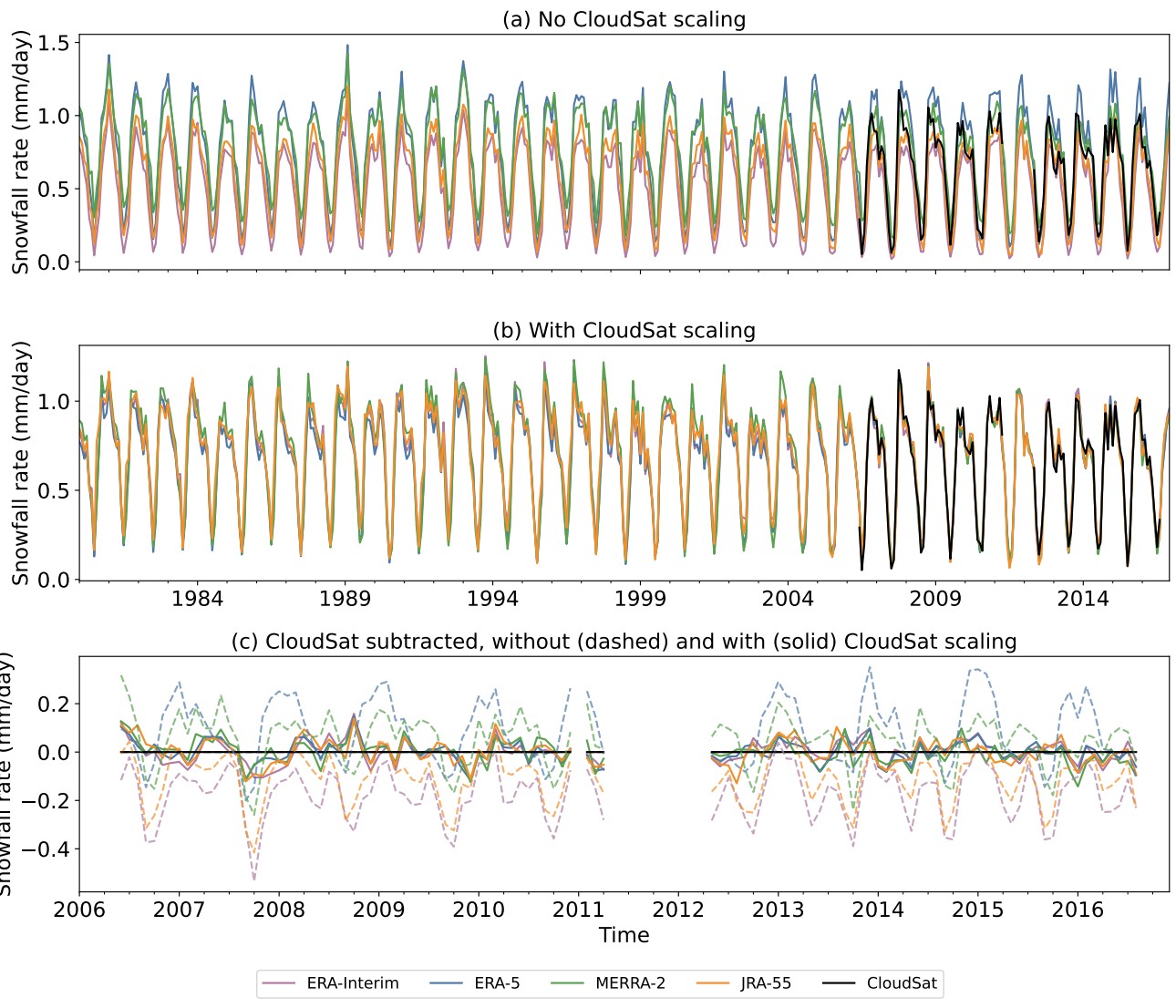

**Figure 1.** Monthly mean snowfall rates from reanalysis products and CloudSat, regionally-averaged over the ocean region in the 60-82°N latitude band (i.e. excluding land), (a) without scaling to CloudSat, and (b) scaled to the CloudSat monthly snowfall climatology. Panel (c) shows the difference between the reanalysis products and CloudSat for the no-scaling (dashed) and with-scaling (solid) cases, from 2006-2016.

previously investigated in this context, is also brought into closer agreement with the other products using this approach. This highlights the continued benefits of this bias-adjustment approach for reconciling reanalysis snowfall products.

## 3.2 Snowfall comparison over ocean and sea ice for the NESOSIM domain

To apply CloudSat scaling over the NESOSIM model domain, a more regionally-refined scaling approach is used. Reanalysis
snowfall rates are scaled to CloudSat measurements from 60-82°N over four quadrants, as described in Cabaj et al. (2020). The CloudSat scaling was previously found to improve agreement in basin-averaged and regionally-averaged snow depths in NESOSIM v1.0 (Cabaj et al., 2020). Some adjustments were made to the scaling for NESOSIM v1.1, which has a larger model domain (Petty et al., 2023b), extending down to 50°N, compared to 60°N for NESOSIM v1.0 (Petty et al., 2018). For the current version of the model, the scaling is performed as follows.

As in previous work, the NESOSIM v1.1 model domain is divided into quadrants with longitudinal boundaries at longitudes 135°W, 45°W, 45°E, and 135°E, respectively, as illustrated in Fig. A1. For each quadrant, within a region between latitudes of 60°N and 82 °N (indicated by solid lines in Fig. A1), CloudSat scaling factors are calculated by dividing the CloudSat monthly snowfall climatology by the reanalysis monthly snowfall climatology averaged over the region. The same CloudSat scaling factors as previously calculated for Cabaj et al. (2020) are used, with the addition of scaling factors for JRA-55 which were not
previously calculated. The scaling factors are linearly interpolated across the pole from corners at latitudes of 45 °N (longitudes of 90°W, 0°E, 90°E, and 180°E, indicated as A-D in Fig. A1), consistent with the interpolation performed for version 1.0 of the model. To account for the extended model domain in NESOSIM version 1.1, the scaling factors are extrapolated southwards from the same corners (A-D) as constant values. This process creates a set of monthly gridded scaling factors which are then multiplied by the daily reanalysis snowfall rates used as input to NESOSIM v1.1, with a different set of scaling factors used
for each month.

Figure 2 shows the impact of CloudSat scaling as applied to NESOSIM model input for monthly climatologies of reanalysis snowfall rates over ocean (which includes both ice and open ocean, with land masked out), and over sea ice only, respectively, regionally-averaged over a representative subset of the different Arctic regions shown in Fig. A1.

CloudSat scaling effectively reconciles differences between reanalysis products for the pan-Arctic ocean region in Fig.
1 during the satellite era, but shows less consistency for individual regions and when ice-covered scenes are broken out. Over the ice-plus-ocean region, for which the CloudSat scaling was originally developed, the CloudSat scaling reconciles differences between the products for most months in most regions. A notable exception is in the Central Arctic region, where the September snowfall values are excessively large for JRA-55 following the application of the CloudSat scaling. This may be because JRA-55 is biased relatively low compared to CloudSat and the other products, so the CloudSat scaling, determined
using ice-plus-ocean scaling factors, greatly increases the snowfall rates, especially in the early part of the sea ice season. Furthermore, since CloudSat observations are limited to latitudes south of 82°N, the scaling factors may be less reliable over more poleward regions. Over the ice-covered region alone, the CloudSat scaling reduces inter-product consistency in some regions and months. Over sea ice, the overall inter-product spread increases in September and October in the Central Arctic, October in the Beaufort Sea, October-November in the Chukchi Sea, all months except January and September for the Kara

Regional snowfall climatologies from reanalysis products before and after CloudSat scaling

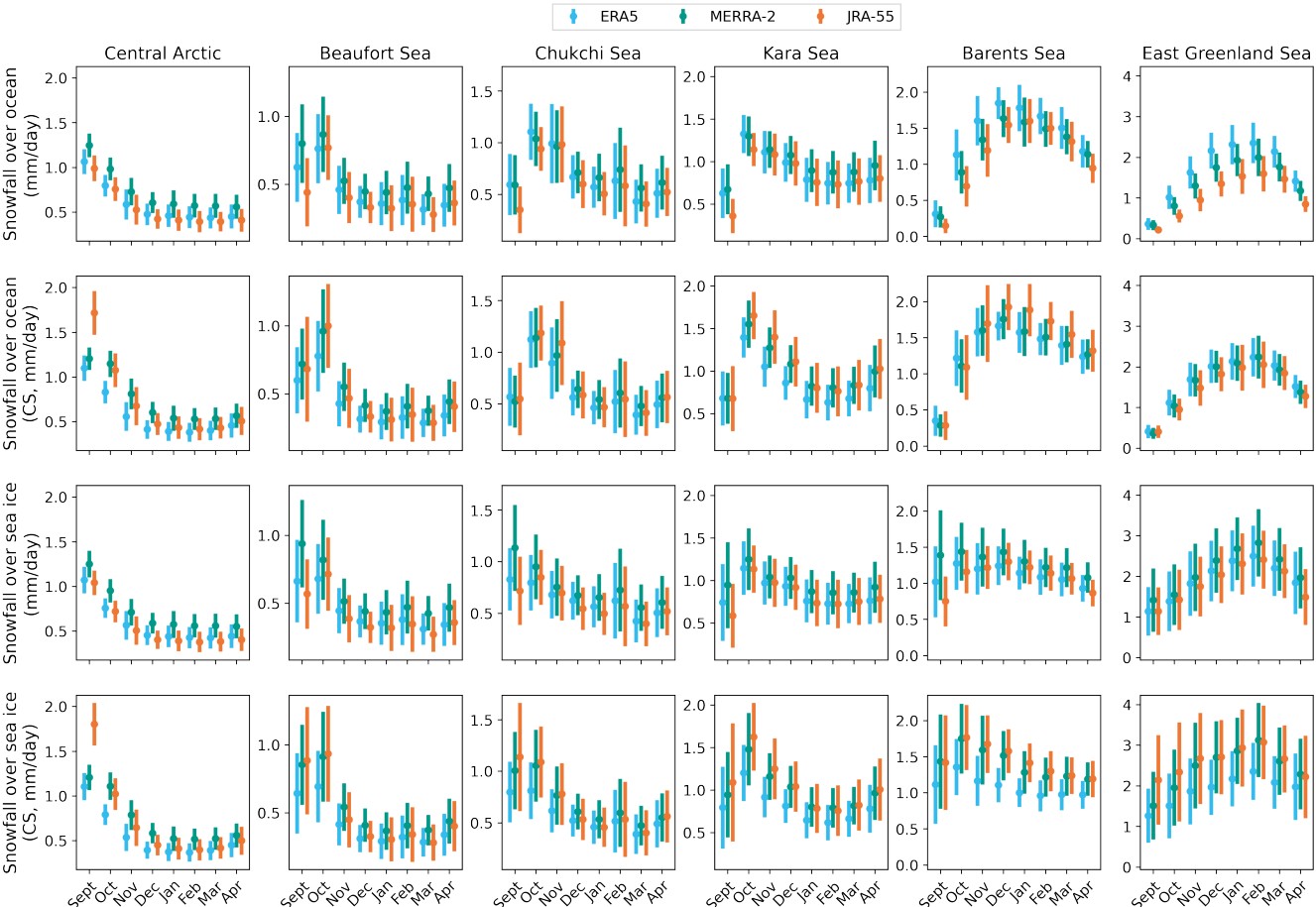

**Figure 2.** Monthly climatologies of regionally-averaged snowfall rates from reanalysis products from 1980-2021; over the ocean region (i.e. including both ice-covered and ice-free ocean) and over sea ice only, without and with CloudSat scaling (CS) applied. Bars represent the interannual standard deviations of the climatologies. Note the differing scales on the vertical axes between regions.

Sea, all months except April and September in the Barents Sea, and all months except April in the East Greenland Sea. In the Kara, Barents, and East Greenland Sea regions in particular, JRA-55 and MERRA-2 are closely reconciled by the CloudSat scaling, but ERA5 is less changed by the scaling, which results in it being biased relatively low with respect to the other products.

Seasonal cycles of snowfall over sea ice may be similar to snowfall over the ice-plus-ocean region in some regions, but
other regions show stark differences. Of the regions shown, the seasonal cycles and magnitudes differ considerably between the two cases in the Chukchi Sea, Barents Sea, and East Greenland Sea regions. By comparison, the differences are notably less stark in the Central Arctic region, which has considerable ice cover even in the early season. In the Kara and Beaufort seas, the seasonality is similar between the two cases, although the magnitudes differ. Many of the regions show a relatively low snowfall over the ocean-plus-ice region in September, but those same snowfall minima are not as prominent in the plots
for the ice-covered regions. This suggests that much of the snow that is present during the early part of the season is coincident with the presence of sea ice. This may be due to ice-covered regions having lower temperatures, which permits the presence of early-season snowfall where other regions may have rainfall. Despite these regional inconsistencies, due to the limited overlap between CloudSat orbits and ice-covered regions which would likewise adversely impact CloudSat scaling if it were restricted to ice-covered regions, we proceed with the established CloudSat scaling factors. We will return to the discussion of issues
related to CloudSat scaling in Section 7.

Discrepancies in reanalysis input yield discrepancies in the output from NESOSIM driven by different reanalysis snowfall products. This motivates the observation-constrained calibration of NESOSIM run with different reanalysis snowfall inputs using an MCMC method, as was previously done in Cabaj et al. (2023). The following section discusses an updated calibration of NESOSIM and the impact on model-derived snow depth and density.

## 4   Impact of MCMC calibration on NESOSIM output

### 4.1   Posterior model parameters

In this study, the same MCMC approach in Cabaj et al. (2023) is repeated, but with the addition of other reanalysis snowfall inputs; MERRA-2 and JRA-55 snowfall are used in addition to ERA5. The MCMC process was run for 10,000 iterations for each snowfall input product. The first 5,000 iterations are discarded to exclude "burn-in" values, as was done in Cabaj et al.
(2023). Nevertheless, in each case, the optimal posterior parameter values did not differ significantly between the first (burn-in) and subsequent (after burn-in) sets of iterations, demonstrating robust convergence of the MCMC process.

Figure 3 shows the posterior distributions obtained from the MCMC calibrations of NESOSIM run with snowfall input from ERA5, JRA-55, and MERRA-2, respectively. The posterior distributions are aggregated from parameter values that are accepted during the MCMC process, and provide both the optimal (maximum-likelihood) parameter values, and estimates of
the parameter uncertainties (Gelman et al., 2013). Numerical values for the optimal parameters and associated uncertainties are shown in Table 2, along with the coefficients of variation and the acceptance rates. The acceptance rate, calculated from the ratio of accepted parameters to the total number of iterations, indicates the efficiency of the MCMC process, with an optimal

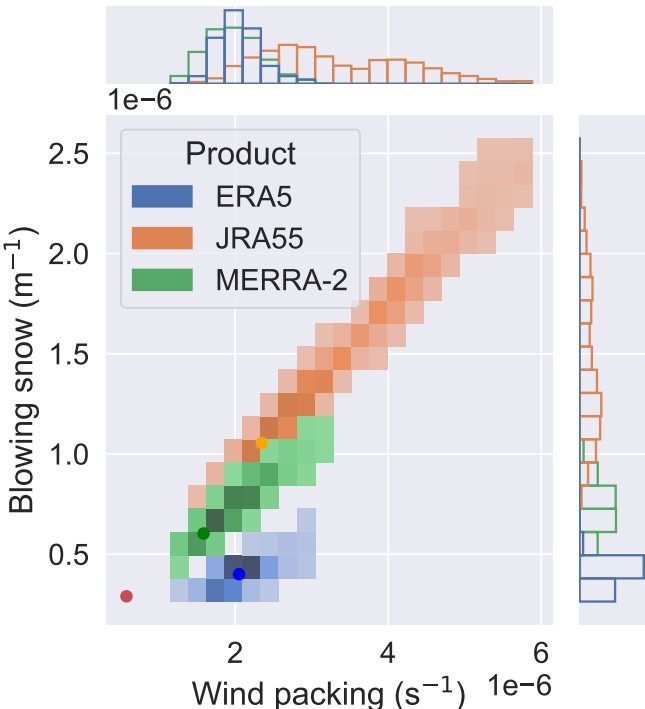

**Figure 3.** Posterior wind packing and blowing snow parameter distributions from MCMC calibration using snow input from ERA5, JRA-55, and MERRA-2, respectively. Note that the distributions have some overlap. The red dot indicates the prior parameter values and the other coloured dots indicate the optimal parameter values for the three respective products. The side panels show the marginal distributions, highlighting the overlap.

| Configuration | WP (s$^{-1}$) | $\sigma_{WP}$ (s$^{-1}$) | $CV_{WP}$ | BS (m$^{-1}$) | $\sigma_{BS}$ (m$^{-1}$) | $CV_{BS}$ | AR |
|---|---|---|---|---|---|---|---|
| Model default (prior) | $5.8 \times 10^{-7}$ | $1 \times 10^{-8}$ | - | $2.9 \times 10^{-7}$ | $1 \times 10^{-8}$ | - | - |
| MCMC-ERA5 | $2.05 \times 10^{-6}$ | $3.11 \times 10^{-7}$ | 15% | $4.01 \times 10^{-7}$ | $5.30 \times 10^{-8}$ | 13% | 29% |
| MCMC-JRA-55 | $2.35 \times 10^{-6}$ | $9.82 \times 10^{-7}$ | 42% | $1.05 \times 10^{-6}$ | $4.01 \times 10^{-7}$ | 38% | 46% |
| MCMC-MERRA-2 | $1.59 \times 10^{-6}$ | $3.40 \times 10^{-7}$ | 21% | $6.03 \times 10^{-7}$ | $1.12 \times 10^{-7}$ | 19% | 33% |

**Table 2.** Optimal parameters from MCMC optimization for different reanalysis inputs, with the default (prior) configuration and the prescribed prior standard deviations included in the first row for reference. WP refers to wind packing and BS refers to blowing snow. MCMC-derived standard deviations are denoted by $\sigma$. $CV$ refers to the respective coefficients of variation for each parameter. AR refers to the acceptance rate of the MCMC optimization; i.e. the percentage of iterations whose test parameter values are accepted.

efficiency for a 2-parameter MCMC process being approximately 23% (Gelman et al., 2013). Coefficients of variation are calculated from the standard deviation of the posterior distribution divided by the posterior parameter value, and quantify the relative spread of the posterior distribution. This provides a quantitative indication of how well-constrained the parameters are

by the MCMC calibration. The posterior distribution of ERA5 is considerably narrower than the distributions of MERRA-2 and JRA-55, with the latter being noticeably broad relative to the posterior distributions of the other two products. The coefficients of variation for the wind packing parameters, as indicated in Table 2 are 15% for ERA5, 42% for JRA-55, and 21% for MERRA-2. The coefficients of variation for the blowing snow parameters are 13% for ERA5, 38% for JRA-55, and 19% for MERRA-2. The JRA-55 parameter distribution has a slight bimodality in both wind packing and blowing snow, although the maximum-likelihood parameter corresponds to the larger mode. The spreads of the parameters for ERA5 and MERRA-2 are more comparable, with the MERRA-2 posterior distributions being somewhat wider than those of ERA5. In terms of departure from the prior values, ERA5 has the closest blowing snow value to the prior, and MERRA-2 has the closest wind packing to the prior. JRA-55 demonstrates the largest departure from the prior parameter values overall. The acceptance rates are included primarily as an indicator of the efficiency of the MCMC process; ERA5 and MERRA2 are relatively close to the optimal (23%) efficiency for a 2-parameter MCMC optimization (Gelman et al., 2013). The comparatively large acceptance rate for JRA-55 suggests that a larger step size could be used for the MCMC optimization for this product, but given that the NESOSIM-MCMC optimization is not excessively computationally expensive, the existing configuration is sufficient for the scope of this study. The correlation between the wind packing and blowing snow parameters may be a consequence of the processes compensating for each other, as described in Cabaj et al. (2023). The wind packing process transfers snow to the lower layer, where the blowing snow process cannot remove snow, so if wind packing is strengthened, the blowing snow process may also be strengthened to compensate and enable additional snow depth reduction.

These results highlight that model parameter tuning is heavily dependent on the forcing dataset. Care must be taken when using reanalysis-based snow-on-sea-ice reconstructions such as NESOSIM with different snow input datasets, since model processes may be less physically representative when different inputs are used. When developing parameterizations for such model processes, it is important to consider that biases in model inputs will be likewise reflected in model parameterizations. Biases in observations used for calibration will also impact the model output, but may be inevitable given the relative scarcity of in situ observations of snow on sea ice. Overall, MCMC can be used to objectively determine model parameters and their uncertainty given uncertainty of inputs.

## 4.2 Snow depth and density uncertainty estimates

Given posterior parameter uncertainties provided from the MCMC calibration, uncertainty in the NESOSIM output can be calculated. Figure 4 shows the depth and density for a single representative late-decade year, with shading representing the associated MCMC-estimated uncertainty for each respective product. The uncertainty is calculated from a 100-parameter ensemble run with the wind packing and blowing snow parameters sampled from the posterior distribution for each respective MCMC calibration, as in Cabaj et al. (2023). This represents uncertainty due to the model parameter uncertainty, and therefore does not characterize all the uncertainties in the model. The day-to-day variability is quite similar between the time series, although some differences remain between the products. NESOSIM with JRA-55 input shows a sharper early-season increase in snow depth compared to the other products, although the late-season snow depth is comparable to those of the other products. Snow density values for the NESOSIM-JRA-55 output are also largest overall, reflecting its stronger wind packing. The

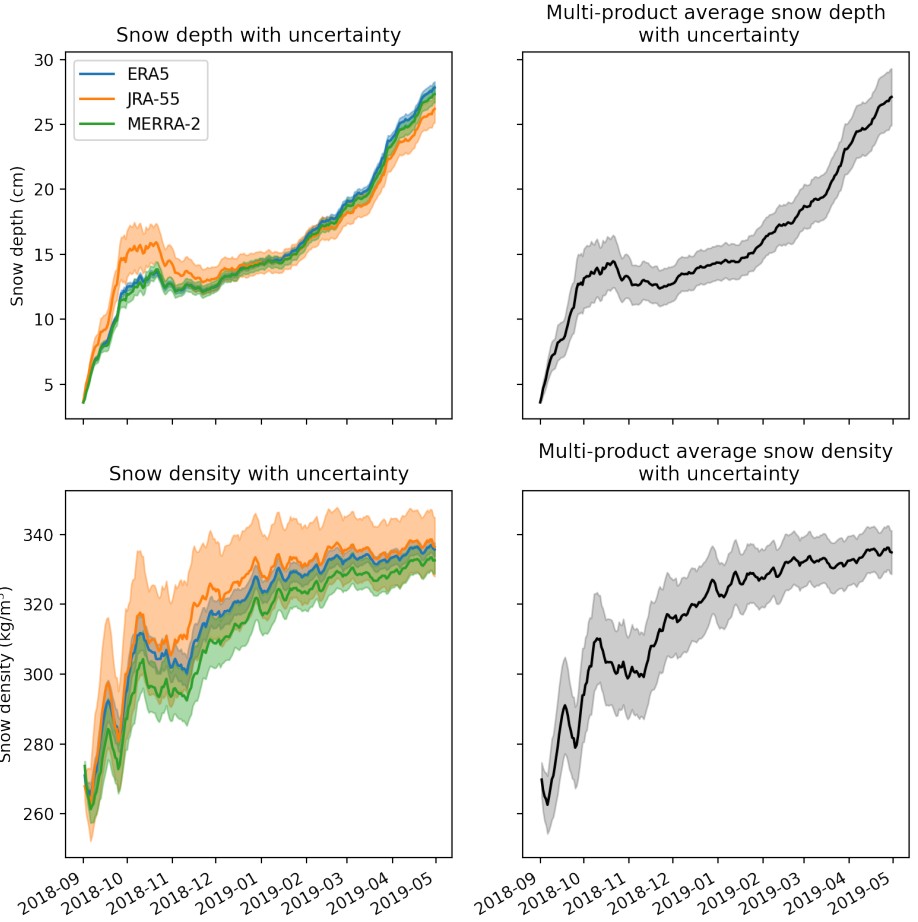

**Figure 4.** Single-year daily snow depth and density time series for each MCMC-calibrated NESOSIM configuration (with snowfall input from ERA5, JRA-55, and MERRA-2 reanalysis products) and for the multi-product average. Shading denotes uncertainty estimated based on the MCMC parameter uncertainty.

agreement in day-to-day density variations is likely a consequence of the identical ERA5 wind inputs to each NESOSIM run, since wind packing is dependent on wind input to NESOSIM. NESOSIM-JRA-55 has the largest uncertainty in snow depth and snow density, which is consistent with the spread of the posterior parameter distributions.

The multi-product average was calculated as the average of the MCMC-calibrated output from NESOSIM for the three different reanalysis inputs. The uncertainty in the multi-product average was calculated using the standard deviation of the

three 100-parameter model-run ensembles for the three reanalysis products. It thus quantifies both the uncertainty due to model parameter uncertainty, and the spread from the use of different model snowfall inputs. Initially, a bootstrap sampling approach was attempted to produce potentially more robust estimates, but it was found that the bootstrap-estimated standard deviations differed from the directly-calculated standard deviations by at most only 0.05%. Hence, the standard deviation of the combined parameter ensemble was used to calculate the multi-product average uncertainty for the depth and density.

To enable more direct uncertainty comparisons, uncertainties and percent uncertainties for depth and density due to model parameter uncertainty are shown in Fig. 5. A plot of monthly interannually averaged uncertainties from 1980-2021 is also available in Fig A2. As in Fig. 4, the percent uncertainty magnitudes reflect the shape of the posterior distributions. The relative insensitivity of NESOSIM snow output to model parameter values, as was observed in Cabaj et al. (2023), persists here; the percent uncertainties are considerably smaller than the NESOSIM parameter uncertainty, represented by coefficients of

variation, as discussed in Section 4.1. The NESOSIM-ERA5 uncertainties are relatively small compared to the other products. The percent uncertainties for all the products attain their initial maxima within approximately 15 days from when the model is initialized, despite the differing snow inputs. This further justifies the choice of 15 days as a "ramp-up" period for uncertainty in Cabaj et al. (2023).

The multi-product-average percent uncertainty is larger than the ERA5 percent uncertainty because it accounts for the inter-

product differences across snowfall input products, reaching a range of 8-18% snow depth uncertainty, and a smaller range of 2-5% uncertainty for snow density. The relatively low percent uncertainty for density may be because the density values are constrained to a relatively narrow range with a maximum prescribed by the model. The multi-product density percent uncertainty is also notably lower than the JRA-55 density percent uncertainty, which suggests that the JRA-55 density data alone has more relative spread compared to all the ensemble data aggregated together. In particular, as seen in Fig. 4, the

JRA-55 uncertainty overlaps considerably with ERA5, and to some extent with MERRA-2. Hence, there is some reduction in the standard deviation when the parameter ensembles are consolidated to construct a single inter-product standard deviation.

### 4.3  Impact of MCMC calibration on snow depth

Figure 6 shows basin-average monthly snow depth climatologies from NESOSIM, illustrating how re-calibrating NESOSIM parameters for each individual reanalysis forcing brings the output snow depths from NESOSIM into better consistency across

the datasets. The average snow depth is lowered somewhat overall, with the multi-product average in April (27.4 cm) now very close in value to the ERA5 end-of-season value (27.3 cm). Given that in both Fig. 6a and b, ERA5 is plotted with its posterior parameters, it follows that the other products have values that more closely match the ERA5 output in Fig. 6b, after the remaining two products have likewise been MCMC-calibrated to the same target observations to which ERA5 was calibrated

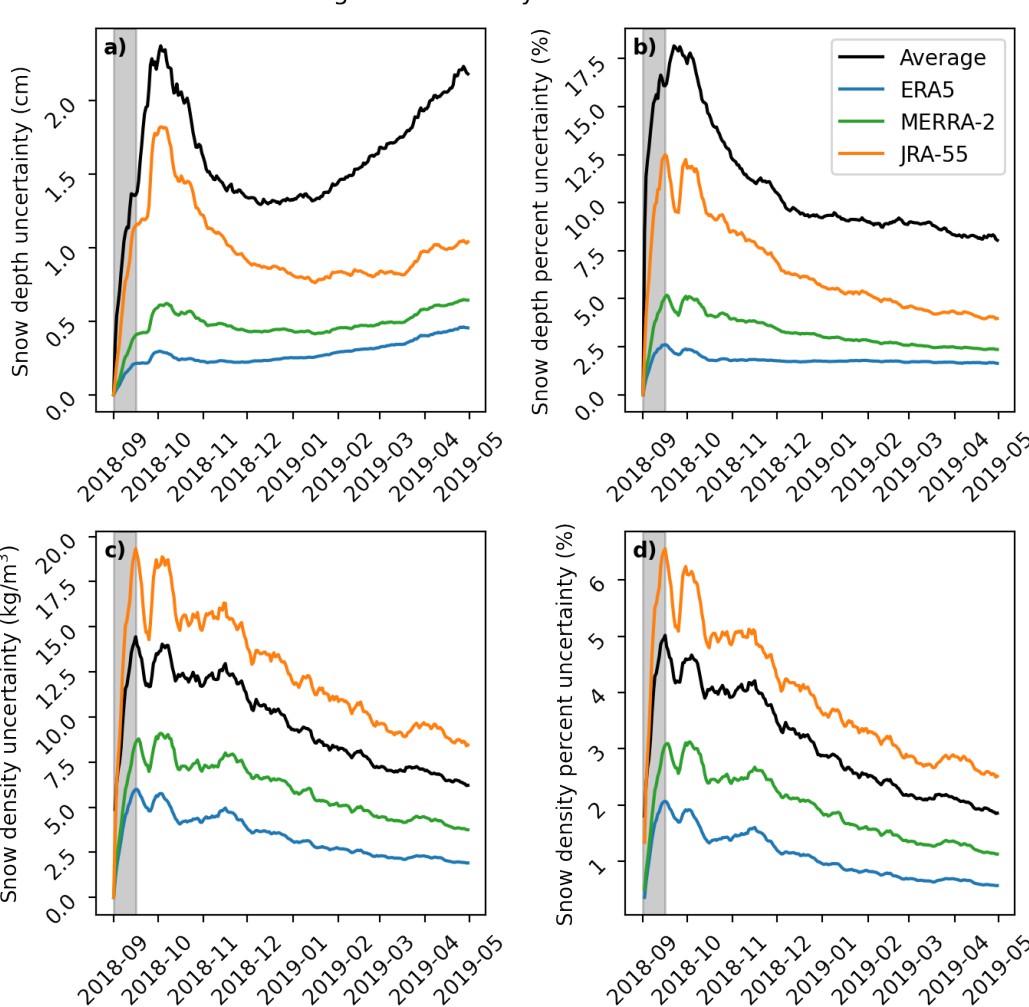

**Figure 5.** Daily uncertainty estimates for snow depth (a,b) and density (c,d) for a single season (2018-2019) of NESOSIM run with all the products separately, and the multi-product average. The absolute uncertainties are shown in (a,c) and percent uncertainties are shown in (b,d). Grey shading indicates the first 15 days of the season.

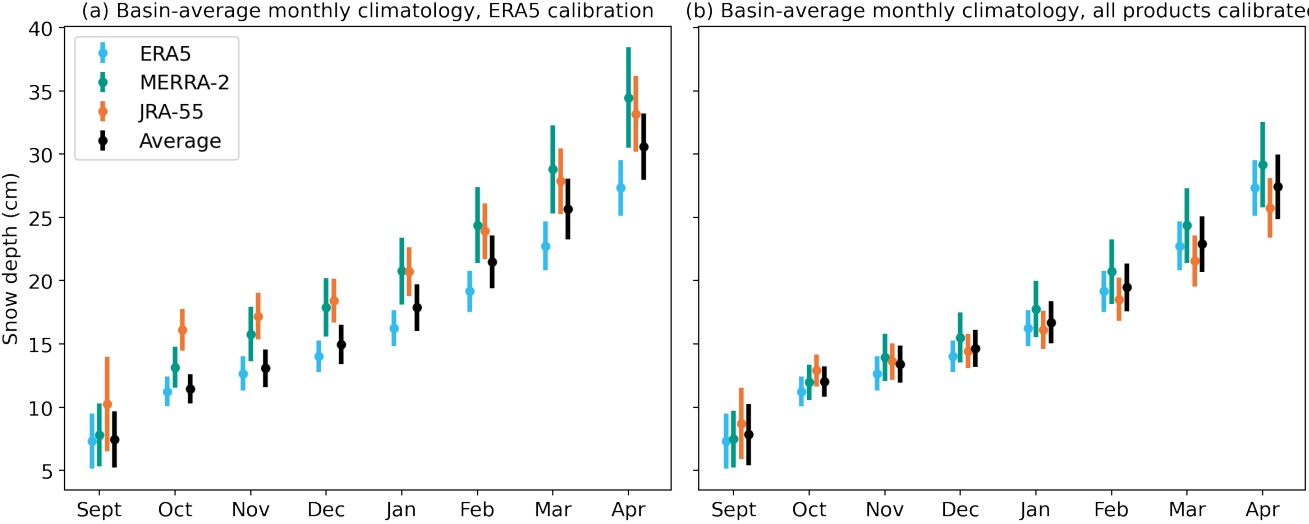

**Figure 6.** Basin-average monthly snow depth climatologies from NESOSIM for 1980-2021, (a) with the model run using the MCMC-ERA5 configuration for all products (i.e. using the same wind packing and blowing snow parameters), and (b) with the parameter values re-tuned to each respective reanalysis input. Bars indicate interannual variability (standard deviation of the climatology), which is also shown separately in Fig. A3.

previously. Some of the relative biases between the products persist; JRA-55 continues to have a relatively large early-season
snow depth which is not seen in the other products, consistent with its early-season snowfall bias. Conversely, at the end of the season, JRA-55 and MERRA-2 previously both exceeded ERA5 at the end of the season, consistent with snowfall biases over sea ice in most regions, particularly over the central Arctic. Following the MCMC calibration, JRA-55 and MERRA-2 bracket ERA5 snow depth on either side, with the multi-product average closely matching the ERA5 values.

The bars in Fig. 6 shows interannual variability of the ERA5-calibrated and individually-calibrated model runs, which is
390 calculated as the standard deviation of the climatology. (These quantities are also plotted separately in Fig. A3). The interannual variability reaches its seasonal peak at the beginning of the season for JRA-55 and ERA5, and at the end of the season for MERRA-2, though the seasonal cycle attains its minimum for all products in October. JRA-55 has the largest interannual variability in September, and in October and onward, MERRA-2 has the largest interannual variability of all the products. MCMC calibration reconciles some of the overall spread in interannual variability between the snow depth outputs, although
there is less agreement in interannual variability between JRA-55 and MERRA-2 in October following the calibration. Both JRA-55's high early-season variability and MERRA-2's high late-season variability decrease somewhat following the MCMC calibration, bringing them into closer agreement with the rest of the products.

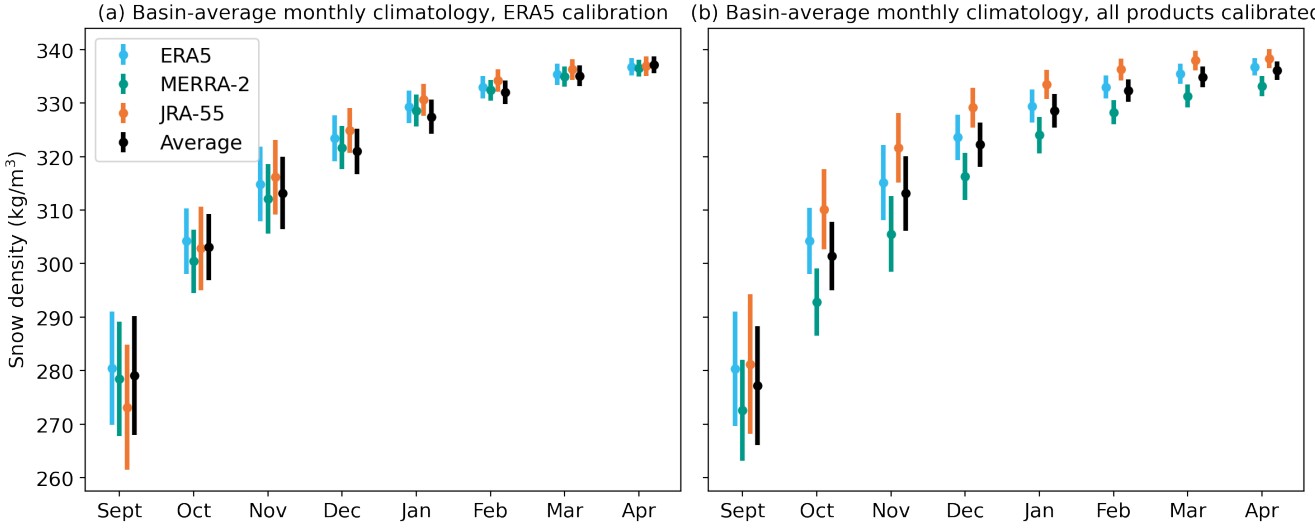

**Figure 7.** Basin-average bulk snow density monthly climatologies for 1980-2021, (a) for the MCMC-ERA5 configuration, (b) with each product calibrated separately. Bars represent the standard deviations of the climatologies, indicating interannual variability.

### 4.4 Impact of MCMC calibration on snow density

Although the MCMC calibration reconciles snow depths for NESOSIM run with different snowfall inputs, the opposite is
400 seen for bulk snow density. Figure 7 shows plots of the basin-average monthly bulk snow density climatologies before and after the calibration. The climatologies show very close agreement when the same (MCMC-ERA5) parameters are used for each NESOSIM run, but differ considerably when the individually-calibrated MCMC parameters are used. This is likely a consequence of how snow density is represented in the model. Since snow density in each layer of NESOSIM is fixed, bulk density is a function of the ratio of snow depths in the two layers. Hence, the bulk density in NESOSIM is strongly sensitive
to the strength of the wind packing process, which transfers snow between the layers. Model runs with different snowfall inputs can still produce similar bulk densities, so long as the same wind packing parameter and wind input are used. The slight differences between the uncalibrated snow density outputs may depend on the timing of snowfall. For example, a high snow accumulation event will reduce the overall bulk density in the short term, but if this accumulation occurs early in the season, more snow may subsequently be transferred to the lower layer, increasing the bulk snow density in the long term if subsequent
accumulation is lower. NESOSIM driven by JRA-55 shows deep snow in the early season relative to other products, which may contribute to its high later-season snow density bias as seen in Fig. 7. Conversely, if the wind packing parameter changes, the modelled density will shift accordingly. The inter-product density differences following MCMC calibration are consistent with the posterior parameter values shown in Fig. 3: the wind packing parameter is largest in JRA-55, which reports the highest bulk snow density value, whereas the smallest wind packing parameter is obtained in the MERRA-2 calibration, which reports
the lowest density value.

The widened spread between products following the calibration also reflects the fact that the density values are relatively under-constrained by the MCMC calibration approach due to the small number of density measurements used. Monthly climatologies of basin-averaged historical density measurements are used as observational constraints for the calibration, due to a relative lack of widespread contemporary density measurements. These density observations are vastly outnumbered by the Operation IceBridge depth observations used in the optimization, which puts more weight on the OIB measurements in the likelihood function. Hence, because the snow depth constraint is stronger, the MCMC calibration will tend to reconcile differences in snow depth while potentially introducing discrepancies in density. Some of this spread may also be related to how the wind packing and blowing snow parameters vary in tandem during the calibration, which may also be a consequence of relatively few density measurements provided. Given that previous work in Cabaj et al. (2023) found that sea ice thickness estimates produced using NESOSIM snow input are more sensitive to snow depth than differences in snow density, we proceed with using the individually-calibrated density values to produce the NESOSIM multi-product-average density, despite their wider spread.

## 4.5 Regional snow-on-sea-ice climatologies

Figure 8 shows regionally-averaged snow depth, and density climatologies by region (with regions as defined in Meier and Stewart (2023)), from NESOSIM-MCMC output, and from SnowModel-LG. Sea ice area calculated from the NOAA/NSIDC Climate Data Record (CDR) product (Peng et al., 2013) is also shown; this product is used in NESOSIM. The sea ice product used in SnowModel-LG differs in that it uses the NASA Team algorithm, whereas the CDR product uses the highest value from the NASA Team and Bootstrap algorithms (Cavalieri et al., 1996; Peng et al., 2013). NESOSIM and SnowModel-LG snow depths agree well in the Central Arctic, Beaufort Sea, and Chukchi Sea regions, but NESOSIM exceeds SnowModel-LG in the Kara, Barents, and East Greenland Sea regions. Notably, NESOSIM shows the greatest snow depths in the East Greenland Sea region, as expected from the presence of the North Atlantic storm track (Webster et al., 2019), but SnowModel-LG records this as a region with much less snow depth ($\sim$27 cm versus $\sim$72 cm in the late-season). Knowing that NESOSIM's simplicity might challenge its realism in high latitude North Atlantic regions like the East Greenland Sea, improved observations of snow on sea ice are critical in such regions. In several peripheral seas of the Arctic Ocean, SnowModel-LG demonstrates a slight leveling off of the snow depth in March and April (as can be seen in the Beaufort, Chukchi, and Kara Seas). Conversely, snow depth in NESOSIM steadily increases in the late-season months in these same regions. This discrepancy may be due to a lack of representation of melt or other snow metamorphosis processes in NESOSIM, since even at high latitudes, some melt is expected at the end of the season. However, other inter-model process differences may also contribute.

For regionally-averaged snow densities from NESOSIM and SnowModel-LG, the limitations of the simple representation of snow density in NESOSIM are apparent, since the density in NESOSIM does not exceed 350 kg/m$^3$, the prescribed maximum density of the model. The seasonal cycles of density in NESOSIM exhibit few regional differences. By contrast, in SnowModel-LG, snow densities and their seasonal cycles vary considerably by region. There is an early-season decline in density in several regions for SnowModel-LG that is not represented in NESOSIM, including the Beaufort, Chukchi, and Kara Seas. The densest snow in SnowModel-LG is present in the East Greenland Sea region, exceeding the maximum snow density possible

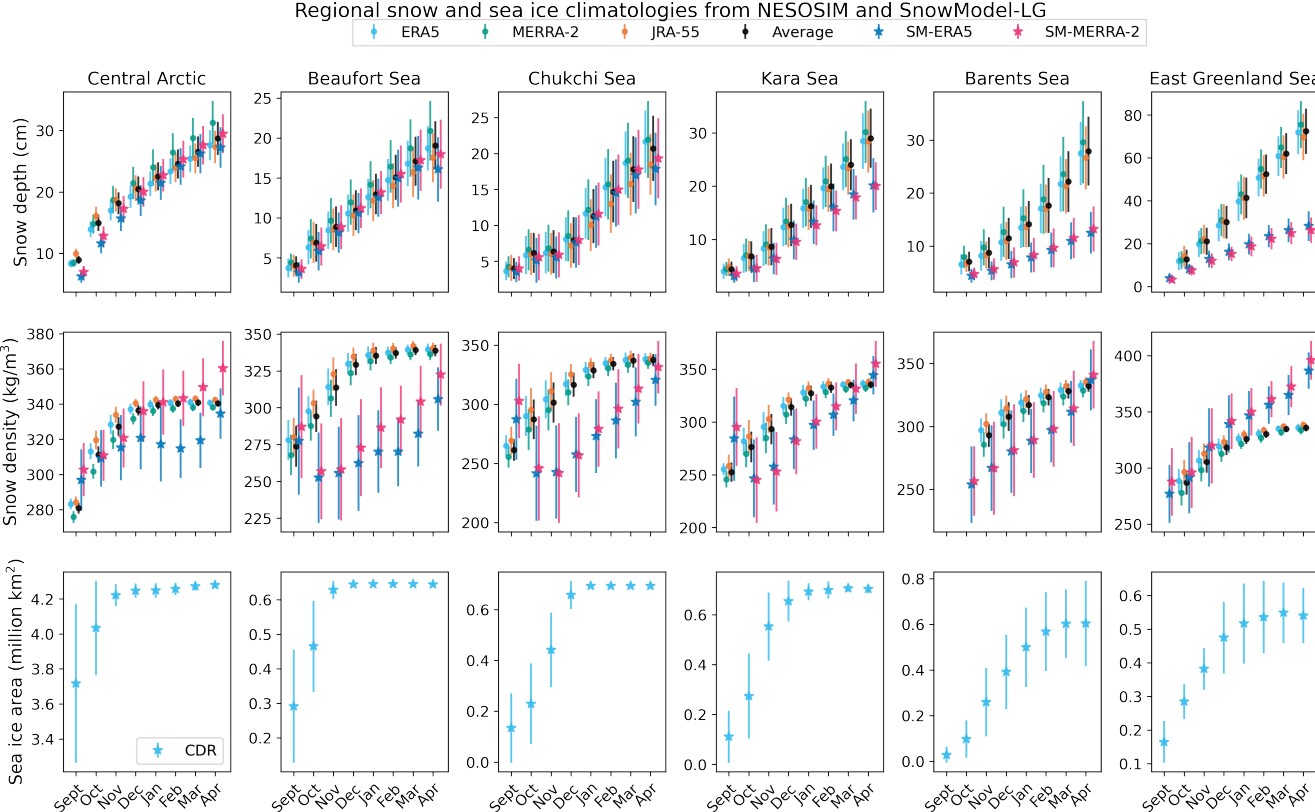

**Figure 8.** Climatologies of regionally-averaged snow depth and density from MCMC-calibrated NESOSIM output and SnowModel-LG output, for 1980-2021. Regional CDR sea ice area climatologies also shown. "Average" indicates the inter-product average for the three NESOSIM configurations. Climatologies from SnowModel-LG driven with ERA5 and MERRA-2 are also shown, with dashed lines. Regions are as described in Fig. A1. Bars indicate interannual variability of each respective climatology, which is quantified by the standard deviation of the climatology.

in NESOSIM as early as January. This high density may explain some of the representational discrepancies for NESOSIM in this region. Because the snow cannot become as dense in NESOSIM, given equal amounts of snowfall input, lower density will yield a deeper snowpack. Nevertheless, differences in snow density representation do not entirely explain differences between SnowModel-LG and NESOSIM. In some regions where NESOSIM has a higher density (e.g. Barents Sea), it likewise has a deeper snowpack.

## 5   Comparison to MOSAiC and IceBird observations

NESOSIM (run with the ERA5-calibrated parameters ("E5config") and individually calibrated parameters ("MCMC")) and SnowModel-LG output is compared to MagnaProbe snow depth (Itkin et al., 2021) and snow density cutter (Macfarlane et al.,

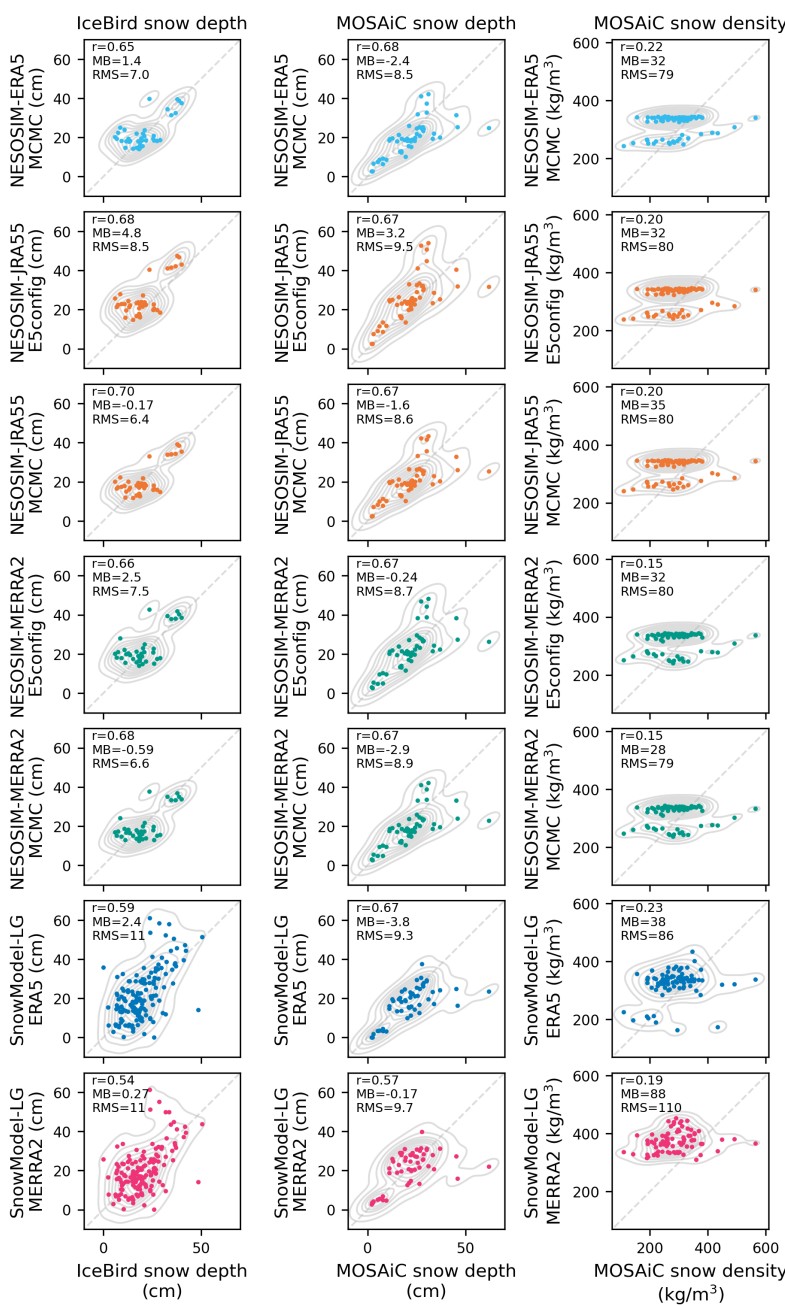

**Figure 9.** Scatter plots with kernel density estimates of NESOSIM and SnowModel-LG output compared to observations aggregated daily to the respective model grids from IceBird airborne snow radar (April 2017 and April 2019) (Jutila et al., 2022; Jutila et al., 2024a, b), MOSAiC MagnaProbe snow depth (2019-2020) (Itkin et al., 2021), and MOSAiC snow density cutters (2019-2020) (Macfarlane et al., 2021, 2022). Correlation (r), mean bias (MB) and root-mean-square difference (RMS) are shown. The 1:1 line is also indicated, for reference.

2021, 2022) observations from the 2019-2020 MOSAiC observational campaign (Macfarlane et al., 2023; Itkin et al., 2023), and to airborne snow radar depth observations from April 2017 and April 2019 from the AWI IceBird observational campaign (Jutila et al., 2022) in Fig. 9. The observed data are aggregated to the respective NESOSIM and SnowModel-LG model grids.

Both NESOSIM and SnowModel-LG snow depths show relatively good agreement with the gridded IceBird and MOSAiC observations, given the considerable differences in spatial and temporal sampling between the modelled and observed values. Overall root-mean-square differences are low (no greater than 10 cm for all products and observations), and mean biases are generally relatively small (absolute value less than 5 cm for all products and observations). Overall observational correlations are somewhat better for NESOSIM than for SnowModel-LG, particularly for SnowModel-LG driven by MERRA-2. Root-mean-square differences are generally comparable between NESOSIM and SnowModel-LG for MOSAiC data, though somewhat lower for NESOSIM when comparing to IceBird. Mean biases for snow depth are largest (in absolute value) for SnowModel-LG driven by ERA5 relative to MOSAiC, and NESOSIM driven by JRA-55 with ERA5 parameters relative to IceBird. Some of the MOSAiC snow depth observations show high values (>60 cm) that are not captured by the models; these observations tend to coincide with measurements made on or near sea ice ridges. Conversely, for some gridded measurements, the models, especially SnowModel-LG, can considerably overestimate snow depths relative to IceBird. Differences in model gridding may also contribute to differences in correlations and biases. Since SnowModel-LG grid has a finer grid than NESOSIM, observations gridded to be compared with NESOSIM are more aggregated and can have fewer extreme values, as is the case for IceBird.

For snow density, SnowModel-LG has more variability than NESOSIM, which is a consequence of the relatively simple representation of snow density in the latter model. Nevertheless, both models have similar challenges with reproducing observed snow density despite the differences in snow density representation between the models, with correlations no greater than 0.23. This disagreement may be partly due to sampling bias in the snow density observations, and the overall difficulty in comparing point measurements to large-scale modelled values. Both models show a high mean bias relative to observed values, with SnowModel-LG driven by MERRA-2 having the largest mean bias, as well as the largest root-mean-square difference. Although NESOSIM has a high bias overall, some snow density measurements were found to exceed 350 kg/m$^3$ (the maximum density the model can represent), highlighting another limitation of the model. Snow density observations from MOSAiC could be used in future work to guide the development of the representation of snow density in NESOSIM to help address this limitation. Several large observed density values (> 400 kg/m$^3$) are not well-represented by either model.

The IceBird and MOSAiC observations have been made available recently, and have not been used to calibrate either NESOSIM or SnowModel-LG. IceBird data captures some regional variability over some regions, and MOSAiC data captures seasonal variability for a single season. Nevertheless, both datasets are limited in their observational coverage, and as such, this comparison provides a limited assessment of the impact of the MCMC calibration. The NESOSIM MCMC calibration improves snow depth agreement slightly relative to IceBird observations, but has mixed results relative to MOSAiC depth and density observations. In general, for snow depth, the effect of the MCMC calibration is to lower the mean bias. This brings the model output into closer agreement with the observations where the model is biased high, as is the case for IceBird, but can degrade agreement if the bias is comparatively low, as is the case for NESOSIM driven by MERRA-2 relative to MOSAiC.

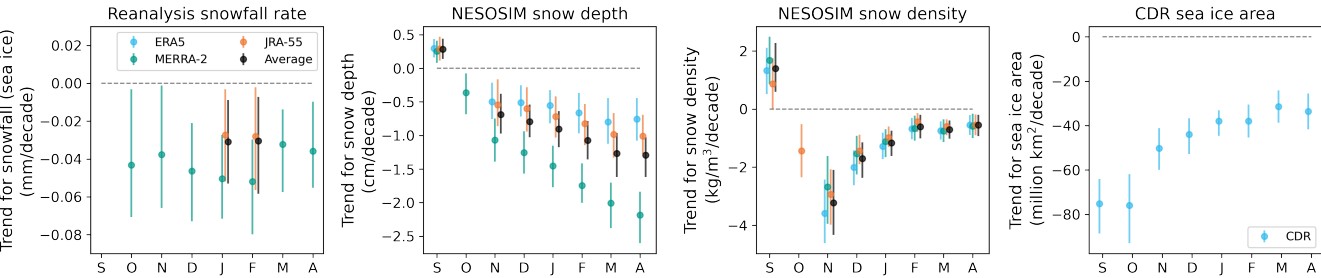

**Figure 10.** Basin-average monthly trends from 1980-2021 for snowfall over sea ice from reanalysis products with CloudSat scaling applied, MCMC-calibrated NESOSIM snow depth and density, and CDR sea ice concentration, calculated using a Theil-Sen trend estimator for all products. "Average" denotes the multi-product average. Error bars indicate a 95% confidence interval as given by the trend estimator; points where there are no trends (the interval overlaps the zero line) are not shown. The grey dashed lines indicate the zero line for reference. SnowModel-LG is excluded from this plot due to differences in model domains.

The correlation is improved for all products relative to IceBird, but is unchanged by the calibration relative to MOSAiC depth and density. The MCMC calibration has mixed impacts on agreement with observed density, but the overall impact of the cal-
ibration is small, likely due to sampling differences and model representational challenges as discussed above. The improved agreement to airborne observations may be partly a consequence of similar observations (from OIB) being used as input to the MCMC calibration.

# 6   Trends in MCMC-calibrated NESOSIM output

Trends were calculated using a Theil-Sen trend estimator, consistent with the approach used by Mudryk et al. (2015). The
Theil-Sen trend estimator produces estimates of trends by finding the median of slopes between all pairs of points in a dataset. This approach allows for the estimation of a trend uncertainty based on a chosen confidence interval; a 95% confidence interval was chosen for this study. In the following discussion, trends are considered significant if the 95% confidence interval does not overlap with zero. If the confidence interval overlaps with zero, we consider there to be no trend.

Basin-average trends from NESOSIM for snowfall over sea ice, snow depth, snow density, and sea ice area are shown in Fig.
10. The trends in snowfall over sea ice are not statistically significant for most products except for a decline for MERRA-2 from October onwards and a decreasing snowfall trend in January and February for JRA-55. The basin-average trends in snow depth from MCMC-calibrated NESOSIM output vary in magnitude by product, but are all broadly similar in sign. MERRA-2 has the strongest trends in the basin-average overall. The trend is found to be negative (declining snow depth) in all months except September, where the trend is significantly positive for all products, and October, where only MERRA-2 shows a (declining)
trend. Snow density trends are generally similar between the products, aside from October, where only JRA-55 shows a trend. Similarity between the snow density trends is expected, since snow density in NESOSIM is less sensitive to snow input, being

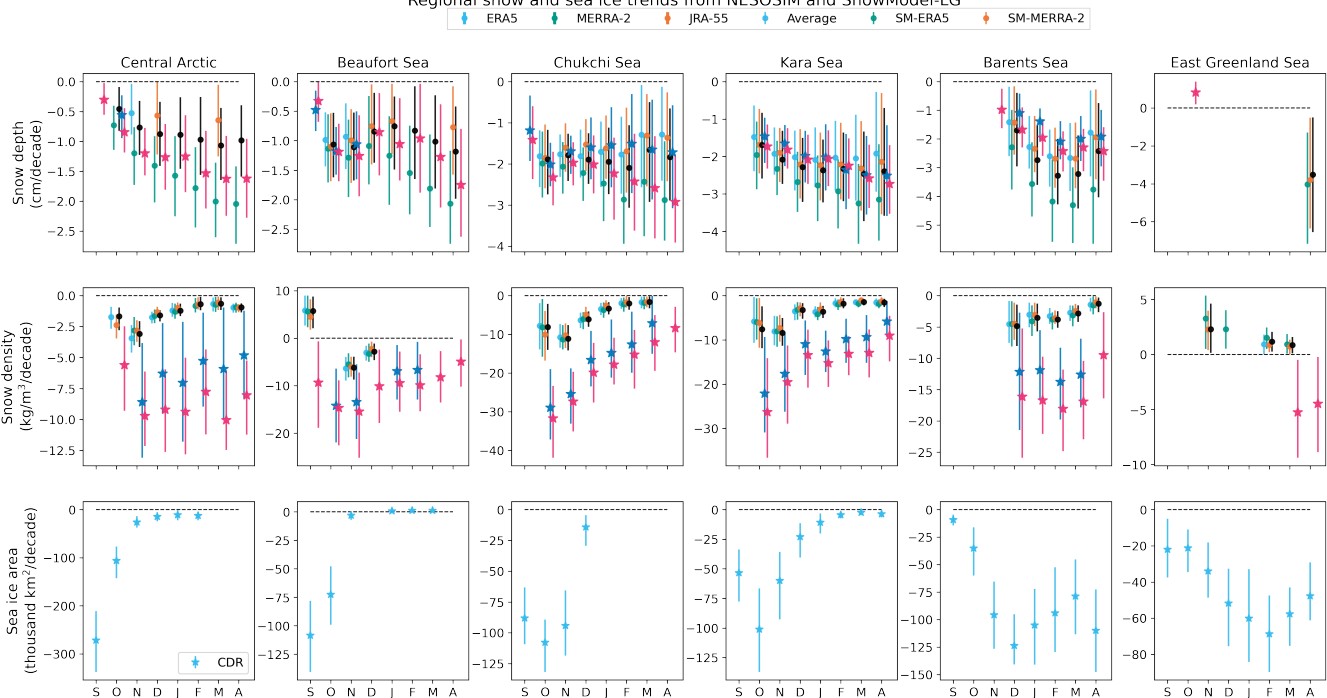

**Figure 11.** Monthly trends for regionally-averaged quantities over the 1980-2021 time period: reanalysis snowfall over sea ice, snow depth, snow density (from NESOSIM and SnowModel-LG), and sea ice area (from the Climate Data Record product). Error bars indicate a 95% confidence interval as given by the trend estimator; points where there are no trends (the interval overlaps the zero line) are not shown.

primarily dependent on wind speed. Since snow density in NESOSIM is limited to the range of 200-350 kg/m$^3$, the density trends may be spuriously low, particularly towards the end of the season, where density values approach the maximum and interannual variability is low (Fig. 7). The comparatively large declining trends in MERRA-2 for depth may result from its

high early-decade snowfall bias relative to the other products. Higher early-year snowfall rates in MERRA-2 can be seen in Fig. 1 and are consistent with findings on Arctic total precipitation in MERRA-2, which is likewise consistently higher in early years (Barrett et al., 2020).

Regional trends in snow depth, snow density, and sea ice area are shown in Fig. 11. Regional trends in snowfall are shown in Fig. A4; although there is regional variation in snowfall trends, most products show no trend for most months, likely due

to high interannual variability of snowfall. A large and significant early-season decline is apparent in the Kara Sea region, but only for the month of September for most products.

Trends in snow depth are generally stronger and more statistically significant than trends in snowfall. Many of the peripheral seas show a significant declining trend for all products from October onward. These trends are consistent with results from Webster et al. (2019), who find delays in sea ice formation particularly in the Chukchi Sea region, and attribute declining snow-

525 on-sea-ice trends partly to the increasingly late sea ice onset in this region. The East Greenland Sea region differs noticeably

from the other regions shown, with no trend for most months except for a slight increase in November for SnowModel-LG driven by MERRA-2, and declines in April only for NESOSIM driven by MERRA-2 and JRA-55. In the Central Arctic region, declines are generally seen only for products driven by MERRA-2. A slight October decline is seen for SnowModel-LG driven by ERA5, a September decline for NESOSIM driven by ERA5, and declines in November and March for NESOSIM driven by JRA-55.

Despite the differences in the snow depth climatologies between NESOSIM and SnowModel-LG, the snow depth trends show considerably more overlap between the two models. This demonstrates how the choice of snowfall input to reanalysis-based snow-on-sea-ice reconstructions can impact the magnitude and significance of derived snow depth trends. In several regions, the strongest declining trends are found in MERRA-2, whereas trends often tend to be smaller or absent for ERA5, for both NESOSIM and SnowModel-LG.

For snow density trends, inter-model differences tend to be larger than inter-product differences. Declining trends are largest around October-November for most products and regions, except in the Barents and East Greenland Seas. The East Greenland Sea region shows increases in snow density for some months and products. NESOSIM and SnowModel-LG disagree on the sign of the snow density trend for September in the Beaufort Sea, and for March in the East Greenland Sea for products driven by MERRA-2. Overall, densities in SnowModel-LG tend to show large declines relative to NESOSIM. As discussed previously, NESOSIM end-of-season density trends may be spuriously small due to NESOSIM snow densities approaching their maximum towards the end of the season, although end-of-season density trends as represented in SnowModel-LG also tend to be smaller.

Sea ice area trends vary by region, but strong declines are found for at least part of the season in all regions shown. In the Central Arctic and the Siberian sector, as well as the Beaufort Sea, the largest declining trends are in the earlier months of the cold season. (Larger trends may be present in months outside of the NESOSIM study period.) When sea ice in these regions attains its maximum extent, the trends largely vanish, suggesting a persistent cold-season cover. Towards the North Atlantic (Barents, East Greenland), larger declines are seen in later months.

To provide a more regional perspective on snow trends, maps of snow depth trends in NESOSIM and SnowModel-LG output are shown in Fig. 12. Corresponding snowfall trends are shown in Supporting Fig. A5. For these plots, trends were also calculated using a Theil-Sen estimator, but only grid squares containing at least 20 years of values were included to exclude spurious trends. Consistently with results from the regional monthly trend plots, there is a lack of snowfall trends over most of the Arctic basin, due to the high interannual variability of Arctic snowfall relative to the magnitude of the trends. Slight increases are seen in the Barents Sea and in the Sea of Okhotsk, and decreasing trends are seen east of Greenland for all products. The depth trends are more robust, highlighting a decline in the peripheral seas consistent with the results shown in the regional plots, as well as some slight declines around Hudson Bay and Labrador Sea. Some significant increasing depth trends north of the Beaufort Sea are found in SnowModel-LG driven by ERA5, as well as in NESOSIM driven by ERA5 and JRA-55, though the products differ on the existence of the increasing trend near the North Pole. The spatial pattern of increasing trends north of Greenland and the Canadian Arctic Archipelago and decreasing trends elsewhere is consistent with the pattern of springtime trends found by other studies, including Webster et al. (2019) and Zhou et al. (2021), although the spatial extent

March snow depth trend (cm/y)

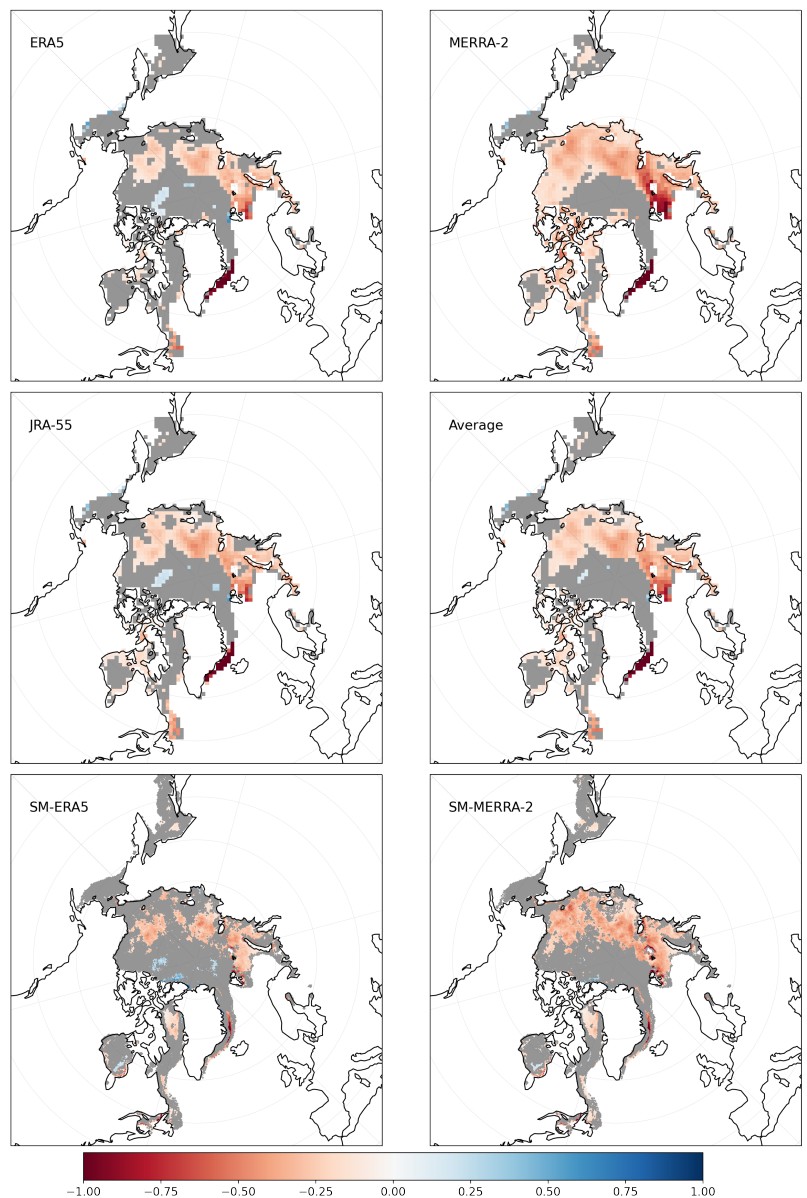

**Figure 12.** Snow depth trend maps for March 1980-2021 from NESOSIM-MCMC run with snowfall input from ERA5, MERRA-2, and JRA-55, and SnowModel-LG with snowfall input from ERA5 and MERRA-2. The snow depth is output from NESOSIM with parameters specific to each separate reanalysis product. The trend in the average of the output of the three NESOSIM runs is also plotted (Average). Regions with no trends (not significant to a 95% confidence interval) are shaded in grey. Note that SnowModel-LG is not provided within the Canadian Arctic Archipelago, so data from that region is absent in this map.

of the significant trends differs. Some differences are expected, since the other studies mentioned examine different months and time periods. There is broad consistency, however, in the declining trend found in the Barents Sea region. The overall large declining trends in depth derived from MERRA-2 are particularly apparent in Fig. 12. ERA5 and JRA-55 agree better with each other on the spatial pattern of the snow depth trends compared to MERRA-2.

The impact of model resolution is apparent, since some of the strong trends seen in the SnowModel-LG output are highly localized. There are small but significant increases in snow depth in Hudson Bay that are absent from the NESOSIM output, and some increases east of Greenland towards the Fram Strait that are not apparent in the NESOSIM output. This highlights the continued need for further analysis of snow on sea ice in these regions, as well as a need for further observations to validate models in these difficult-to-characterize regions. Nevertheless, the broad patterns of trends between NESOSIM and

SnowModel-LG are similar, and there is good agreement between NESOSIM and SnowModel-LG for the Central Arctic and adjacent regions.

## 7    Discussion

The results of this study highlight the value of producing a snow-on-sea-ice product that accounts for uncertainty in model input and formulation, and sparse observations. We find that snowfall climatologies differ considerably between the ice-covered

region of the Arctic Ocean and the full ice-plus-ocean region. This result is expected, given that sea ice controls atmosphere-ocean moisture and heat fluxes, which in turn influence high-latitude precipitation. In particular, cumuliform snowfall observed by CloudSat has been shown to vary seasonally with sea ice cover (Kulie and Milani, 2017; Kulie et al., 2016). This impacts the CloudSat scaling, which performs well over the ocean basin, but has more difficulty reconciling snowfall over sea ice in some cases, particularly in September where the largest basin-average accumulation takes place as seen in Fig. 2. This was

not considered in Cabaj et al. (2020). The CloudSat scaling factors applied to NESOSIM were calculated over ocean regions, including both sea-ice-free and sea-ice-covered regions, with new factors calculated for JRA-55. Although the scaling of model snowfall input to CloudSat reconciled inter-product differences in snow depth for NESOSIM v1.0, inconsistencies remain in NESOSIM v1.1 snow depth output even with the application of the CloudSat calibration. In general, the CloudSat scaling performs best in aggregate over larger regions, since smaller regions may have high variability which may not necessarily

be accounted for by the scaling. Re-calculating the CloudSat scaling factors masked exclusively over sea ice may not be feasible due to the relative lack of CloudSat measurements over sea ice. In regions such as the Greenland Sea where sea ice is present only in a very narrow region along the coast, CloudSat reports much less cold-season snowfall relative to the reanalysis products, which suggests that CloudSat may not be adequately sampling snowfall events in the region. This is illustrated in Supporting Fig. A6, where CloudSat fails to reproduce the climatology and interannual variability found in the reanalysis

products when restricted to over-ice observations in the Greenland/Norwegian Sea region, despite agreeing comparatively better with reanalysis products in other regions. As a result, constructing scaling factors using CloudSat restricted over the ice-covered region yields excessively low (< 18 cm) basin-average snow depths. Hence, in this work, CloudSat scaling factors are calculated based on snowfall over land-free regions, including both open ocean and ice-covered ocean. Nevertheless,

future work may entail some revision of the existing CloudSat scaling factors over sea ice, particularly for JRA-55. A more regionally-refined calibration may be appropriate, with the caveat that aggregating CloudSat observations over smaller regions may introduce additional uncertainty due to observational undersampling, as is likely the case with the Greenland/Norwegian Sea region. Refining the calibration using more contemporary forthcoming snowfall measurements from satellite missions such as EarthCARE (Wehr et al., 2023) and using more sophisticated calibration techniques may be other options for future work.

This study also investigates calibration of NESOSIM wind packing and blowing snow parameters using an MCMC process when different reanalysis products are used for NESOSIM snowfall input. The MCMC parameter tuning is dependent on the choice of snowfall input to NESOSIM. Given the discrepancies between the NESOSIM output products prior to the MCMC calibration, and the fact that they are all being calibrated to the same observational target, it is unsurprising that the posterior parameter distributions obtained from the calibration differ. This has some implications for the model physics, since it suggests that the representation of the physics in the calibrated model is highly dependent on the input. Caution must be taken, then, when interpreting the wind packing and blowing snow parameters at face value, because wide ranges of these parameters can produce physically reasonable model output. This also has implications for other reanalysis-based snow-on-sea-ice estimates, which tend to make use of a selected reanalysis product. As with NESOSIM, snow depth and density in SnowModel-LG are dependent on the choice of reanalysis input, even with the corrections applied in SnowModel-LG to the reanalysis snowfall inputs used (Liston et al., 2020). Using automated model parameter recalibration when changing snowfall inputs used for NESOSIM and other snow-on-sea-ice models provides an objective means to address this issue.

The MCMC calibration improves agreement in both the magnitude and the interannual variability of the snow depths output from NESOSIM forced with different reanalysis snowfall products, but it reduces agreement in snow density. This is likely a consequence of two factors: the relative lack of density constraints in the MCMC calibration, and the snow density being more sensitive to the wind packing factor than to the snow inputs examined in this study. With different snow inputs, when the wind packing and blowing snow parameters were the same for all runs, there was relatively minimal variation in the density. Given that the density does not depend on the total snow depth, but rather, on the proportion of snow in each layer, one expects the density to be relatively insensitive to snow input and more sensitive to differences in the parameters. Nevertheless, the lack of density constraints in the MCMC calibration may also be an influence, since if density were more strongly constrained, the parameters would be optimized to produce output with a narrower spread in density between the products. Despite this, the estimated density uncertainty in the multi-product average is also quite low, highlighting how the densities produced by NESOSIM are limited to a relatively narrow range due to constraints imposed by the model itself. Since other models produce higher densities (as seen in the comparison with SnowModel-LG), and observations indicate the presence of denser snow than what can be produced by NESOSIM (King et al., 2020; Itkin et al., 2023; Macfarlane et al., 2023), the density assumptions in NESOSIM may need to be revisited. The matter of scale must also be considered, because density measurements are highly localized, and NESOSIM represents the bulk density over large regions, consistent with its coarse resolution. Overall, this result highlights the need for including additional observational density constraints in the calibration, and an eventual reformulation of NESOSIM's representation of density.

In regional comparisons with SnowModel-LG, NESOSIM snow depth is found to generally agree over the Central Arctic region, but agreement is weaker in the peripheral seas, particularly towards the end of the sea ice season. Overall, NESOSIM snow depth tends to be biased high relative to SnowModel-LG. SnowModel-LG includes loss processes not currently captured by NESOSIM, such as snow melt, which may contribute to the inter-product differences. NESOSIM and SnowModel-LG also have different approaches for addressing sea ice drift. The Lagrangian approach of SnowModel-LG allows it to track individual ice parcels, providing a more complex representation of the contribution of sea ice dynamics to the snow budget. NESOSIM presents a more simplified approach, but is less computationally expensive to run. Limitations in NESOSIM's representation of snow density may also impact the agreement with SnowModel-LG; snow density in NESOSIM is limited to a maximum value of 350 kg/m$^3$, whereas it can attain larger values in SnowModel-LG. NESOSIM snow depth is biased especially high relative to SnowModel-LG in the East Greenland Sea region, where its density conversely has a low bias. SnowModel-LG includes snow grain parameterizations that are absent in NESOSIM, which allows for the representation of processes that may be essential to quantifying variations in snow density. Nevertheless, discrepancies in snow density representation are likely not the sole explanatory factor for inter-model differences, since in some regions, NESOSIM is biased high relative to SnowModel-LG even while the densities are more comparable.

Despite the differences between NESOSIM and SnowModel-LG in terms of model processes and complexity, the models generally perform comparably in comparisons with in situ MagnaProbe snow depth (Itkin et al., 2021) and density cutter (Macfarlane et al., 2021, 2022) observational data from the 2019-2020 MOSAiC observational campaign, though some larger inter-model differences are seen in comparison with April 2017 and 2019 IceBird snow radar observations (Jutila et al., 2024a, b; Jutila et al., 2022). The highly localized nature of the in situ measurements from MOSAiC makes comparisons with comparatively coarse gridded model output challenging. Both models represent snow depth relatively well with respect to MOSAiC and IceBird (correlations upwards of 0.54 and RMSD < 11 cm), but have more difficulty representing observed snow density (correlations < 0.23). SnowModel-LG driven by MERRA-2 is notable for having the lowest snow depth correlation and the highest root-mean-square difference relative to MOSAiC and IceBird, but also the smallest mean bias compared to all other products for MOSAiC. The root-mean-square difference and mean bias for snow density are also high for this product, which is biased high overall and does not reproduce any lower density values coincident with the observations. This highlights how even with broad-scale observational adjustments such as those made to the reanalysis input to SnowModel-LG, the use of different snow forcings can still have a large impact on model output. The MCMC calibration has mixed impacts on NESOSIM agreement with MOSAiC, and the overall magnitude of the impacts is small. However, there is slight improvement to correlations and biases relative to IceBird. The observational data used in the MCMC calibration differs considerably from MOSAiC, since it includes sparser observational data from multiple years, and, in the case of OIB measurements, spanning larger spatial extents. The measurement approach of IceBird more closely resembles that of OIB; both being airborne campaigns making measurements in April. Some of the differences between observed and modelled values are likely a consequence of sampling bias. MOSAiC, for example, includes a variety of measurements made on different days which were made with the intention of sampling various snow conditions. Thus, MOSAiC measurements show a high degree of variability compared to modelled values, especially as different observational sites change location throughout the season (Itkin et al., 2023). Given the differ-

ing scales involved, it is unsurprising that the current configurations of the models can have difficulty reproducing outlying observed values. Model simplifications likely also play a role in the discrepancies, such as in the case of NESOSIM snow density. Since snow in NESOSIM cannot be removed from the lower layer (for a given grid cell, it can only decrease as a consequence of sea ice motion), end-of-season densities are expected to approach 350 kg/m$^3$ as an increasing proportion of the snow in each grid cell is old (lower-layer) snow. This yields a limited range of possible representations of snow values, and a large number of snow densities near 350 kg/m$^3$ as seen in Fig. 9. Future work could explore using IceBird and MOSAiC measurements to help constrain NESOSIM model output and, in the case of MOSAiC, guide the improved representation of NESOSIM snow density. Caution would be necessary to avoid overfitting, given the spatial and temporal limitations of the observations. Nevertheless, high-quality contemporary airborne and in situ observations of snow on sea ice such as those from IceBird and MOSAiC provide a new path to to assess and improve model representation of snow on sea ice.

There are many possibilities for further refinements to NESOSIM and to the MCMC calibration approach. NESOSIM is a comparatively low-resolution model with highly parameterized snow processes. This makes it computationally inexpensive, facilitating both MCMC optimization and the rapid production of ice thickness estimates. A key advancement presented by the MCMC calibration lies in the ease of quickly generating updated parameter estimates as new input products are introduced. Additional reanalysis products not examined in this study could be investigated as possible additional inputs to NESOSIM, especially when existing products are inevitably superseded by updated versions. The MCMC calibration itself could also be adjusted; future work could investigate the use of additional observational constraints on snow depth and especially on snow density. High-quality contemporary snow density observations exist from several observational campaigns such as MOSAiC (Wagner et al., 2022), and although the measurements available during a single day may be highly localized, making use of these observations with an appropriate uncertainty estimate could help better constrain the wind packing parameter, yielding more representative estimates of snow on Arctic sea ice. With the inclusion of additional observational constraints, calibration of additional model parameters in NESOSIM could also be explored. Future work could also investigate impacts of a similar calibration using more complex models, or added complexity in NESOSIM. Several processes, including snow redistribution by wind, sublimation, and melt and refreeze processes are simplified in NESOSIM. Given the warming conditions in the Arctic, it may be particularly beneficial to represent melt processes in NESOSIM. NESOSIM is run over the September-April period to exclude the melt season, but melt also occurs throughout the year, with a trend towards earlier onset (Stroeve and Notz, 2018). In the current calibration process, observed melt may hence be misrepresented in NESOSIM as a decrease in depth due to densification or blowing snow. Future work could explore parameterizations of additional processes, such as sublimation and snow redistribution by wind to improve snow depth and density estimates across a variety of environmental conditions.

The relatively coarse resolution of NESOSIM may impact its representativeness, since some snow-on-sea-ice processes operate on very small spatial scales and short timescales. The sea ice advection and divergence processes in NESOSIM represent a spatially-averaged tendency of snow to be redistributed with sea ice motion, but may fail to capture small-scale effects from localized ridging and small-scale leads often seen in observational studies (Itkin et al., 2023; Macfarlane et al., 2023). The amount of blowing-snow loss due to leads has been observed to be influenced by strong winds and warm air temperatures from Arctic cyclone events, which may be challenging to capture in the current configuration of NESOSIM (Clemens-Sewall et al.,

2023). The coarse time resolution also limits the model's ability to capture rapid changes in snow depth due to short-term accu-
mulation events. In a broader modelling context, high-resolution modelling may be necessary to adequately capture small-scale

processes (Lecomte et al., 2015). NESOSIM could be run at a higher resolution to take advantage of the higher resolution of
available drift products to better capture the influence of sea ice motion. However, sub-gridscale parameterization would still
be necessary to better capture smaller-scale effects. The representation of sea ice differs between reanalysis products, and may
not be coincident with the observational sea ice concentration used as input to NESOSIM in this work. This, in conjunction
with regridding, may introduce some artefacts in regions of marginal sea ice cover such as the Greenland Sea region.

The ERA5 wind product was used in all configurations in this study to isolate the contribution of snowfall to NESOSIM,
since snowfall is the primary input to the NESOSIM budget. In observational comparisons in the Arctic, ERA5 has been
found to perform relatively well compared to other reanalysis products, including JRA-55 and MERRA-2, which motivates the
choice of ERA5 over other products (Graham et al., 2019a, b). However, the choice of reanalysis wind input may also have an
impact on NESOSIM output. The wind packing and blowing snow processes take effect only when wind speed exceeds the 5

710 m/s wind action threshold. If wind speeds from different input products are on differing sides of the threshold, wind-related
snow processes may take effect at a given location and time for one product and not another. The strength of the blowing
snow process is also dependent on wind speed. Future work could investigate the impact of differing wind input products to
NESOSIM.

Uncertainties derived from the MCMC parameter uncertainty for each product reflect the widths of the posterior distributions

produced from the MCMC process. The percent uncertainties in the model depth and density output are considerably smaller
than the percent uncertainty of the posterior parameters (expressed as coefficients of variation). This is consistent with the
result in Cabaj et al. (2023) that highlights the relative insensitivity of NESOSIM to the model parameters. However, the snow
depth and density uncertainties for NESOSIM run with MERRA-2 and JRA55 are larger than the uncertainties for NESOSIM-
ERA5 alone, and likewise, the aggregated multi-product uncertainties exceed the MCMC-ERA5 values. This highlights the

720 value of accounting for uncertainties due to differences in reanalysis input products. The multi-product snow depth uncertainty
spans a more reasonable 8-18% range compared to the <3% uncertainty of MCMC-ERA5 alone. The estimated snow density
uncertainty is relatively small, particularly when compared to the 40 kg/m$^3$ uncertainty prescribed for the ICESat-2 product
(Petty et al., 2020) based on the in-situ snow observations compiled and analyzed by Warren et al. (1999). The uncertainty
estimated for NESOSIM in this study is likely underestimated due to the limited density range represented by the model. Near

the end of the season, the densities in many grid cells may be near the maximum 350 kg/m$^3$ density value, limiting the possible
density variation and thus decreasing the spread in parameter-ensemble density values. This calls into question the assumption
of Gaussian uncertainty distributions, and it may be beneficial to revisit the calibration with non-symmetrical distributions in
future work. An analogous underestimate of uncertainty takes place in the early season, where uncertainties are artificially low
due to the common point of initialization for all the parameter ensemble runs. However, in general, by around the 15th model

730 day, the percent uncertainty saturates, and moreover, this saturation is observed not just for ERA5, (as in Cabaj et al. (2023),)
but for each individual product. Overall, the estimated uncertainties in the NESOSIM-MCMC-average product must be treated

with caution, since they do not fully characterize all sources of uncertainty, but they can be used to provide a more robust estimate of uncertainty from the NESOSIM model input and calibration assumptions.

Inter-product differences in snow depth and density may have substantial impacts on estimates of sea ice thickness from sea ice altimetry measurements. For example, given representative values of lidar freeboard, and representative densities of snow, ice, and water, if snow depth estimates with a 5 cm difference are used to estimate sea ice thickness, the difference in derived sea ice thickness can be as large as 30 cm (Giles et al., 2007). Thus, if trends differ between snow products, trends in derived sea ice thickness will be impacted as well. For sea ice freeboard, a snow product with a decreasing trend would impose an increasing derived ice thickness trend on top of any trend in the freeboard itself. Interannual variability in snow was found to strongly influence sea ice volume derived from CryoSat-2 altimetry measurements (Bunzel et al., 2018). Hence, differing snow depth trends (or lack thereof) between products could lead to differing conclusions on trends in derived sea ice thickness.

Intercomparisons of reanalysis products (and quantities derived from them) have some associated caveats. Insight into Arctic precipitation can be gained from the analysis of reanalysis precipitation trends (Boisvert et al., 2018), but caution may be necessary in their interpretation due to discontinuities in the assimilation (Barrett et al., 2020), although contemporary reanalysis systems include bias corrections that mitigate some of the issues introduced from these discontinuities. It is important to consider inter-product differences due to reanalysis inputs because differing snow depth trends between model outputs may have impacts on climate sensitivity estimates due to influences on sea ice. Coupled climate model simulations have found contrasting climate impacts of snow on Arctic sea ice due to competing influences on congelation sea ice growth and surface melt (Holland et al., 2021), but snow-free summers may increase sea ice melt (Webster et al., 2021). Thus, by influencing sea ice thickness, a declining snow depth trend could influence trends in atmosphere-ice heat fluxes, which in turn could influence sea ice extent and other climate variables. There are also discrepancies between the representation of sea ice cover in the three reanalysis products used, which may yield larger inter-product differences particularly in regions with thinner sea ice (Barrett et al., 2020). Averaging multiple data products together is a well-established approach, and the development of blended land snow products motivates this study (Mudryk et al., 2015, 2018). Constructing a multi-product average using a wider range of input products, and incorporating other models and observations, could be of future interest.

## 8 Conclusions

Quantifying snow on Arctic sea ice is an ongoing challenge, and existing approaches face difficulties due to spatial and temporal sampling discrepancies, relative biases, and the sparse availability of in situ validation data. Nevertheless, NESOSIM has free parameters which can be observationally calibrated for different snowfall inputs to reconcile inter-product biases. Averaging together model outputs run with different snow inputs can also account for relative differences between the products, and thus, we construct a snow-on-sea-ice product that averages the output of NESOSIM with calibrated snowfall input from ERA5, JRA-55, and MERRA-2, after calibrating each model output to depth and density observations using an MCMC process.

MCMC calibration of NESOSIM with different snowfall inputs following the approach in Cabaj et al. (2023) reconciles differences in snow depth between NESOSIM run with different reanalysis inputs, but enhances differences in snow density.

The posterior parameter distributions obtained from the calibration differ between the products, with JRA-55 yielding the largest values for the wind packing and blowing snow parameters, and yielding posterior parameter distributions with the largest spread compared to those of the other two products.

When MCMC-calibrated regionally-aggregated NESOSIM monthly climatologies from 1980-2021 are compared to SnowModel-LG, good agreement in snow depth is found in the Central Arctic Ocean and nearby regions, though NESOSIM has a high bias relative to SnowModel-LG in more peripheral regions. Snow densities differ greatly between NESOSIM and SnowModel-LG, both in magnitude and seasonality, likely as a consequence of the comparatively simpler representation of snow density in NE-SOSIM and the weaker constraints in the calibration, although it is challenging to ascertain accuracy in this poorly-observed quantity. NESOSIM and SnowModel-LG compare similarly to airborne snow depth measurements from the 2017 and 2019 IceBird campaigns and in situ snow depth and density measurements from the 2019-2020 MOSAiC observational campaign, and both models have challenges with representing snow density in particular. The discrepancies may be partly due to insufficient representation of physical processes in models, but also likely result from differences in spatial scales and in timing between the modelled and observed values, as well as measurement sampling biases.

Trends in MCMC-calibrated NESOSIM run with the different products over the 1980-2021 time period broadly agree, with decreasing trends being strongest for NESOSIM run with MERRA-2 snow input. Basin-average snow depth on Arctic sea ice from NESOSIM is declining in most months and for most products except in September, where there is a slight increasing trend, and trends are not statistically significant in October except for MERRA-2, where there is a modest decline. Regional snow depth trends vary in magnitude and statistical significance, but most regions show a declining trend in snow depth on sea ice over the mid-to-late season. In regions where climatologies may disagree between models, snow depth trends can nevertheless more closely agree between models. The choice of reanalysis snow input can greatly impact the magnitude and statistical significance of snow depth trends, and thus, trends derived from reanalysis-based reconstructions of snow on sea ice must be treated with caution. In general, when using reanalysis-based reconstructions of snow on sea ice, the impact of reanalysis input must be accounted for, since changing the reanalysis input may yield less representative model processes.

The uncertainties from MCMC-ERA5 are low relative to other products, and combining uncertainties from the MCMC calibrations for additional reanalysis products yields a more reasonable estimate of basin-average snow depth uncertainty, accounting for uncertainties due to model parameter calibration and choice of reanalysis input. Estimates of snow density uncertainty remain relatively low, likely due to the implicit constraints on snow density imposed in the model, since the fixed density values in each layer impose minimum and maximum density values.

Overall, the findings in this study motivate the continued need for widespread in situ observations of snow on Arctic sea ice, particularly for snow density. In the meantime, however, synthesizing existing models and observations can help provide best-guess estimates of snow on Arctic sea ice. The consensus snow depth product produced in this work incorporates uncertainties from both reanalysis and parameter uncertainties, albeit limited by the simplicity of NESOSIM. Future work in synthesizing models and observations could entail incorporating additional observations and reanalysis products, or possibly applying similar calibration and blending approaches to other snow-on-sea-ice products.

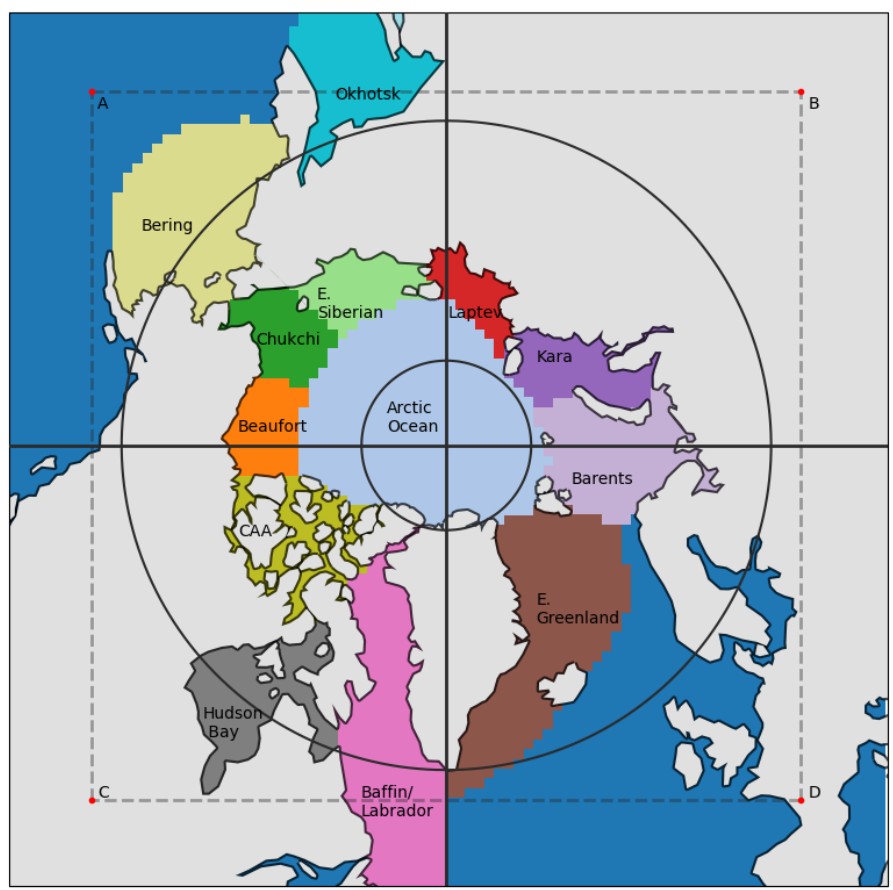

**Figure A1.** Map of the NESOSIM v1.1 model domain, with NSIDC-defined regions (Meier and Stewart, 2023) investigated in this study identified. The lower (60°N) and upper (82°N) limits of the latitude band from which CloudSat measurements were aggregated for the CloudSat climatology scaling are indicated with dark grey contours. The quadrants of the domain used for the CloudSat scaling as applied to the NESOSIM reanalysis input are likewise indicated. Dashed lines indicate the bounds of the model domain for the previous version of NESOSIM, and A-D indicate the corner points from which the CloudSat scaling factors are linearly interpolated.

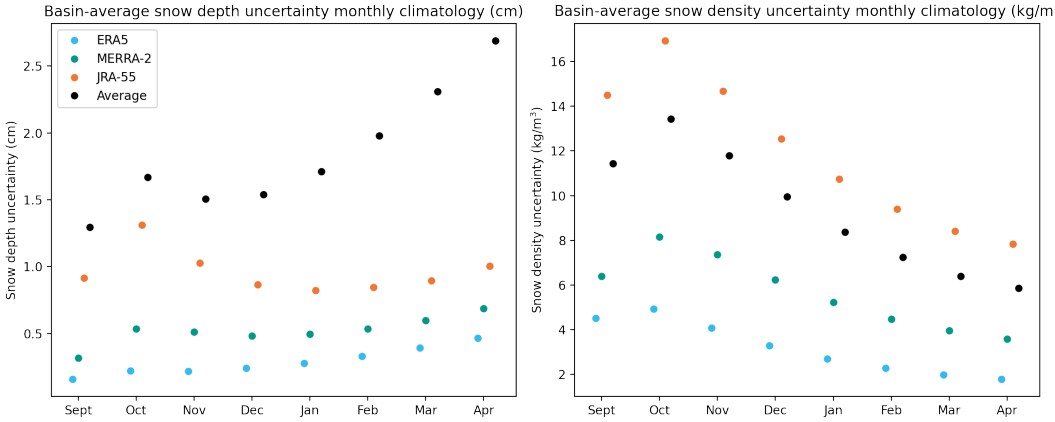

**Figure A2.** Basin-average monthly climatologies (1980-2021) of estimated uncertainties for NESOSIM snow depth and density for all products separately, and the multi-product average.

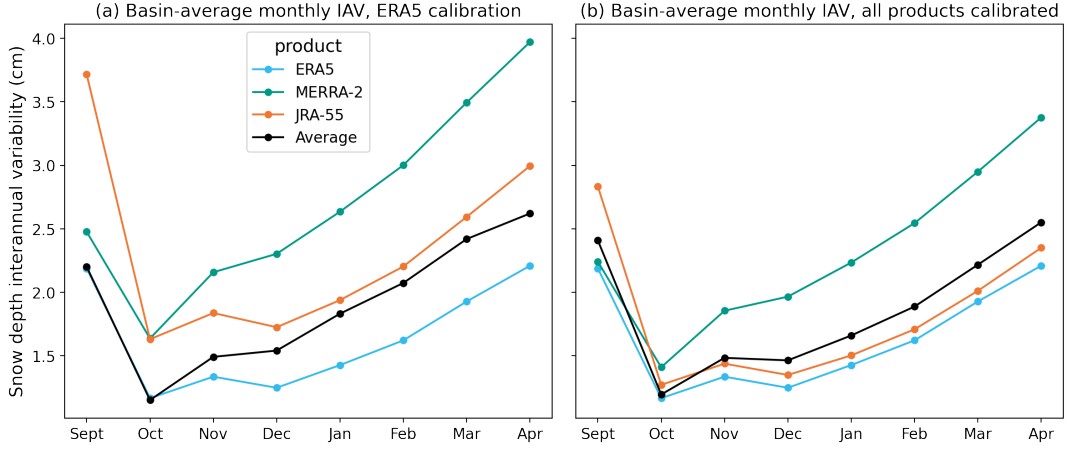

**Figure A3.** Interannual variability (IAV) of snow depth for NESOSIM, as shown in the bars in Fig. 6 (a) with MCMC-ERA5 calibration, (b) with all products calibrated. This value is calculated as the standard deviation over the time period for the monthly mean for each given month.

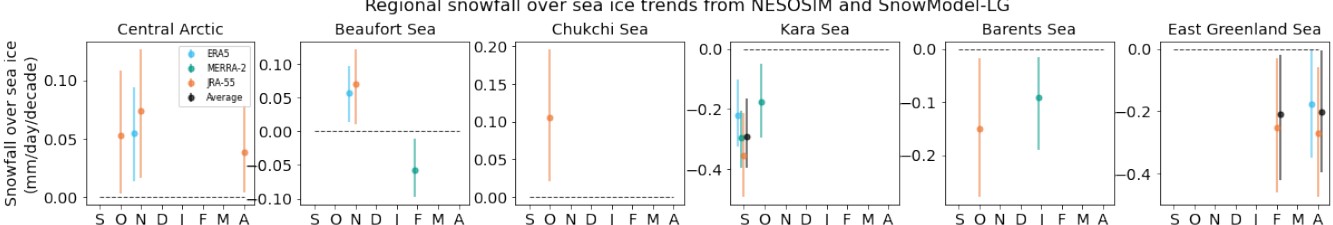

**Figure A4.** Monthly trends for regionally-averaged reanalysis snowfall over sea ice quantities over the 1980-2021 time period. Bars indicate 95% confidence intervals.

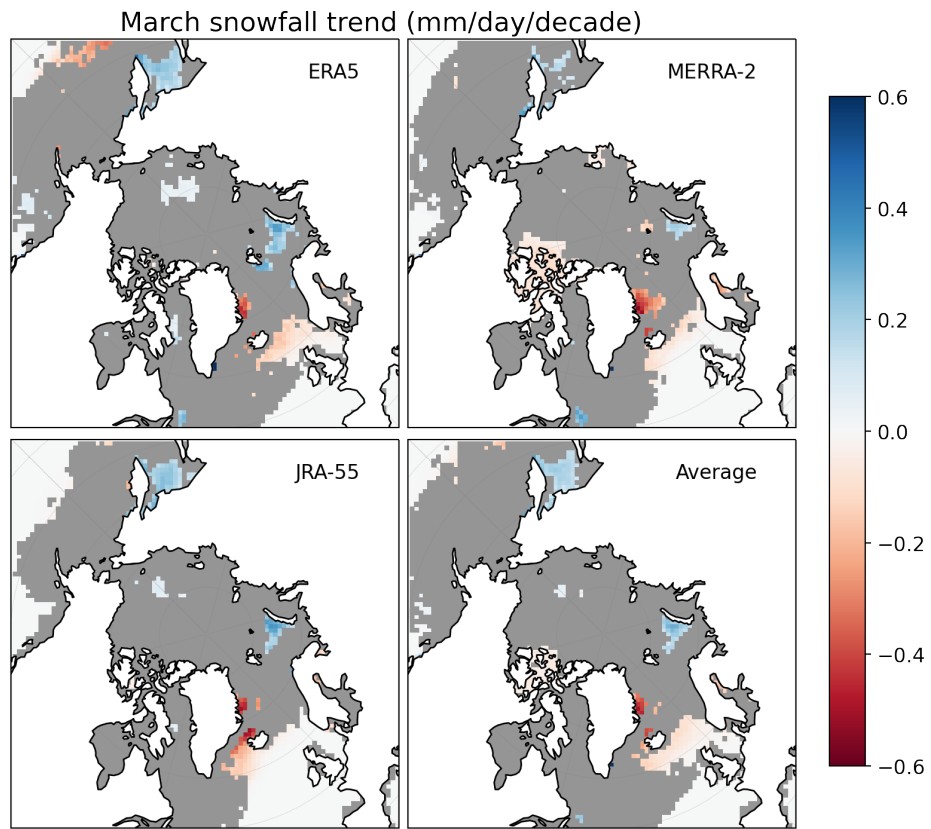

**Figure A5.** Snowfall trend maps over ocean for March 1980-2021 for ERA5, MERRA-2, JRA-55 and the average of the three reanalysis products (Average). Snowfall is provided for each respective reanalysis product regridded to the NESOSIM domain. Regions with no trends (not significant to a 95% confidence interval) are shaded in grey. Snowfall trends over land-covered regions are not shown.

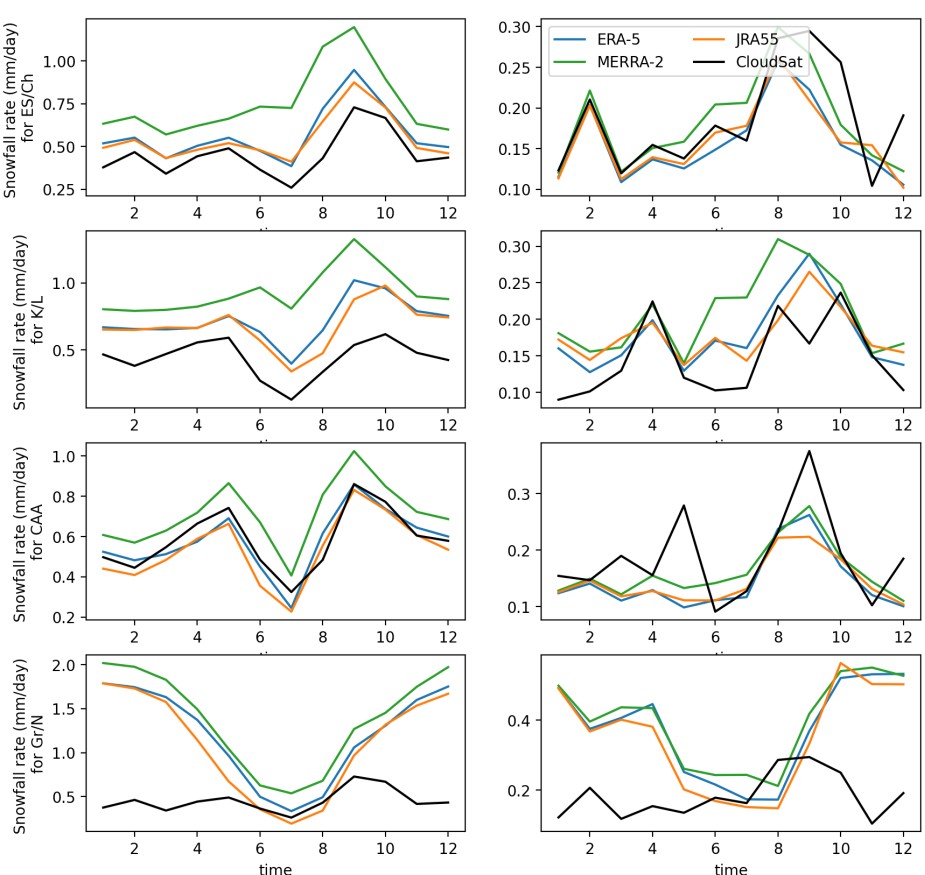

**Figure A6.** Monthly climatologies (left column) and interannual variability (IAV, right column) of snowfall over sea ice in the 60-82°N latitude band within the four quadrants, which are illustrated on the map in Fig. A1. CloudSat fails to adequately represent the monthly climatology in the Greenland/Norwegian Sea region.

## Appendix B: MCMC calibration

The MCMC calibration is carried out in this work following the approach in Cabaj et al. (2023), and is described as follows:

1. Begin with a model run with prior parameter values $a_0$ and observed values $y$, and calculate the log-likelihood $\log(p(y|a_0))$. The prior parameter values are associated with prior parameter distributions $p(a_0)$ for which the mean is $a_0$ and the uncertainty is a prescribed prior parameter uncertainty value. These prior values are given in Table 2. In the following iterative loop, set $a_{current} = a_0$.

2. For each subsequent step in the Markov chain:

   (a) Choose new parameters $a_{test}$ a small step from $a_{current}$, with the step chosen from $p(a_0)$; a normal distribution centered at a0 whose standard deviation is determined by the prior parameter uncertainty.

   (b) Calculate the new test log-likelihood function $\log(p(y|a_{test}))$.

   (c) Calculate the log-likelihood difference $R = \log(p(y|a_{test})) - \log(p(y|a_{current}))$. If $R > \log(U(0,1))$ (where $U(0,1)$ is chosen from a uniform distribution between 0 and 1), then the new parameters are accepted, and $a_{current} := a_{test}$.

As in Cabaj et al. (2023), the observations used for the calibration of NESOSIM are snow depth measurements from the median of airborne Operation IceBridge (OIB) measurements (Petty et al., 2020) and from CRREL-Dartmouth buoys (Perovich et al., 2019), and historical snow density measurements from Soviet drifting stations (Radionov et al., 1997; Mallett et al., 2022). OIB measurements are available exclusively in March and April, and represent the majority of the observations used for calibrating the parameters. Basin-averaged monthly climatologies are used for the drifting station and buoy measurements, and OIB measurements are aggregated to daily averages over the NESOSIM model grid. This aggregation helps to mitigate the impact of observational biases due to the relatively sparse and infrequent observations in these datasets.

Log-likelihood is used to reduce the number of exponential operations calculated, and thus reduce computational costs. The log-likelihood function used in this study is the same as that used in Cabaj et al. (2023), shown below:

$$L = -\frac{1}{2}\sum_{i=1}^{M}\frac{(h_{N,i}-h_{o,i})^2}{u_{h_o}^2} - \frac{1}{2}\sum_{j=1}^{8}\frac{(\langle\rho_{N,j}\rangle-\langle\rho_{d,j}\rangle)^2}{\langle u_{\rho_{d,j}}\rangle^2} - \frac{1}{2}\sum_{k=1}^{8}\frac{(\langle h_{N,k}\rangle-\langle h_{b,k}\rangle)^2}{\langle u_{h_{b,k}}\rangle^2}. \tag{B1}$$

Here, $M$ denotes the number of grid points with Operation IceBridge snow depth measurements, $h_{N,i}$ denotes NESOSIM snow depth output values for a given grid point, and $h_{o,i}$ denotes corresponding OIB snow depth measurements aggregated to a single grid point for a single day. $u_{h_o}$ denotes OIB observational uncertainty. $\rho$ denotes snow density, with subscripts $N$ for NESOSIM and $d$ for drifting stations, respectively, and with $u_{\rho_{d,j}}$ denoting the corresponding uncertainty. $h_b$ denotes CRREL-Dartmouth buoy depth measurements, with corresponding uncertainties $u_{h_{b,k}}$. Angle brackets denote basin-averaged monthly climatologies, and the indices $j$ and $k$ denote months from September to April. The observational uncertainties also account for estimated errors of representativeness in each term.

The acceptance step in the MCMC algorithm allows for the avoidance of local maxima, and posterior parameter distributions are obtained from the distributions of accepted parameters. As in Cabaj et al. (2023), all distributions (the prior parameter distribution, the likelihood function, and the posterior distribution) are assumed to be Gaussian. The modes of the posterior distributions provide optimal values for the parameters. Parameter uncertainty can be estimated from the spread of the posterior distributions. This parameter uncertainty may be propagated through the model to provide estimates of model uncertainty

due to parameter uncertainty. Additional parameters may also be calibrated using the MCMC process, but in previous work, limitations were found due to observations not providing sufficiently strong constraints for the optimization to provide suitable optimal parameter values (Cabaj et al., 2023).

To enable NESOSIM to be run with MCMC parameter calibration, the model was modified to keep model output in memory, minimizing the number of file I/O operations and providing a 20% speedup for MCMC model runs (Cabaj et al., 2023). The

NESOSIM-MCMC code was also adapted to enable the calibration to be run with different reanalysis snowfall input products for this study. This highlights the versatility of NESOSIM as a model well-suited to observational calibration.

*Code and data availability.* The NASA Eulerian Snow On Sea Ice Model (Petty et al., 2018) is available at https://zenodo.org/records/6342069, modified for MCMC at https://doi.org/10.5281/zenodo.7644948. The final calibrated multi-product average with uncertainties, including the individually-calibrated model runs and the CloudSat scaling factors, is provided at https://doi.org/10.5281/zenodo.13307800 (Cabaj et al., 2024). SnowModel-LG (Liston et al., 2020; Stroeve et al., 2020; Liston et al., 2021) output was obtained from the NSIDC at https://nsidc.org/data/nsidc-0758/versions/1. Forcing data for NESOSIM (including atmospheric input from ECMWF ERA5 (Hersbach et al., 2020, 2023), NOAA/NSIDC sea ice concentration (Meier et al., 2021), EUMETSAT OSI SAF sea ice drift (Lavergne et al., 2010; EUMETSAT Ocean and Sea Ice Satellite Application Facility, 2017), and NSIDCv4 Polar Pathfinder sea ice drift (Tschudi et al., 2019)) and processed Operation IceBridge data (Petty et al., 2023b) is available on Zenodo at https://zenodo.org/record/7051062. Additional forcing data for NESOSIM was regridded from MERRA-2 (Gelaro et al., 2017; Global Modeling and Assimilation Office (GMAO), 2015), obtained from NASA GES DISC at https://doi.org/10.5067/7MCPBJ41Y0K6, and JRA-55 (Kobayashi et al., 2015; Japan Meteorological Agency, Japan, 2013), obtained from the NCAR RDA at https://doi.org/10.5065/D6HH6H41. The CloudSat 2C-SNOW-PROFILE product Wood et al. (2013, 2014); Wood and L'Ecuyer (2018) was obtained from https://www.cloudsat.cira.colostate.edu/data-products/2c-snow-profile. In addition to the processed Operation IceBridge data mentioned above, NESOSIM MCMC calibration also makes use of processed snow input from Soviet drifting stations (Mallett et al., 2022; Radionov et al., 1997), and CRREL-Dartmouth buoy observations (Perovich et al., 2022). AWI IceBird data (Jutila et al., 2022; Jutila et al., 2024a, b) was obtained from https://doi.org/10.1594/PANGAEA.932790 and https://doi.org/10.1594/PANGAEA.966057. MOSAiC MagnaProbe snow depth data (Itkin et al., 2021) was obtained from https://doi.org/10.1594/PANGAEA.937781, and MOSAiC snow density cutter data (Macfarlane et al., 2021, 2022) was obtained from https://doi.org/10.1594/PANGAEA.935934. We thank all those who contributed to MOSAiC (Nixdorf et al., 2021).

*Author contributions.* AC designed this study, with input from PJK and AAP. AC ran the model simulations based on model code originally developed by AAP (with subsequent contributions from AC). AC conducted the analysis in the article, with input from all co-authors. AC prepared the manuscript with contributions from all co-authors.

*Competing interests.* The authors declare that they have no conflict of interest.

*Acknowledgements.* AC and PJK conducted this study with support from the Canadian Space Agency Earth System Science: Data Analyses Fund, Grant 16SAUSSNOW. AAP gratefully acknowledges support from NASA under grant number 80NSSC23K1253, awarded by the Cryospheric Sciences program (solicitation: NNH22ZDA001N-ICESAT2). The authors would also like to thank Dr. Walt Meier for his helpful feedback on a precursor to this manuscript.

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
