# Peer review of "Investigating the impact of reanalysis snow input on an observationally calibrated snow-on-sea-ice reconstruction"

_EGUsphere, 2024_

## Referee Comment (RC1)

Review of manuscript

"Investigating the impact of reanalysis snow input on an observationally calibrated snow-on-sea-ice reconstruction" by  Cabaj et al.

The paper addresses the critical issue of relying on reanalysis data in undersampled regions like the Arctic, where limited observations—especially during harsh winters—lead to significant uncertainties in key surface variables such as snowfall and snow depth.  This is problematic since reanalysis data are often used to benchmark models, drive satellite-derived algorithms, and reconstruct products like snow-on-sea-ice estimates. Studies show significant variability in Arctic snowfall across reanalyses due to observational gaps and differing model physics, which can lead to misleading evaluations and derived trends.

The authors focus on understanding discrepancies in snowfall estimates from three major reanalysis products—ERA5, JRA-55, and MERRA-2—when used as inputs for the NASA Eulerian Snow On Sea Ice Model (NESOSIM). They employ Markov chain Monte Carlo (MCMC) techniques to calibrate NESOSIM's snow depth and density parameters, addressing complexities not extensively covered in previous studies. While this work highlights  the complexity of modeling snow on Arctic sea ice and the challenges associated with choosing and calibrating reanalysis products.It would benefit from a stronger emphasis on the need for improved observational data and robust calibration methodologies to enhance the reliability of model outputs.

The manuscript is well-structured and generally clear, but defining technical terms earlier would improve readability.

I thus recommend accepting this submission after revision that considers a few major and several minor comments.

Major comments:

1. Line 23-28. The introduction highlights the challenges posed by biases in reanalysis products. I would think the motivation was quite clear. However, a more focused explanation of how previous research improves on model calibration efforts, such as those by SnowModel-LG, would provide clearer context.

2. Table 1. The issue of coarse resolution does not seem to be explicitly addressed in the discussion. I think one limitation of this study is the coarse spatial resolution used for both the NASA Eulerian Snow On Sea Ice Model (NESOSIM) and the reanalysis data (ERA5, MERRA-2, JRA-55). NESOSIM operates on a 100 km × 100 km grid, and the reanalysis data ranges from 0.25° to 1.25° grids, which may obscure important sub-grid scale processes.

The authors should provide more specific examples of the sub-grid scale processes impacted by the coarse spatial resolution, such as snow redistribution due to ice ridge formation, wind-blown snow dynamics, or small-scale leads that can significantly alter local snow accumulation and density. These sub-grid scale processes could significantly influence

snow depth, density, and heat fluxes, particularly in heterogeneous regions like the marginal seas or areas with dynamic ice cover. The coarse resolution also limits the model's ability to capture fine spatial variability, which may result in oversimplifications when calibrating model parameters, especially for regions with rapid snow accumulation or melt.

Including references to recent high-resolution modeling studies that have explored these processes would help contextualize this limitation. Future work could benefit from incorporating downscaling techniques or nested models to better resolve local variability. High-resolution observational data from satellites or in-situ measurements could also improve model validation and enhance regional accuracy. Addressing these limitations would help refine the model's ability to capture critical snow-ice-atmosphere interactions at finer scales, improving both regional forecasts and large-scale trend assessments.

The use of ERA5 wind input for all NESOSIM runs can introduce biases. One concern is inconsistencies in sea ice representation between datasets—ERA5 uses SST/SIC from HadISST2/OSI SAF, while NESOSIM uses NSIDC SIC data for all runs. These discrepancies can alter surface roughness and wind stress, affecting snow redistribution and compaction. This mismatch can lead to errors in snow depth estimates, particularly in regions with dynamic ice conditions. Would you comment on the potential effect of using ERA5A wind inputs combined with different reanalysis data? Since wind patterns influence snow redistribution and compaction, the choice of a single wind product may not fully capture the sensitivity of snow depth to wind dynamics. Considering product-specific wind inputs or conducting sensitivity tests with different wind datasets could strengthen the reliability of the model results. Furthermore, in Line 84, ERA5 uses a threshold of SIC>20% to distinguish between open ocean and sea ice cover, whereas MERRA-2 uses a 50% threshold (Line 93). Could this difference in SIC thresholds introduce artifacts, particularly in regions with marginal sea ice cover, potentially influencing the weakest or strongest trends in snow depth as discussed in Lines 450-451?

Additionally, the choice of initializing the model in September each year may overlook key early-season snow accumulation events, especially in regions where snow can start accumulating in late summer. Considering the importance of accurately capturing the initial snow state, it would be beneficial to assess the impact of starting the model earlier in the season or adjusting the initialization timing based on regional climatologies。

3. The results are presented clearly but raise questions about the handling of snow density, where the model calibration reconciles snow depths well but leaves density poorly constrained. The authors highlight that this might be due to a lack of density observations, but a more detailed exploration of potential biases introduced by the model's simplicity could be useful.

Given the current limitations of the MCMC calibration for snow density, the authors could consider exploring multi-level Bayesian models that integrate different observational datasets (e.g., buoy measurements, satellite data). This approach could provide more reliable density estimates by better accounting for observational gaps and biases.

In addition to wind packing and blowing snow, other snow-atmosphere interactions, such as sublimation, melt, and refreeze processes, are also simplified in NESOSIM. These processes are crucial for understanding seasonal changes in snow density and depth, and

their absence may contribute to biases. Future work could explore parameterizations for these interactions to improve density and depth estimates across diverse environmental conditions.

4. The model's representation of snow density relies heavily on parameterizations of processes like wind packing and blowing snow, which may not fully capture the complex physical processes occurring at different scales or in diverse environmental conditions. This can lead to uncertainties in the snow density estimates, which might not be fully representative of actual conditions across the Arctic. By integrating additional independent datasets, such as in-situ observations from buoys, satellite-derived snow density estimates, or regional field campaigns, the model calibration can be further refined, reducing the risk of biases and providing a more robust validation of the snow density outputs.

It would also be useful to discuss how the observed variability in snow depth trends between reanalysis products could alter climate sensitivity estimates. For example, different snow depth trends could affect sea ice model sensitivity to atmospheric drivers such as warming temperatures or shifting storm patterns. A brief discussion on this topic would highlight the broader implications of inter-product variability.

Since snow depth and density are crucial for calculating sea ice thickness from satellite altimetry, the variability observed between reanalysis products could have substantial impacts on the interpretation of sea ice thickness trends. Expanding the discussion on the implications for sea ice thickness estimates would emphasize the broader significance of these findings and strengthen the rationale for using multi-product approaches.

Minor comments:

1. From the analyses, both wind packing and blowing snow appear to be critical processes in this study. Therefore, their relevance, physical processes, and mechanisms affecting snow depth and density should be clearly explained in the introduction. Additionally, how these processes are represented and utilized within the NASA Eulerian Snow On Sea Ice Model (NESOSIM) should be outlined. Considering the broader audience of this study, beyond just NESOSIM users, a brief explanation in the introduction or methods section would significantly enhance clarity and understanding.

2.Figures 8 and 10 are visually dense, and the small font size makes them challenging to interpret. Including a statistical summary for each panel (e.g., mean and standard deviation) either alongside the figures or in a supplementary table would greatly enhance clarity. In particular, the Figure 8 caption lacks sufficient detail: the overlapping colored lines and shaded areas are difficult to distinguish and not clearly defined in the caption. It should be explicitly stated what each represents, and the method used to quantify the interannual variability indicated by the shaded area should also be clarified.

Another concern is the interpretation of overlapping uncertainty/internal variability envelopes in the figures (Figs.2,4,6,7,8,9,10). When the envelopes for different reanalysis products overlap, it can be challenging to visually assess whether the observed differences are statistically significant.

3. Table 2 could benefit from a clearer introduction in the main text discussing the importance of the acceptance rates and coefficients of variation in MCMC results, especially for readers less familiar with Bayesian technique

4. Introduction to MCMC: The explanation of the MCMC process in the methods section is detailed but slightly dense. Consider simplifying the language in the initial description to cater to a broader audience or moving more technical details to a supplementary section.

Adding a brief explanation of why MCMC was chosen over other calibration methods would further support the use of Bayesian techniques and enhance the methodological discussion.

5. Trends in units of 'per decade.': This adjustment will help avoid the need for four decimal places in Figs.9-12 and improve readability.

6. CloudSat Discussion: The section discussing the use of CloudSat data might benefit from clarification on the limitations of this dataset, particularly the reduced reliability for latitudes north of 82°N. This could be stated earlier in the methods subsection 2.2.

7. In the results section discussing snow density, consider adding a few more sentences to highlight how the differences in reanalysis products might specifically affect the observed snow depths. Additionally, expanding on the implications of snow depth differences for sea ice thickness estimates would add clarity for practical application

8. When referring to previous studies that employed ERA5 or other reanalysis products, try to explicitly state how the multi-product approach improves over previous single-product studies. This would strengthen the rationale for using multiple reanalysis inputs rather than relying solely on a single product.

---

## Referee Comment (RC2)

Review of Cabaj et al. "Investigating the impact of reanalysis snow input on an observationally calibrated snow-on-sea-ice reconstruction"

This study aims to investigate the impact of different reanalysis snow input in snow-on-sea-ice reconstructions (snow depth and density) provided by NASA's Eulerian Snow On Sea Ice Model (NESOSIM) (Petty et al., 2018). In an earlier study by Cabaj et al. (2023) wind packing and blowing snow parameters were calibrated in NESOSIM using a Markov chain Monte Carlo (MCMC) approach, with ERA5 snowfall. The same MCMC approach is used here, with snowfall inputs from additional reanalysis (MERRA-2 and JRA-55). All reanalysis snowfall was first calibrated to CloudSat following Cabaj et al. (2020). The impact of the MCMC calibration to NESOSIM and the snow depth and density uncertainties are assessed and discussed. NESOSIM outputs of snow depth and density are regionally compared to SnowModel-LG (Liston et al., 2020), a Langragian snow evolution modeling system for sea ice applications. Pan-Arctic and regional monthly trends of snow depth, density and volume derived from both NESOSIM and SnowModel-LG are examined and discussed.

The paper addresses scientific questions within the scope of TC. The title reflects the contents and the abstract provides a concise summary of the study. The overall presentation of the paper is well-structured and the language is clear and coherent. However, while the central findings and conclusions are supported by the data, there are areas where the manuscript would benefit from additional context and clarification. My recommendation is to reconsider the paper after major revisions, to improve its overall quality and precision.

Major comments:

1. NESOSIM and MCMC calibration
Attention should be payed in this section to ensure a coherent description of the methodology used for calibrating the parameters, so that the method can be repeated by anyone to reproduce the results. Some examples are given:
- Line 145: Rewrite into "MCMC is an algorithm applied to Bayesian problems where, given prior information of the parameters..."
- Line 147: "in this case, a log-likelihood function". Function of what?
- Line 149: Replace "prior" with "initial"? In Bayesian problems "prior" refers to a distribution, when in your case you provide a single value. Same in the caption of Figure 3.
- Line 152: "with step size chosen from the distribution". Need to be more specific.
- Equation 1: You can add the uncertainty related to errors of representativeness of the observations, in each term of the equation.
- Line 174: "all distributions are assumed to be Gaussian". Specify which distributions.
- Figure 3. Discuss why we see correlation between the properties, especially in JRA-55.

2. The scaling issue and the sub-grid variability of the snow properties is not discussed enough in the paper. More attention should be paid in this, especially as point measurements of density are used in the calibration.

3. Why do you use ERA5 wind in all runs? Consider using wind inputs from different reanalysis data to investigate their effect in the calibration.

4. You compare post calibration results from NESOSIM to SnowModel-LG but not to independent measurements like passive microwave products or airborne campaigns (OIB/IceBird). Adding comparison to independent measurements will strengthen the study. Regarding the comparison of NESOSIM to SnowModel-LG you should consider the effect of Eulerian (NESOSIM) vs. Langragian (SnowModel-LG) approach when discussing the differences between the model results. Is SnowModel-LG also forced with CloudSat scaled reanalysis forcing? If not, this is another aspect that needs to be emphasized and discussed.

5. Why does the analysis stop in 2019? Consider extending to 2022, so MOSAiC observations can be included in the MCMC calibration.

Minor comments:

1. Need to specify the blowing snow parameter better. I assume it refers to a snow loss term to the atmosphere (i.e., sublimation) and the open ocean. Make clear that snow is not blown from one 100 km x 100 km grid to another.

2. Figures 4 and 5 include only one season. Consider an inter-annual average monthly evolution plot for all properties and their uncertainties.

---

## Author Comment (AC1)

Response to Reviewer #1

Note that the reviewer comments are italicized and our responses are in blue. Where we make changes to existing quoted passages from the text, additions are underlined and deletions are struck through.

*Review of manuscript*

*"Investigating the impact of reanalysis snow input on an observationally calibrated snow-on-sea-ice reconstruction" by Cabaj et al.*

*The paper addresses the critical issue of relying on reanalysis data in undersampled regions like the Arctic, where limited observations—especially during harsh winters—lead to significant uncertainties in key surface variables such as snowfall and snow depth. This is problematic since reanalysis data are often used to benchmark models, drive satellite-derived algorithms, and reconstruct products like snow-on-sea-ice estimates. Studies show significant variability in Arctic snowfall across reanalyses due to observational gaps and differing model physics, which can lead to misleading evaluations and derived trends.*

*The authors focus on understanding discrepancies in snowfall estimates from three major reanalysis products— ERA5, JRA-55, and MERRA-2—when used as inputs for the NASA Eulerian Snow On Sea Ice Model (NESOSIM). They employ Markov chain Monte Carlo (MCMC) techniques to calibrate NESOSIM's snow depth and density parameters, addressing complexities not extensively covered in previous studies. While this work highlights the complexity of modeling snow on Arctic sea ice and the challenges associated with choosing and calibrating reanalysis products. It would benefit from a stronger emphasis on the need for improved observational data and robust calibration methodologies to enhance the reliability of model outputs.*

*The manuscript is well-structured and generally clear, but defining technical terms earlier would improve readability.*

*I thus recommend accepting this submission after revision that considers a few major and several minor comments.*

We thank the reviewer for this helpful feedback, and we are including our responses below. Regarding the general point on the emphasis for the need of observation data, one major change we are planning to include to address this is the inclusion of a discussion on a comparison of MOSAiC snow depth and density observations to NESOSIM and SnowModel-LG, discussed further below.

*Major  comments*

1. *Line 23-28. The introduction highlights the challenges posed by biases in reanalysis products. I would think the motivation was quite clear. However, a more focused explanation of how previous research improves on model calibration efforts, such as those by SnowModel-LG, would provide clearer context.*

Thank you for the suggestion. We do discuss model calibration in later paragraphs, and as such, will add some additional explanation at line 46 as follows:

"The implementation of this calibration approach was motivated by the fact that the free parameters in NESOSIM had been previously manually calibrated by comparing model output to observations (Petty et al., 2018). Between NESOSIM model versions, the ERA-Interim dataset, previously used for input to the model, was superseded by ERA5. Updating this dataset provided motivation for the development of a new automated parameter calibration process. The need for estimates of snow-on-sea-ice uncertainty for sea ice thickness retrieval applications further motivated the choice of the MCMC approach, since the MCMC process provides estimates of parameter uncertainty."

We also propose modifying Line 59 as follows:

"SnowModel-LG likewise includes observation-based calibration, namely an assimilation-based bias correction to precipitation to bring modelled snow depth into agreement with ground-based and remote sensing observations, including Operation IceBridge measurements (Liston et al., 2020, Stroeve et al., 2020)."

References:

Liston, G. E., Itkin, P., Stroeve, J., Tschudi, M., Stewart, J. S., Pedersen, S. H., Reinking, A. K., and Elder, K.: A Lagrangian Snow-Evolution System for Sea-Ice Applications (SnowModel-LG): Part I—Model Description, *Journal of Geophysical Research: Oceans*, 125, e2019JC015913, https://doi.org/10.1029/2019JC015913, 2020.

Petty, A. A., Webster, M., Boisvert, L., and Markus, T.: The NASA Eulerian Snow on Sea Ice Model (NESOSIM) v1.0: initial model development and analysis, *Geoscientific Model Development*, 11, 4577–4602, https://doi.org/10.5194/gmd-11-4577-2018, 2018.

Stroeve, J., Liston, G. E., Buzzard, S., Zhou, L., Mallett, R., Barrett, A., Tschudi, M., Tsamados, M., Itkin, P., and Stewart, J. S.: A Lagrangian Snow Evolution System for Sea Ice Applications (SnowModel-LG): Part II—Analyses, *Journal of Geophysical Research: Oceans*, 125, e2019JC015900, https://doi.org/10.1029/2019JC015900, 2020.

2. *Table 1. The issue of coarse resolution does not seem to be explicitly addressed in the discussion. I think one limitation of this study is the coarse spatial resolution used for both the NASA Eulerian Snow On Sea Ice Model (NESOSIM) and the reanalysis data (ERA5, MERRA-2, JRA-55). NESOSIM operates on a 100 km × 100 km grid, and the reanalysis data ranges from 0.25° to 1.25° grids, which may obscure important sub-grid scale processes.*

   *The authors should provide more specific examples of the sub-grid scale processes impacted by the coarse spatial resolution, such as snow redistribution due to ice ridge formation, wind-blown snow dynamics, or small-scale leads that can significantly alter local snow accumulation and density. These sub-grid scale processes could significantly influence snow depth, density, and heat fluxes, particularly in heterogeneous regions like the marginal seas or areas with dynamic ice cover. The coarse resolution also limits the model's ability to capture fine spatial variability, which may result in oversimplifications when calibrating model parameters, especially for regions with rapid snow accumulation or melt.*

We agree with the reviewer that the coarse spatial resolution is a limitation of this work, and based on the reviewer's comment, we suggest the following addition to the discussion at line 575:

"The relatively coarse resolution of NESOSIM may impact its representativeness, since some snow-on-sea-ice processes operate on very small spatial scales and short timescales. The sea ice advection and divergence processes in NESOSIM represent a spatially-averaged tendency of snow to be redistributed with sea ice motion, but may fail to capture small-scale effects from localized ridging and small-scale leads often seen in observational studies (Itkin et al., 2023; Macfarlane et al., 2023). The amount of blowing-snow loss due to leads has been observed to be influenced by strong winds and warm air temperatures from Arctic cyclone events, which may be challenging to capture in the current configuration of NESOSIM (Clemens-Sewall et al., 2023). The coarse time resolution also limits the model's ability to capture rapid changes in snow depth due to short-term accumulation events. In a broader modelling context, high-resolution modelling may be necessary to adequately capture small-scale processes (Lecomte et al., 2015). NESOSIM could be run at a higher resolution to take advantage of the higher resolution of available drift products to better capture the influence of sea ice motion. However, sub-gridscale parameterization would still be necessary to better capture smaller-scale effects.

References:

Clemens-Sewall, D., Polashenski, C., Frey, M. M., Cox, C. J., Granskog, M. A., Macfarlane, A. R., Fons, S. W., Schmale, J., Hutchings, J. K., von Albedyll, L., Arndt, S., Schneebeli, M., and Perovich, D.: Snow Loss Into Leads in

Arctic Sea Ice: Minimal in Typical Wintertime Conditions, but High During a Warm and Windy Snowfall Event, *Geophysical Research Letters*, 50, e2023GL102816, https://doi.org/10.1029/2023GL102816, 2023.

Itkin, P., Hendricks, S., Webster, M., et al.: Sea ice and snow characteristics from year-long transects at the MOSAiC Central Observatory, *Elementa: Science of the Anthropocene*, 11, 00048, https://doi.org/10.1525/elementa.2022.00048, 2023.

Lecomte, O., Fichefet, T., Flocco, D., Schroeder, D., and Vancoppenolle, M.: Interactions between wind-blown snow redistribution and melt ponds in a coupled ocean–sea ice model, *Ocean Modelling*, 87, 67–80, https://doi.org/10.1016/j.ocemod.2014.12.003, 2015.

Macfarlane, A. R., Schneebeli, M., Dadic, R. et al: A Database of Snow on Sea Ice in the Central Arctic Collected during the MOSAiC expedition, *Sci Data*, 10, 398, https://doi.org/10.1038/s41597-023-02273-1, 2023.

*Including references to recent high-resolution modeling studies that have explored these processes would help contextualize this limitation. Future work could benefit from incorporating downscaling techniques or nested models to better resolve local variability. High-resolution observational data from satellites or in-situ measurements could also improve model validation and enhance regional accuracy. Addressing these limitations would help refine the model's ability to capture critical snow-ice-atmosphere interactions at finer scales, improving both regional forecasts and large-scale trend assessments.*

We agree that higher resolution would be beneficial, and that future work could benefit from incorporating downscaling approaches/nested models. We have included reference to modelling and observation studies describing small-scale processes in our suggested revision above. The relative simplicity of NESOSIM (including its comparatively coarse resolution) was a decision to enable to rapid production of snow-on-sea-ice estimates for operational purposes; particularly the production of sea ice thickness estimates from ICESat-2 altimetry measurements.

Regarding high-resolution observational data, taking into account this comment and comments from other reviewers, we propose to also include some discussion comparing NESOSIM and SnowModel-LG to snow depth and density observations from the MOSAiC campaign which we will briefly summarize below. We find that this comparison highlights some of the challenges the models have with coarse resolution not necessarily representing small-scale processes. Observed snow depth and density tend to be highly variable within a single model grid cell.

Comparison of NESOSIM and SnowModel-LG to MOSAiC observations

Below is a brief summary with key points for a comparison to observations we intend to include in our revised manuscript; more detail and discussion will be added when we prepare the revised manuscript. We compare output from NESOSIM and SnowModel-LG to snow depth and density measurements (Macfarlane et al., 2023) obtained during the 2019-2020 Multidisciplinary drifting Observatory for the Study of Arctic Climate (MOSAiC) campaign (Nicolaus et al., 2022). Snow depths were measured using magnaprobes (Itkin et al., 2021), and in previous studies have been noted to be relatively thin (Itkin et al., 2023). Bulk snow densities used in this comparison were calculated from density cutter measurements, which sample densities at varying depths within a snow pit (Macfarlane et al., 2022). Snow was sampled over a variety of conditions, including ridges and leads, and snow over first-year and multi-year ice (Macfarlane et al., 2023).

To compare with gridded snow model outputs, MOSAiC observations are collocated to the nearest model grid point, and then averaged by day for each grid point. These values are then compared to the corresponding model grid point value, excluding dates during which NESOSIM output is unavailable. Below, we present figures aggregated by month.

[Figure]

[Figure]

Figure A: (left): MOSAiC magnaprobe snow depth (Itkin et al., 2021) compared to NESOSIM snow depth, before ("E5config") and after ("MCMC") individual dataset calibration to observations; triangles indicate individually-calibrated datasets. (right): MOSAiC snow depth compared to SnowModel-LG snow depth with different snow forcings. May-August are excluded due to the absence of NESOSIM data. Error bars represent 1 standard deviation of the monthly mean (with MOSAiC data also including contributions from the daily standard deviation).

Figure A shows monthly-averaged MOSAiC snow depth measurements (Itkin et al., 2021) compared to NESOSIM and SnowModel-LG. Note that aggregated MOSAiC outputs may differ slightly between figure panels because the model output is being aggregated to different model grids. Both models show general good agreement with the observations, with some products showing slight biases. The uncalibrated NESOSIM output driven by JRA55 has a general high bias relative to the other products (and a daily mean bias of 3.2 cm relative to MOSAiC). Differences in seasonal cycles are apparent between the models. Compared to MOSAiC, several NESOSIM products are biased low in October-November 2019, and some products are biased high in March-April. SnowModel-LG (particularly when driven by MERRA-2) is conversely biased slightly high in November and December. Nevertheless, overall agreement is close, with daily root-mean-square difference not exceeding 10 cm for all products relative to MOSAiC.

[Figure]

Figure B: Monthly averages of gridded MOSAiC snow density cutter data vs. monthly averages of coincident model output data. Error bars indicate 1 standard deviation. (left): MOSAiC compared to NESOSIM, before ("E5config") and after (MCMC) the MCMC calibration. (right): MOSAiC compared to SnowModel-LG. Only months with at least 8 measurements are shown.

Figure B shows monthly averages of MOSAiC snow density cutter measurements (Macfarlane et al., 2022) compared to NESOSIM and SnowModel-LG. Prior to gridded collocation with the models, bulk density for each measurement event was calculated from the average of sampled densities weighted by sample snow thickness. NESOSIM snow density from all products shows relatively little variation over the time period, whereas SnowModel-LG snow density shows more seasonality. Both models show a high mean bias relative to observed

values, with SnowModel-LG driven by MERRA-2 having the largest daily mean bias (60 kg/m$^3$). The comparatively high variability of the observed values is also apparent.

Below are tables with daily (gridded) comparison statistics for NESOSIM and SnowModel-LG with respect to MOSAiC, including correlation, root-mean-square difference (RMSD) and mean bias error (MBE), for reference.

Table A: Daily comparison statistics for NESOSIM and SnowModel-LG comparisons to MOSAiC snow depth observations

|  | NESOSIM-ERA5-MCMC | NESOSIM-JRA55-E5config | NESOSIM-JRA55-MCMC | NESOSIM-MERRA2-E5config | NESOSIM-MERRA2-MCMC | SnowModel-LG-ERA5 | SnowModel-LG-MERRA2 |
|---|---|---|---|---|---|---|---|
| Pearson Correlation to MOSAiC | 0.68 | 0.67 | 0.67 | 0.67 | 0.67 | 0.64 | 0.58 |
| RMSD (cm) | 8.5 | 9.5 | 8.6 | 8.7 | 8.9 | 9.3 | 10 |
| MBE (cm) | -2.4 | 3.2 | -1.6 | -0.24 | -2.9 | -0.26 | 1.8 |

Table B: Daily comparison statistics for NESOSIM and SnowModel-LG comparisons to MOSAiC snow density observations

|  | NESOSIM-ERA5-MCMC | NESOSIM-JRA55-E5config | NESOSIM-JRA55-MCMC | NESOSIM-MERRA2-E5config | NESOSIM-MERRA2-MCMC | SnowModel-LG-ERA5 | SnowModel-LG-MERRA2 |
|---|---|---|---|---|---|---|---|
| Pearson Correlation to MOSAiC | 0.22 | 0.20 | 0.20 | 0.15 | 0.15 | 0.24 | 0.16 |
| RMSD (kgm$^{-3}$) | 79 | 80 | 80 | 80 | 79 | 80 | 93 |
| MBE (kgm$^{-3}$) | 32 | 32 | 35 | 32 | 28 | 32 | 60 |

References (abbreviated for this response, full references will be included in the article):

Itkin, P., Hendricks, S., Webster, M., et al.: Sea ice and snow characteristics from year-long transects at the MOSAiC Central Observatory, *Elementa: Science of the Anthropocene*, 11, 00048, https://doi.org/10.1525/elementa.2022.00048, 2023.

Macfarlane, A. R., Schneebeli, M., Dadic, R. et al.: A Database of Snow on Sea Ice in the Central Arctic Collected during the MOSAiC expedition, *Sci Data*, 10, 398, https://doi.org/10.1038/s41597-023-02273-1, 2023.

Nicolaus, M., Perovich, D. K., Spreen, G. et al.: Overview of the MOSAiC expedition: Snow and sea ice, *Elementa: Science of the Anthropocene,* 10, 000046, https://doi.org/10.1525/elementa.2021.000046, 2022.

Dataset references:

Itkin, Polona; Webster, Melinda; Hendricks, Stefan, et al. (2021): Magnaprobe snow and melt pond depth measurements from the 2019-2020 MOSAiC expedition [dataset]. *PANGAEA*, https://doi.org/10.1594/PANGAEA.937781

Macfarlane, Amy R; Schneebeli, Martin; Dadic, Ruzica, et al. (2022): Snowpit snow density cutter profiles measured during the MOSAiC expedition [dataset]. *PANGAEA*, https://doi.org/10.1594/PANGAEA.940214, In: Macfarlane, AR et al. (2021): Snowpit raw data collected during the MOSAiC expedition [dataset bundled publication]. *PANGAEA*, https://doi.org/10.1594/PANGAEA.935934

*The use of ERA5 wind input for all NESOSIM runs can introduce biases. One concern is inconsistencies in sea ice representation between datasets—ERA5 uses SST/SIC from HadISST2/OSI SAF, while NESOSIM uses NSIDC SIC data for all runs. These discrepancies can alter surface roughness and wind stress, affecting snow redistribution and compaction. This mismatch can lead to errors in snow depth estimates, particularly in regions with dynamic ice conditions. Would you comment on the potential effect of using ERA5A wind inputs combined with different reanalysis data? Since wind patterns influence snow redistribution and compaction, the choice of a single wind product may not fully capture the sensitivity of snow depth to wind dynamics. Considering product-specific wind inputs or conducting sensitivity tests with different wind datasets could strengthen the reliability of the model results. Furthermore, in Line 84, ERA5 uses a threshold of SIC>20% to distinguish between open ocean and sea ice cover, whereas MERRA-2 uses a 50% threshold (Line 93). Could this difference in SIC thresholds introduce artifacts, particularly in regions with marginal sea ice cover, potentially influencing the weakest or strongest trends in snow depth as discussed in Lines 450-451?*

Regarding the different SIC representation in the reanalysis products, we agree that these could potentially introduce some artefacts if the precipitation processes in reanalyses are impacted via e.g. surface moisture fluxes, although we would consider this as one of the contributing factors to inter-product differences in reanalysis snowfall rates, among other factors. We briefly discussed this at line 608. We note that NESOSIM does not account for surface roughness; snow distribution within model grid cells is considered to be uniform. For running NESOSIM, we do not use different SIC products for different snow products. We will add the following to the discussion at line 575 to address this:

"The representation of sea ice differs between reanalysis products, and may not be coincident with the observational sea ice concentration used as input to NESOSIM in this work. This, in conjunction with regridding, may introduce some artefacts in regions of marginal sea ice cover such as the Greenland Sea region."

We kept the wind inputs the same to isolate the contribution of snow input to NESOSIM model output. We note that ERA5 wind performs relatively well compared to other reanalysis products in Arctic observational studies over sea ice (cf. Graham et al. 2019a, 2019b). The focus on snow was motivated by the fact that snowfall is the primary input to the NESOSIM snow budget, and when we were updating the model reanalysis input, we found that changing snowfall had a stronger impact on the snow output. However, we will acknowledge the impact of wind as follows in the discussion after line 575:

"The ERA5 wind product was used in all configurations in this study to isolate the contribution of snowfall to NESOSIM, since snowfall is the primary input to the NESOSIM budget. In observational comparisons in the Arctic, ERA5 has been found to perform relatively well compared to other reanalysis products, including JRA-55 and MERRA-2, which motivates the choice of ERA5 over other products (Graham et al., 2019a, 2019b). However, the choice of reanalysis wind input may also have an impact on NESOSIM output. The wind packing and blowing snow processes take effect only when wind speed exceeds the 5 m/s wind action threshold. If wind speeds from different input products are on differing sides of the threshold, wind-related snow processes may take effect at a given location and time for one product and not another. The strength of the blowing snow process is also dependent on wind speed. Future work could investigate the impact of differing wind input products to NESOSIM."

References:

Graham, R. M., Cohen, L., Ritzhaupt, N., Segger, B., Graversen, R. G., Rinke, A., Walden, V. P., Granskog, M. A., and Hudson, S. R.: Evaluation of Six Atmospheric Reanalyses over Arctic Sea Ice from Winter to Early Summer, *Journal of Climate*, 32, 4121–4143, https://doi.org/10.1175/JCLI-D-18-0643.1, 2019a.

Graham, R. M., Hudson, S. R., and Maturilli, M.: Improved Performance of ERA5 in Arctic Gateway Relative to Four Global Atmospheric Reanalyses, *Geophysical Research Letters*, 46, 6138–6147, https://doi.org/10.1029/2019GL082781, 2019b.

*Additionally, the choice of initializing the model in September each year may overlook key early-season snow accumulation events, especially in regions where snow can start accumulating in late summer. Considering the importance of accurately capturing the initial snow state, it would be beneficial to assess the impact of starting the model earlier in the season or adjusting the initialization timing based on regional climatologies.*

The choice was made to initialize NESOSIM in September based on several factors. Firstly, August is a complex month for snow-on-sea-ice representation, since melt may be occurring, which may not be represented in NESOSIM. Instead of attempting to represent August snow evolution, NESOSIM is initialized with initial snow depth values which attempt to capture climatological snow-on-sea-ice conditions. Furthermore, September was chosen for initialization since it is the month during which Arctic sea ice attains its minimum extent, so that NESOSIM can model snow-on-sea-ice from the start of seasonal sea ice growth to the melt season. The motivation for this choice is discussed in Petty et al., 2023. We also note that we have previously investigated calibrating initial conditions, (Cabaj et al., 2023) although our previous investigation found that the initial condition values were underconstrained.

References:

Cabaj, A., Kushner, P. J., and Petty, A. A.: Automated Calibration of a Snow-On-Sea-Ice Model, *Earth and Space Science*, 10, e2022EA002655, https://doi.org/10.1029/2022EA002655, 2023.

Petty, A. A., Keeney, N., Cabaj, A., Kushner, P., and Bagnardi, M.: Winter Arctic sea ice thickness from ICESat-2: upgrades to freeboard and snow loading estimates and an assessment of the first three winters of data collection, *The Cryosphere*, 17, 127–156, https://doi.org/10.5194/tc-17-127-2023, 2023.

3. *The results are presented clearly but raise questions about the handling of snow density, where the model calibration reconciles snow depths well but leaves density poorly constrained. The authors highlight that this might be due to a lack of density observations, but a more detailed exploration of potential biases introduced by the model's simplicity could be useful.*

We agree that representation of snow density is an ongoing challenge. We have discussed part of what we suspect is a contributing factor to biases in the model; namely, the imposed maximum snow density value, at line 560. Further to this, our proposed inclusion of comparisons to in situ MOSAiC snow density cutter observations (included above) can provide some insight to this point. We note that this comparison has associated caveats due to representational differences between comparatively coarse-resolution models and point observations. Nevertheless, both SnowModel-LG and NESOSIM have comparable difficulty reproducing snow density from these observations. However, NESOSIM snow density has low seasonal variability relative to both SnowModel-LG and MOSAiC. Since snow in NESOSIM cannot be removed from the lower layer (for a given grid cell, it can only decrease as a consequence of sea ice motion), end-of-season densities are expected to approach 350 kg/m$^3$ as an increasing proportion of the snow in each grid cell is old (lower-layer) snow. We will further expand our discussion to include these points.

*Given the current limitations of the MCMC calibration for snow density, the authors could consider exploring multi-level Bayesian models that integrate different observational datasets (e.g., buoy measurements, satellite data). This approach could provide more reliable density estimates by better accounting for observational gaps and biases.*

Thank you for the suggestion. We agree that such an exploration could be of interest in general, but we do not necessarily think this would be a beneficial modification to our current study, given the simplicity of the NESOSIM model parameter space and the limited ability of the NESOSIM model to represent a wide range of snow densities, since the model is limited to two layers and one densification process. However, this could be an option for

calibrating a possible future version of NESOSIM with a more complex representation of snow density or additional snow processes, and thus would be an approach we would consider for a future study.

*In addition to wind packing and blowing snow, other snow-atmosphere interactions, such as sublimation, melt, and refreeze processes, are also simplified in NESOSIM. These processes are crucial for understanding seasonal changes in snow density and depth, and their absence may contribute to biases. Future work could explore parameterizations for these interactions to improve density and depth estimates across diverse environmental conditions.*

We agree, and hope that this study can be a motivation for additional work to address these points. Guided by the reviewer's comment, we will expand the discussion at line 571 to mention this: "Several processes, including snow redistribution by wind, sublimation, and melt and refreeze processes are simplified in NESOSIM." and at line 575: "Future work could explore parameterizations of additional processes, such as sublimation and snow redistribution by wind to improve snow depth and density estimates across a variety of environmental conditions."

4. *The model's representation of snow density relies heavily on parameterizations of processes like wind packing and blowing snow, which may not fully capture the complex physical processes occurring at different scales or in diverse environmental conditions. This can lead to uncertainties in the snow density estimates, which might not be fully representative of actual conditions across the Arctic. By integrating additional independent datasets, such as in-situ observations from buoys, satellite-derived snow density estimates, or regional field campaigns, the model calibration can be further refined, reducing the risk of biases and providing a more robust validation of the snow density outputs.*

We agree, however, given that NESOSIM in its current state has a very simplified representation of snow density, including additional measurements at this time may not yield major improvements. We would like to note that the calibration already incorporates in situ observations from buoys (as described in e.g. Line 159). Also, the challenge with incorporating observations from field campaigns is that such measurements tend to be highly localized, which may lead to "overfitting" if a large number of measurements are located in a small geographical region, or if only a single season is sampled. The primary dataset used for the calibration is OIB airborne measurements, which were repeated over several seasons and cover a relatively wide spatial range. The buoys likewise cover a wide spatial range, and historical density measurements are used to also provide relatively widespread density estimates which span several decades, to provide an estimate of the snow density seasonal cycle. Localized density measurements may be difficult to incorporate since it is likely that values of snow density larger than what NESOSIM in its current configuration can represent may be present.

Based on several reviewer suggestions, we propose including a comparison to MOSAiC in situ measurements as discussed above, which highlights the difficulty in comparing comparatively coarse-resolution snow-on-sea-ice models such as NESOSIM and SnowModel-LG against localized measurements.

*It would also be useful to discuss how the observed variability in snow depth trends between reanalysis products could alter climate sensitivity estimates. For example, different snow depth trends could affect sea ice model sensitivity to atmospheric drivers such as warming temperatures or shifting storm patterns. A brief discussion on this topic would highlight the broader implications of inter-product variability.*

Thank you for the suggestion, we propose expanding the discussion at line 608 to include the following: "The differing snow depth trends between model outputs due to different reanalysis snowfall inputs may have impacts on climate sensitivity estimates due to its influences on sea ice. Coupled climate model simulations have found contrasting climate impacts of snow on Arctic sea ice due to competing influences on congelation sea ice growth and surface melt (Holland et al., 2021), but snow-free summers may increase sea ice melt (Webster et al., 2021). Thus, by influencing sea ice thickness, a declining snow depth trend could influence trends in atmosphere-ice heat fluxes, which in turn could influence sea ice extent and other climate variables."

References:

Holland, M. M., Clemens-Sewall, D., Landrum, L., Light, B., Perovich, D., Polashenski, C., Smith, M., and Webster, M.: The influence of snow on sea ice as assessed from simulations of CESM2, *The Cryosphere*, 15, 4981–4998, https://doi.org/10.5194/tc-15-4981-2021, 2021.

Webster, M. A., DuVivier, A. K., Holland, M. M., and Bailey, D. A.: Snow on Arctic Sea Ice in a Warming Climate as Simulated in CESM, *Journal of Geophysical Research: Oceans*, 126, e2020JC016308, https://doi.org/10.1029/2020JC016308, 2021.

*Since snow depth and density are crucial for calculating sea ice thickness from satellite altimetry, the variability observed between reanalysis products could have substantial impacts on the interpretation of sea ice thickness trends. Expanding the discussion on the implications for sea ice thickness estimates would emphasize the broader significance of these findings and strengthen the rationale for using multi-product approaches.*

We appreciate the reviewer's acknowledgement of the importance of this context, and agree that this context is good to keep in mind, since NESOSIM was primarily developed for deriving ice thickness from sea ice altimetry. We will clarify in the NESOSIM model description that the model was developed for this purpose, and we propose also adding the following to the discussion at line 604:

"Inter-product differences in snow depth and density may have substantial impacts on estimates of sea ice thickness from sea ice altimetry measurements. For example, given representative values of lidar freeboard, and representative densities of snow, ice, and water, if snow depth estimates with a 5 cm difference are used to estimate sea ice thickness, the difference in derived sea ice thickness can be as large as 30 cm (Giles et al, 2007). Thus, if trends differ between snow products, trends in derived sea ice thickness will be impacted as well. For sea ice freeboard, a snow product with a decreasing trend would impose an increasing derived ice thickness trend on top of any trend in the freeboard itself. Interannual variability in snow was found to strongly influence sea ice volume derived from CryoSat-2 altimetry measurements (Bunzel et al., 2018). Hence, differing snow depth trends (or lack thereof) between products could lead to differing conclusions on trends in derived sea ice thickness."

References:

Bunzel, F., Notz, D., and Pedersen, L. T.: Retrievals of Arctic Sea-Ice Volume and Its Trend Significantly Affected by Interannual Snow Variability, *Geophysical Research Letters*, 45, 11,751-11,759, https://doi.org/10.1029/2018GL078867, 2018.

Giles, K. A., Laxon, S. W., Wingham, D. J., Wallis, D. W., Krabill, W. B., Leuschen, C. J., McAdoo, D., Manizade, S. S., and Raney, R. K.: Combined airborne laser and radar altimeter measurements over the Fram Strait in May 2002, *Remote Sensing of Environment*, 111, 182–194, https://doi.org/10.1016/j.rse.2007.02.037, 2007.

*Minor comments:*

*1. From the analyses, both wind packing and blowing snow appear to be critical processes in this study. Therefore, their relevance, physical processes, and mechanisms affecting snow depth and density should be clearly explained in the introduction. Additionally, how these processes are represented and utilized within the NASA Eulerian Snow On Sea Ice Model (NESOSIM) should be outlined. Considering the broader audience of this study, beyond just NESOSIM users, a brief explanation in the introduction or methods section would significantly enhance clarity and understanding.*

We attempted to explain these processes in Section 2.3 starting at line 134, and propose changing it to more clearly explain as follows:

"Wind packing controls the amount of snow transferred between layers,  decreasing the snow depth and increasing the bulk snow density as snow is transferred from the upper (less dense) layer to the lower (denser) layer. The blowing snow process acts only on the upper snow layer, and decreases the snow depth in the upper

layer linearly with wind speed. The blowing snow term includes an atmosphere loss and an open-water loss term, which are prescribed separately in NESOSIM v1.1 (Petty et al., 2023). The open-water loss term accounts for sea ice concentration, with regions of lower sea ice concentration experiencing more open water loss. For the purpose of [...]"

*2.Figures 8 and 10 are visually dense, and the small font size makes them challenging to interpret. Including a statistical summary for each panel (e.g., mean and standard deviation) either alongside the figures or in a supplementary table would greatly enhance clarity. In particular, the Figure 8 caption lacks sufficient detail: the overlapping colored lines and shaded areas are difficult to distinguish and not clearly defined in the caption. It should be explicitly stated what each represents, and the method used to quantify the interannual variability indicated by the shaded area should also be clarified.*

Thank you for the feedback. We will adjust the font size in the subsequent revision of the article, and will also include as a supplement a summary table of the numerical values of climatological means and standard deviations (or trend values with confidence intervals, as applicable) for each quantity and each month, since we agree this would be helpful information to have for clarity.

As suggested by another reviewer, we have adjusted Figure 10 and other trend figures to exclude non-significant values, and to represent the values with bars, as follows. Note that this figure is not finalized and we will make further adjustments with legend placement and font size prior to the next revision, but we include it for visual reference. Our proposed changes to Fig. 8 and its caption are shown further below.

[Figure]

*Another concern is the interpretation of overlapping uncertainty/internal variability envelopes in the figures (Figs.2,4,6,7,8,9,10). When the envelopes for different reanalysis products overlap, it can be challenging to visually assess whether the observed differences are statistically significant.*

Thank you for the feedback. As mentioned above, by suggestion from another reviewer, we have now left out values that are not statistically significant in the trend plots, and for the climatology plots, we have also opted to show error bars instead of shading as shown below, with Fig. 8 serving as a representative figure for the proposed changes. The modified caption for Figure 8 is also shown below. As previously stated, note that some changes to e.g. text size and legend placement are still outstanding, but will be finalized for the next revision of the article.

[Figure]

Regional snow and sea ice climatologies from NESOSIM and SnowModel-LG

Figure 8: Climatologies of regionally-averaged snow depth, density, and volume from MCMC-calibrated NESOSIM output and SnowModel-LG output, for 1980-2019. Regional CDR sea ice area climatologies also shown. "Average" indicates the inter-product average for the three NESOSIM configurations. Climatologies from SnowModel-LG driven with ERA5 and MERRA-2 are also shown, with dashed lines. Regions are as described in Fig. A1. Bars indicate interannual variability of each respective climatology, which is quantified by the standard deviation of the climatology.

*3. Table 2 could benefit from a clearer introduction in the main text discussing the importance of the acceptance rates and coefficients of variation in MCMC results, especially for readers less familiar with Bayesian technique*

Thank you for the suggestion. We described acceptance rate further later in the article but agree that it would benefit from an earlier introduction. As such, we propose modifying at line 273 as follows:

"The acceptance rate, calculated from the ratio of accepted parameters to the total number of iterations, indicates the efficiency of the MCMC process, with an optimal efficiency for a 2-parameter MCMC process being approximately 23% (Gelman et al., 2013). Coefficients of variation are calculated from the standard deviation of the posterior distribution divided by the posterior parameter value, and quantify the relative spread of the posterior distribution. This provides a quantitative indication of how well-constrained the parameters are by the MCMC calibration. The posterior distribution of ERA5 [...]"

Reference:

Gelman, A., Carlin, J. B., Stern, H. S., Dunson, D. B., Vehtari, A., and Rubin, D. B.: Bayesian Data Analysis, *CRC Texts in Statistical Science*, CRC Press, Boca Raton, third edn., 2013.

*4. Introduction to MCMC: The explanation of the MCMC process in the methods section is detailed but slightly dense. Consider simplifying the language in the initial description to cater to a broader audience or moving more technical details to a supplementary section.*

*Adding a brief explanation of why MCMC was chosen over other calibration methods would further support the use of Bayesian techniques and enhance the methodological discussion.*

Following the reviewer's suggestion, we have decided to move the detailed explanation of the MCMC process to a supplemental section. For an explanation of why MCMC was chosen of other methods, we propose adding the following at line 147:

"Using an MCMC approach for calibrating NESOSIM allows for the automated estimation of free parameters, which were previously manually estimated via comparison to observations (Petty et al., 2018). An added benefit of this approach is that it yields posterior distributions of the parameters, which provide an estimate of parameter uncertainty."

Per another reviewer's suggestions, note that we will make the following changes to the explanation of the MCMC process, for added clarity:

- Line 145 rewritten as "MCMC is an algorithm applied to Bayesian problems where, given prior information of the parameters…" as suggested
- Line 147: replacing with "a log-likelihood function of the difference between model output and selected aggregated observations used for the calibration, weighted by the uncertainty in the observations." to clarify
- Added at Line 149: "The prior parameter values are associated with prior parameter distributions $p(a_0)$ for which the mean is $a_0$ and the uncertainty is a prescribed prior parameter uncertainty value. These prior values are given in Table 1." We will also add the prescribed prior parameter uncertainties ($1 \times 10^{-8}$ $s^{-1}$ for wind packing and $1 \times 10^{-8}$ $m^{-1}$ for blowing snow) to Table 1 for reference.
- Line 152: rephrase to: "with the subsequent step chosen from $p(a_0)$; a normal distribution centered at $a_0$ whose standard deviation is determined by the prior parameter uncertainty"
- Equation 1: we will add a clarification in the description that observation uncertainties also account for estimated errors of representativeness in each term.
- Line 174: rephrasing to "all distributions (the prior parameter distribution, the likelihood function, and the posterior distribution) are assumed to be Gaussian"

*5. Trends in units of 'per decade.': This adjustment will help avoid the need for four decimal places in Figs.9-12 and improve readability.*

Thank you for the helpful suggestion. We will adjust the units accordingly in the subsequent revision of the manuscript.

*6. CloudSat Discussion: The section discussing the use of CloudSat data might benefit from clarification on the limitations of this dataset, particularly the reduced reliability for latitudes north of 82°N. This could be stated earlier in the methods subsection 2.2.*

We agree that these are necessary details to mention, and propose adding the following:

Line 105: "CloudSat's ground track had latitudinal coverage between 82°N and 82°S, and hence, no measurements are available near the pole. To mitigate ground clutter contamination of near-surface returns, near-surface snowfall rate measurements are retrieved from the 3rd vertical bin above ocean surfaces, or the 5th vertical bin above sea ice (as determined by a climatological sea ice mask) (Wood & L'Ecuyer, 2018). Data quality flags are applied to exclude potentially contaminated observations as described in Cabaj et al., 2020."

References:

Cabaj, A., Kushner, P. J., Fletcher, C. G., Howell, S., and Petty, A. A.: Constraining Reanalysis Snowfall Over the Arctic Ocean Using CloudSat Observations, *Geophysical Research Letters*, 47, e2019GL086426, https://doi.org/10.1029/2019GL086426, 2020.

Wood, N. B., & L'Ecuyer, T. S.. Level 2C Snow Profile Process Description and Interface Control Document, Product Version P1R05. NASA JPL CloudSat project (Document revision 0), 2018. Retrieved from http://www.cloudsat.cira.colostate.edu/sites/default/files/products/files/2C-SNOW-PROFILE_PDICD.P1_R05.rev0_.pdf

*7. In the results section discussing snow density, consider adding a few more sentences to highlight how the differences in reanalysis products might specifically affect the observed snow depths. Additionally, expanding on the implications of snow depth differences for sea ice thickness estimates would add clarity for practical application*

Thank you for the suggestion. Regarding the snow density section, assuming that the reviewer meant to ask for us to highlight how differing reanalysis products might affect the snow densities, we note that as we discuss, we do not expect snowfall to have as much of a contribution to snow density. However, we can state the following at Line 358:

"The slight differences between the uncalibrated snow density outputs may result from the influence of snowfall. For example, depending on the timing of snowfall, reanalysis snowfall may impact snow density in different ways; a high snow accumulation event will reduce the overall bulk density in the short term, but if this accumulation occurs early in the season, more snow may subsequently be transferred to the lower layer, increasing the bulk snow density in the long term if subsequent accumulation is lower. For example, NESOSIM driven by JRA55 shows deep snow in the early season relative to other products, which may contribute to its high later-season snow density bias as seen Fig. 7a."

Regarding reanalysis impacts on snow depths, we can further expand as follows at Line 337, since we did intend to include this in our manuscript but may not have been explicit enough in our discussion:

"Some of the relative biases between the products persist; JRA-55 continues to have a relatively large early-season snow depth which is not seen in the other products, consistent with its early-season snowfall bias. Conversely, at the end of the season, JRA-55 and MERRA-2, which previously both exceeded ERA5 at the end of the season, consistent with snowfall biases over sea ice in most regions, particularly over the central Arctic. now Following the MCMC calibration, JRA-55 and MERRA-2 bracket ERA5 snow depth it on either side, with the multi-product average closely matching the ERA5 values. "

The impact on sea ice thickness is connected to our response earlier to Major Comment 4 as discussed above; we will expand to include a discussion of impacts on sea ice thickness from differences in snow depth.

*8. When referring to previous studies that employed ERA5 or other reanalysis products, try to explicitly state how the multi-product approach improves over previous single-product studies. This would strengthen the rationale for using multiple reanalysis inputs rather than relying solely on a single product.*

Thank you for the suggestion, we have previously stated this briefly in our discussion at Line 610 but will also restate this earlier in the motivation at Line 57, and propose expanding further:

"Multi-dataset approaches help to reveal biases between datasets, and facilitate the characterization of dataset uncertainties."

We will expand similarly where other previous single-product studies are mentioned, as suggested by the reviewer.

---

## Author Comment (AC2)

Response to Reviewer #2

Note that the reviewer comments are italicized and our responses are in blue. Where we make changes to existing quoted passages from the text, additions are underlined and deletions are struck through.

*This study aims to investigate the impact of different reanalysis snow input in snowon- sea-ice reconstructions (snow depth and density) provided by NASA's Eulerian Snow On Sea Ice Model (NESOSIM) (Petty et al., 2018). In an earlier study by Cabaj et al. (2023) wind packing and blowing snow parameters were calibrated in NESOSIM using a Markov chain Monte Carlo (MCMC) approach, with ERA5 snowfall. The same MCMC approach is used here, with snowfall inputs from additional reanalysis (MERRA-2 and JRA-55). All reanalysis snowfall was first calibrated to CloudSat following Cabaj et al. (2020). The impact of the MCMC calibration to NESOSIM and the snow depth and density uncertainties are assessed and discussed. NESOSIM outputs of snow depth and density are regionally compared to SnowModel-LG (Liston et al., 2020), a Langragian snow evolution modeling system for sea ice applications. Pan-Arctic and regional monthly trends of snow depth, density and volume derived from both NESOSIM and SnowModel-LG are examined and discussed.*

*The paper addresses scientific questions within the scope of TC. The title reflects the contents and the abstract provides a concise summary of the study. The overall presentation of the paper is well-structured and the language is clear and coherent. However, while the central findings and conclusions are supported by the data, there are areas where the manuscript would benefit from additional context and clarification. My recommendation is to reconsider the paper after major revisions, to improve its overall quality and precision.*

We thank the reviewer for the comments, and appreciate the assistance on providing additional context and clarification. Our responses are below. One major change we are planning to address some of the reviewer comments is to compare MOSAiC snow depth and density observations to NESOSIM and SnowModel-LG, discussed further below.

*Major comments:*

*1. NESOSIM and MCMC calibration*

*Attention should be payed in this section to ensure a coherent description of the methodology used for calibrating the parameters, so that the method can be repeated by anyone to reproduce the results. Some examples are given:*

We appreciate the specific examples, which enable us to enhance the clarity of this work. We will go line by line and describe our implementations of the suggested changes.

*Line 145: Rewrite into "MCMC is an algorithm applied to Bayesian problems where, given prior information of the parameters…"*

Thank you for the suggestion, we will rewrite Line 145 to "MCMC is an algorithm applied to Bayesian problems where, given prior information of the parameters…" as suggested.

*Line 147: "in this case, a log-likelihood function". Function of what?*

Line 147: replacing with "a log-likelihood function of the difference between model output and selected aggregated observations used for the calibration, weighted by the uncertainty in the observations." to clarify.

*Line 149: Replace "prior" with "initial"? In Bayesian problems "prior" refers to a distribution, when in your case you provide a single value. Same in the caption of Figure 3.*

Thank you for the suggestion. We would like to clarify that we do mean this as a Bayesian prior; the initial value is effectively the mean of the prior distribution, for which we also provide a prior standard deviation (prior parameter uncertainty estimate). We will add the following to clarify this point as follows:

"The prior parameter values are associated with prior parameter distributions $p(a_0)$ for which the mean is $a_0$ and the uncertainty is a prescribed prior parameter uncertainty value. These prior values are given in Table 1."

We will also add the prescribed prior parameter uncertainties ($1 \times 10^{-8}$ s$^{-1}$ for wind packing and $1 \times 10^{-8}$ m$^{-1}$ for blowing snow) to Table 1 for reference.

*Line 152: "with step size chosen from the distribution". Need to be more specific.*

Line 152: We suggest rephrasing to better clarify: "with the subsequent step chosen from $p(a_0)$; a normal distribution centered at $a_0$ whose standard deviation is determined by the prior parameter uncertainty"

*Equation 1: You can add the uncertainty related to errors of representativeness of the observations, in each term of the equation.*

Equation 1: we will add a clarification in the description that observation uncertainties also account for estimated errors of representativeness in each term.

*Line 174: "all distributions are assumed to be Gaussian". Specify which distributions.*

Line 174: rephrasing to "all distributions (the prior parameter distribution, the likelihood function, and the posterior distribution) are assumed to be Gaussian"

*Figure 3: Discuss why we see correlation between the properties, especially in JRA-55.*

We note that this was also discussed in previous work, but we propose discussing this as follows: "The correlation between the wind packing and blowing snow parameters may be a consequence of the processes compensating for each other, as described in Cabaj et al., 2023. The wind packing process transfers snow to the lower layer, where the blowing snow process cannot remove snow, so if wind packing is strengthened, the blowing snow process may also be strengthened to compensate and enable additional snow depth reduction."

Reference:

Cabaj, A., Kushner, P. J., and Petty, A. A.: Automated Calibration of a Snow-On-Sea-Ice Model, *Earth and Space Science*, 10, e2022EA002655, https://doi.org/10.1029/2022EA002655, 2023.

*2. The scaling issue and the sub-grid variability of the snow properties is not discussed enough in the paper. More attention should be paid in this, especially as point measurements of density are used in the calibration.*

We did consider these points when conducting the analysis but we agree that it would be beneficial to mention them in the paper. Following this comment and a comment from another reviewer, we suggest the following addition to the discussion at line 575:

"The relatively coarse resolution of NESOSIM may impact its representativeness, since some snow-on-sea-ice processes operate on very small spatial scales and short timescales. The sea ice advection and divergence processes in NESOSIM represent a  spatially-averaged tendency of snow to be redistributed with sea ice motion, but it does not capture small-scale effects from localized ridging and small-scale leads often seen in observational studies (Macfarlane et al., 2023). The amount of blowing-snow loss due to leads has been observed to be influenced by strong winds and warm air temperatures from Arctic cyclone events, which may be challenging to capture in the current configuration of NESOSIM (Clemens-Sewall et al., 2023). In a broader modelling context, high-resolution modelling may be necessary to adequately capture small-scale processes (Lecomte et al., 2015). NESOSIM could be run at a higher resolution to take advantage of the higher resolution of available drift products to better capture the influence of sea ice motion. However, sub-gridscale parameterization would still be necessary to better capture smaller-scale effects."

References:

Clemens-Sewall, D., Polashenski, C., Frey, M. M., Cox, C. J., Granskog, M. A., Macfarlane, A. R., Fons, S. W., Schmale, J., Hutchings, J. K., von Albedyll, L., Arndt, S., Schneebeli, M., and Perovich, D.: Snow Loss Into Leads in Arctic Sea Ice: Minimal in Typical Wintertime Conditions, but High During a Warm and Windy Snowfall Event, *Geophysical Research Letters*, 50, e2023GL102816, https://doi.org/10.1029/2023GL102816, 2023.

Lecomte, O., Fichefet, T., Flocco, D., Schroeder, D., and Vancoppenolle, M.: Interactions between wind-blown snow redistribution and melt ponds in a coupled ocean–sea ice model, *Ocean Modelling*, 87, 67–80, https://doi.org/10.1016/j.ocemod.2014.12.003, 2015.

Macfarlane, A. R., Schneebeli, M., Dadic, R. et al: A Database of Snow on Sea Ice in the Central Arctic Collected during the MOSAiC expedition, *Sci Data*, 10, 398, https://doi.org/10.1038/s41597-023-02273-1, 2023.

Regarding the density measurements used in the calibration, we would like to clarify that although the source of the density values is point measurements, density observations are aggregated to a monthly climatology to address representativeness. This is stated in Line 162: "Basin-averaged monthly climatologies are used for the drifting station and buoy measurements, and OIB measurements are aggregated to daily averages over the NESOSIM model grid." That said, we will add the following to further clarify: "This aggregation helps to mitigate the impact of observational biases due to the relatively sparse and infrequent observations in these datasets."

*Why do you use ERA5 wind in all runs? Consider using wind inputs from different reanalysis data to investigate their effect in the calibration.*

We chose to use ERA5 wind in all runs because we wanted to isolate the contribution of snow input to NESOSIM model output. The focus on snow was motivated by the fact that snowfall is the primary input to the NESOSIM snow budget, and when we were updating the model reanalysis input, we found that changing snowfall had a stronger impact on the snow output. We note that ERA5 has been found to perform well relative to other reanalysis wind products in studies over Arctic sea ice (Graham et al., 2019a, 2019b), which motivates the choice of ERA5. We will acknowledge the impact of wind as follows in the discussion after line 575:

"The ERA5 wind product was used in all configurations in this study to isolate the contribution of snowfall to NESOSIM, since snowfall is the primary input to the NESOSIM budget. In observational comparisons in the Arctic, ERA5 has been found to perform relatively well compared to other reanalysis products, including JRA-55 and MERRA-2, which motivates the choice of ERA5 over other products (Graham et al., 2019a, 2019b). However, the choice of reanalysis wind input nevertheless may have an impact on NESOSIM output. The wind packing and blowing snow processes take effect only when wind speed exceeds the 5 m/s wind action threshold. If wind speeds from different input products are on differing sides of the threshold, wind-related snow processes may take effect at a given location and time for one product and not another. The strength of the blowing snow process is also dependent on wind speed. Future work could investigate the impact of differing wind input products to NESOSIM."

References:

Graham, R. M., Cohen, L., Ritzhaupt, N., Segger, B., Graversen, R. G., Rinke, A., Walden, V. P., Granskog, M. A., and Hudson, S. R.: Evaluation of Six Atmospheric Reanalyses over Arctic Sea Ice from Winter to Early Summer, *Journal of Climate*, 32, 4121–4143, https://doi.org/10.1175/JCLI-D-18-0643.1, 2019a.

Graham, R. M., Hudson, S. R., and Maturilli, M.: Improved Performance of ERA5 in Arctic Gateway Relative to Four Global Atmospheric Reanalyses, *Geophysical Research Letters*, 46, 6138–6147, https://doi.org/10.1029/2019GL082781, 2019b.

*You compare post calibration results from NESOSIM to SnowModel-LG but not to independent measurements like passive microwave products or airborne campaigns (OIB/IceBird). Adding comparison to independent measurements will strengthen the study. Regarding the comparison of NESOSIM to SnowModel-LG you should consider the effect of Eulerian (NESOSIM) vs. Langragian (SnowModel-LG) approach when discussing the*

*differences between the model results. Is SnowModel-LG also forced with CloudSat scaled reanalysis forcing? If not, this is another aspect that needs to be emphasized and discussed.*

We agree that comparison to independent measurements would be beneficial to the study. Motivated by suggestions below (and from other reviewers), we have decided to compare to snow depth and density measurements from MOSAiC. We have also conducted some preliminary comparisons with IceBird, but due to the coarseness of the model grids and the proximity of IceBird observations to coastal areas, the coincident coverage between IceBird observations and model output is comparatively limited. We will present some results from a preliminary comparison of MOSAiC output with NESOSIM and SnowModel-LG below after we address the other points in this paragraph.

We agree that making note of the differences between the Eulerian and Lagrangian approaches would be helpful to discuss the differences between model results. We will include the following at line 560:

"inter-product differences. NESOSIM and SnowModel-LG also have different approaches for addressing sea ice drift. The Lagrangian approach of SnowModel-LG allows it to track individual ice parcels and thus can more robustly account for the contribution of sea ice dynamics to the snow budget. NESOSIM presents a more simplified approach, but is less computationally expensive to run. Limitations in NESOSIM's [...]"

SnowModel-LG is not forced with CloudSat-scaled reanalysis forcing; whereas we run NESOSIM ourselves, the SnowModel-LG output we use is provided by the NSIDC. We will clarify in our manuscript that SnowModel-LG does not use CloudSat-scaled inputs, although it does include other observation-based adjustments of its input data. We propose a modification to Line 59 as follows:

"SnowModel-LG likewise includes observation-based calibration, namely an assimilation-based bias correction to precipitation to bring modelled snow depth into agreement with ground-based and remote sensing observations, including Operation IceBridge measurements (Liston et al., 2020, Stroeve et al., 2020)."

References:

Liston, G. E., Itkin, P., Stroeve, J., Tschudi, M., Stewart, J. S., Pedersen, S. H., Reinking, A. K., and Elder, K.: A Lagrangian Snow-Evolution System for Sea-Ice Applications (SnowModel-LG): Part I—Model Description, *Journal of Geophysical Research: Oceans*, 125, e2019JC015913, https://doi.org/10.1029/2019JC015913, 2020.

Stroeve, J., Liston, G. E., Buzzard, S., Zhou, L., Mallett, R., Barrett, A., Tschudi, M., Tsamados, M., Itkin, P., and Stewart, J. S.: A Lagrangian Snow Evolution System for Sea Ice Applications (SnowModel-LG): Part II—Analyses, *Journal of Geophysical Research: Oceans*, 125, e2019JC015900, https://doi.org/10.1029/2019JC015900, 2020.

Returning back to the topic of comparison to independent measurements, below we present a brief summary of a comparison to MOSAiC observations, which we intend to incorporate into the subsequent revision of our manuscript, below.

Comparison of NESOSIM and SnowModel-LG to MOSAiC observations

Below is a brief summary with key points for a comparison to observations we intend to include in our revised manuscript; more detail and discussion will be added when we prepare the revised manuscript. We compare output from NESOSIM and SnowModel-LG to snow depth and density measurements (Macfarlane et al., 2023) obtained during the 2019-2020 Multidisciplinary drifting Observatory for the Study of Arctic Climate (MOSAiC) campaign (Nicolaus et al., 2022). Snow depths were measured using magnaprobes (Itkin et al., 2021), and in previous studies have been noted to be relatively thin (Itkin et al., 2023). Bulk snow densities used in this comparison were calculated from density cutter measurements, which sample densities at varying depths within a snow pit (Macfarlane et al., 2022). Snow was sampled over a variety of conditions, including ridges and leads, and snow over first-year and multi-year ice (Macfarlane et al., 2023).

To compare with gridded snow model outputs, MOSAiC observations are collocated to the nearest model grid point, and then averaged by day for each grid point. These values are then compared to the corresponding model grid point value, excluding dates during which NESOSIM output is unavailable. Below, we present figures aggregated by month.

[Figure]

Figure A: (left): MOSAiC magnaprobe snow depth (Itkin et al., 2021) compared to NESOSIM snow depth, before ("E5config") and after ("MCMC") individual dataset calibration to observations; triangles indicate individually-calibrated datasets. (right): MOSAiC snow depth compared to SnowModel-LG snow depth with different snow forcings. May-August are excluded due to the absence of NESOSIM data. Error bars represent 1 standard deviation of the monthly mean (with MOSAiC data also including contributions from the daily standard deviation).

Figure A shows monthly-averaged MOSAiC snow depth measurements (Itkin et al., 2021) compared to NESOSIM and SnowModel-LG. Note that aggregated MOSAiC outputs may differ slightly between figure panels because the model output is being aggregated to different model grids. Both models show general good agreement with the observations, with some products showing slight biases. The uncalibrated NESOSIM output driven by JRA55 has a general high bias relative to the other products (and a daily mean bias of 3.2 cm relative to MOSAiC). Differences in seasonal cycles are apparent between the models. Compared to MOSAiC, several NESOSIM products are biased low in October-November 2019, and some products are biased high in March-April. SnowModel-LG (particularly when driven by MERRA-2) is conversely biased slightly high in November and December. Nevertheless, overall agreement is close, with daily root-mean-square difference not exceeding 10 cm for all products relative to MOSAiC.

[Figure]

Figure B: Monthly averages of gridded MOSAiC snow density cutter data vs. monthly averages of coincident model output data. Error bars indicate 1 standard deviation. (left): MOSAiC compared to NESOSIM, before ("E5config") and after (MCMC) the MCMC calibration. (right): MOSAiC compared to SnowModel-LG. Only months with at least 8 measurements are shown.

Figure B shows monthly averages of MOSAiC snow density cutter measurements (Macfarlane et al., 2022) compared to NESOSIM and SnowModel-LG. Prior to gridded collocation with the models, bulk density for each measurement event was calculated from the average of sampled densities weighted by sample snow thickness. NESOSIM snow density from all products shows relatively little variation over the time period, whereas SnowModel-LG snow density shows more seasonality. Both models show a high mean bias relative to observed values, with SnowModel-LG driven by MERRA-2 having the largest daily mean bias (60 kg/m$^3$). The comparatively high variability of the observed values is also apparent.

Below are tables with daily (gridded) comparison statistics for NESOSIM and SnowModel-LG with respect to MOSAiC, including correlation, root-mean-square difference (RMSD) and mean bias error (MBE), for reference.

Table A: Daily comparison statistics for NESOSIM and SnowModel-LG comparisons to MOSAiC snow depth observations

|  | NESOSIM-ERA5-MCMC | NESOSIM-JRA55-E5config | NESOSIM-JRA55-MCMC | NESOSIM-MERRA2-E5config | NESOSIM-MERRA2-MCMC | SnowModel-LG-ERA5 | SnowModel-LG-MERRA2 |
|---|---|---|---|---|---|---|---|
| Pearson Correlation to MOSAiC | 0.68 | 0.67 | 0.67 | 0.67 | 0.67 | 0.64 | 0.58 |
| RMSD (cm) | 8.5 | 9.5 | 8.6 | 8.7 | 8.9 | 9.3 | 10 |
| MBE (cm) | -2.4 | 3.2 | -1.6 | -0.24 | -2.9 | -0.26 | 1.8 |

Table B: Daily comparison statistics for NESOSIM and SnowModel-LG comparisons to MOSAiC snow density observations

|  | NESOSIM-ERA5-MCMC | NESOSIM-JRA55-E5config | NESOSIM-JRA55-MCMC | NESOSIM-MERRA2-E5config | NESOSIM-MERRA2-MCMC | SnowModel-LG-ERA5 | SnowModel-LG-MERRA2 |
|---|---|---|---|---|---|---|---|
| Pearson Correlation to MOSAiC | 0.22 | 0.20 | 0.20 | 0.15 | 0.15 | 0.24 | 0.16 |
| RMSD (kgm$^{-3}$) | 79 | 80 | 80 | 80 | 79 | 80 | 93 |
| MBE (kgm$^{-3}$) | 32 | 32 | 35 | 32 | 28 | 32 | 60 |

References (abbreviated for this response, full references will be included in the article):

Itkin, P., Hendricks, S., Webster, M., et al.: Sea ice and snow characteristics from year-long transects at the MOSAiC Central Observatory, Elementa: Science of the Anthropocene, 11, 00048, https://doi.org/10.1525/elementa.2022.00048, 2023.

Macfarlane, A. R., Schneebeli, M., Dadic, R. et al.: A Database of Snow on Sea Ice in the Central Arctic Collected during the MOSAiC expedition, Sci Data, 10, 398, https://doi.org/10.1038/s41597-023-02273-1, 2023.

Nicolaus, M., Perovich, D. K., Spreen, G. et al.: Overview of the MOSAiC expedition: Snow and sea ice, Elementa: Science of the Anthropocene, 10, 000046, https://doi.org/10.1525/elementa.2021.000046, 2022.

Dataset references:

Itkin, Polona; Webster, Melinda; Hendricks, Stefan, et al. (2021): Magnaprobe snow and melt pond depth measurements from the 2019-2020 MOSAiC expedition [dataset]. PANGAEA, https://doi.org/10.1594/PANGAEA.937781

Macfarlane, Amy R; Schneebeli, Martin; Dadic, Ruzica, et al. (2022): Snowpit snow density cutter profiles measured during the MOSAiC expedition [dataset]. PANGAEA, https://doi.org/10.1594/PANGAEA.940214, In: Macfarlane, AR et al. (2021): Snowpit raw data collected during the MOSAiC expedition [dataset bundled publication]. PANGAEA, https://doi.org/10.1594/PANGAEA.935934

*Why does the analysis stop in 2019? Consider extending to 2022, so MOSAiC observations can be included in the MCMC calibration.*

We initially intended to include 4 decades in this work; hence, we performed the analysis from 1980-2019. Following the reviewer suggestion, we have produced NESOSIM output up to April 2023, the most recent period for which all required NESOSIM input forcings are available. We will update the dataset associated with this article to include the updated NESOSIM output. For the purposes of the SnowModel-LG comparisons, however, we need to limit the analysis to end in April 2021 for consistency (since SnowModel-LG output is not available after July 2021). *To the editor: We will adjust all NESOSIM and SnowModel-LG output/comparison figures to include these additional years in the subsequent manuscript revision and revise our manuscript accordingly*.

A challenge when selecting datasets to use for the MCMC calibration is to avoid overfitting to observational datasets. The observations we chose for the MCMC calibration were chosen because they were available over many years, and because they covered a wide spatial extent. For example, Operation IceBridge, although limited to the spring season, covers several different regions with measurements spanning multiple NESOSIM grid cells for each measurement day, and several seasons of data are available. Since MOSAiC measurements are available for only a single season, and are generally localized to a single grid point per day, incorporating them into the MCMC calibration may result in overfitting.

However, as discussed above, we are now proposing including a brief comparison of NESOSIM and SnowModel-LG snow depth and snow density to MOSAiC into our article, to provide a set of independent measurements to assess the impact of the MCMC calibration, and to compare the representation of snow in NESOSIM and SnowModel-LG. This comparison reveals some of the challenges with comparing gridded products with point measurements, since there is high variability in the measured snow depths and densities which is not represented by the models.

*Minor comments:*

*1. Need to specify the blowing snow parameter better. I assume it refers to a snow loss term to the atmosphere (i.e., sublimation) and the open ocean. Make clear that snow is not blown from one 100 km x 100 km grid to another.*

Yes, the blowing snow term refers to loss to the atmosphere and open ocean. We will add two sentences at Line 139 to clarify this. "The blowing snow term is exclusively a loss term and does not include redistribution. When snow is lost from a grid cell via this process, it is removed, not redistributed to another grid cell."

Following suggestions from other reviewers, we also propose expanding the text at Line 134 as follows for additional clarity:

"Wind packing controls the amount of snow transferred between layers,  decreasing the snow depth and increasing the bulk snow density as snow is transferred from the upper (less dense) layer to the lower (denser) layer. The blowing snow process acts only on the upper snow layer, and decreases the snow depth in the upper layer linearly with wind speed. The blowing snow term includes an atmosphere loss and an open-water loss term, which are prescribed separately in NESOSIM v1.1 (Petty et al., 2023). The open-water loss term accounts for sea ice concentration, with regions of lower sea ice concentration experiencing more open water loss. For the purpose of [...]"

*2. Figures 4 and 5 include only one season. Consider an inter-annual average monthly evolution plot for all properties and their uncertainties.*

We wanted to show a representative single season so that the uncertainties due specifically to the different products would be apparent, since in the inter-annual average, the monthly uncertainties are less than the standard deviation of the monthly climatology. *To the editor: In the revision, we propose to include some representative inter-annual average monthly plots of this in a supplement.*

---

## Author Comment (AC3)

Response to Reviewer #3

Note that the reviewer comments are italicized and our responses are in blue. Where we make changes to existing quoted passages from the text, additions are underlined and deletions are struck through.

*Review: Cabaj et al, Investigating the impact of reanalysis snow input on an observationally calibrated snow-on-sea-ice reconstruction*

*General*

*The paper investigates the impact of snowfall rates from three reanalysis products on two calibrated parameter values in the NASA Eulerian Snow on Sea Ice Model (NESOSIM). NESOSIM is a two-layer snow budget model used to calculate depth and density of snow on sea ice for polar (mostly Arctic) oceans. NESOSIM is forced by snowfall, wind speed, sea ice concentration, and ice motion. It accounts for horizontal transport of snow cover by ice drift, densification and metamorphosis of the snow pack, and loss of snow cover to leads through wind erosion and transport. The current paper focuses on calibrating paramaters that control these last two processes, and on correcting reanalysis snowfall. It builds on two earlier papers: one paper that presented a snowfall bias correction procedure using CloudSat as the target snowfall; and a second paper that presented a Markov Chain Monte Carlo automated calibration method.*

*It is not clear to me what new insights have been gained by the current study or if any new method or solution is presented. The automated calibration procedure using Monte Carlo Markov Chain is described in Cabaj et al (2023) and the snowfall bias correction is presented in Cabaj et al (2020). Although the current study evaluates using multiple reanalysis products, I do not think this evaluation in it's current form adds much that is new.*

We thank the reviewer for the feedback. This study builds on previous work in important ways by quantifying how uncertainties from distinctive sources of forcing could be strongly linked to parameter uncertainty in snow on sea ice modeling. The particular application, here, combining NESOSIM and MCMC optimization, enables automated parameter calibration that can account for differences in forcing (which we have chosen to limit to snowfall in this case to keep the paper manageable). We are not aware in the current literature of this kind of systematic model development approach, including uncertainty quantification, at the level provided here. Without the relatively low cost of NESOSIM, such an approach would have been impossible (see next reply). This study provides a strong example for how to calibrate models and account for forcing uncertainty for an important and uncertain climate metric (snow on Arctic sea ice). Because of reviewer interest, and our interest, in the critical measurements from MOSAiC, we will now also include a limited comparison with field measurements from the campaign, to further place our results and the uncertainty analysis in context of recent observations independent of the calibration process.

*NESOSIM is a relatively simple conceptual model that uses simple parameterizations of blowing snow losses and densification. At a fundamental level the model can be seen as a transfer function that corrects for biases in snowfall products. The paper demonstrates that parameter values for blowing snow and densification are sensitive to the snowfall product. It also demonstrates that the two parameters chosen for calibration are highly interdependent. The first point is well known. There are numerous examples of model sensitivity to input in the statistical and Earth science literature, including in the terrestrial snow modelling and snow hydrology literature. The second point should not be surprising because the blowing snow parameterization removes snow (reducing the bias in snowfall input) and the densification parameterization (the rate at which low density new snow is "transformed" into high density old snow) increases/retains snow (increasing the bias in snowfall input). I suspect a "brute-force" evaluation on the parameter space would have shown this interdependence. The question for the modeller is how to constrain this interdependence.*

We respectfully disagree with the suggestion that *NESOSIM is a relatively simple conceptual model*. In fact, NESOSIM is regularly applied operationally (providing snow depth and density estimates on Arctic sea ice for estimation of sea ice thickness from ICESat-2 altimetry (Petty et al., 2023)), and has been shown to perform well

quantitatively in previous publications, provided suitable sampling and interpretation are applied (Zhou et al., 2021). Even with its simplicity, NESOSIM is not dissimilar from other commonly-used reanalysis-based approaches used for satellite sea ice thickness estimation, and it improves on simpler approaches such as the widely-used fractional scaling of the Warren et al. (1999) climatology (as used in e.g. Kwok and Cunningham, 2015; though still used as a point of comparison in more recent studies such as Zhou et al., 2021). By contrast, *conceptual models* are used primarily for pedagogical studies or for idealized analytical studies. Our study capitalizes on NESOSIM's low cost and simplicity to carry out a thorough characterization of its uncertainty across the important dimensions of its free-parameter uncertainty and snow-input related forcing uncertainty. In this sense, NESOSIM serves as a benchmark model that can be improved upon for similar automated calibration approaches using more comprehensive models. But we disagree that the model should be dismissed in and of itself as is suggested here.

The motivation for using an MCMC approach was to avoid the very "brute-force" approach the reviewer suggests using. Prior to the use of this approach, we did not know that the parameter space would be smooth, and we are glad that the inter-dependence makes sense to the reviewer, as it does to us. The quantitative estimate of this dependency and its dependence on forcing dataset are new results, even if the sign of the dependence could have been anticipated physically. Using automated parameter calibration positions us well for future changes in forcing datasets and model updates. For example, when NESOSIM was previously updated to use ERA5 input instead of ERA-Interim, the free parameters needed to be adjusted, and the parameters were previously selected by hand based on best estimates to agreement with selected observations. An automated approach was sought out to streamline and better justify this process, and to quantify parameter uncertainty around the optimal estimate (which brute-force approaches are incapable of doing). As ERA5 and other updated reanalysis products come onstream, we will be able to re-calibrate NESOSIM and its updates relatively quickly, guided by the experience outlined in this study. We will add some mention of these points in the discussion.

References: Kwok, R., & Cunningham, G. F. (2015). Variability of Arctic sea ice thickness and volume from CryoSat-2. *Philosophical transactions. Series A, Mathematical, physical, and engineering sciences*, *373*(2045), 20140157. https://doi.org/10.1098/rsta.2014.0157

Petty, A. A., Keeney, N., Cabaj, A., Kushner, P., and Bagnardi, M.: Winter Arctic sea ice thickness from ICESat-2: upgrades to freeboard and snow loading estimates and an assessment of the first three winters of data collection, *The Cryosphere*, 17, 127-156, https://doi.org/10.5194/tc-17-127-2023, 2023.

Warren, S. G., Rigor, I. G., Untersteiner, N., Radionov, V. F., Bryazgin, N. N., Aleksandrov, Y. I., and Colony, R.: Snow Depth on Arctic Sea Ice, *Journal of Climate*, 12, 1814-1829, https://doi.org/10.1175/1520-0442(1999)012<1814:SDOASI>2.0.CO;2, 1999.

Zhou, L., Stroeve, J., Xu, S., Petty, A., Tilling, R., Winstrup, M., Rostosky, P., Lawrence, I. R., Liston, G. E., Ridout, A., Tsamados, M., and Nandan, V.: Inter-Comparison of Snow Depth over Arctic Sea Ice from Reanalysis Reconstructions and Satellite Retrieval, *The Cryosphere*, 15, 345-367, https://doi.org/10.5194/tc-15-345-2021, 2021.

*I also do not think that the approach described here gets at the question of uncertainty. There are (at least) three sources of uncertainty in models: uncertainty related to input fields, parameter uncertainty, and model structural uncertainty. Although the scaling with CloudSat address some of the input bias and uncertainty, the different parameter distributions for the different reanalaysis products suggest that input uncertainty is also included in the assessment of parameter uncertainty. I would suggest that a better understanding of input and parameter uncertainty would be gained by calibrating the model on measured snowfall from MOSAiC or the drifting stations, and then using these parameter estimates in a bias correction step. The model (parameterizations and bias correction) should then be tested against data that has not been used for calibration. This is an important step in any modelling study but has not been included in the present study.*

It is of course not possible within a single study to fully address all sources of uncertainty that the reviewer leads with here, nor did we claim to do so. Our approach was to look at an existing snow-on-sea-ice estimation workflow and identify key parts of it that could be improved through automated calibration, so we limited our analysis to that part of the reanalysis we thought would play the strongest controlling role – the snowfall input. CloudSat was chosen for snowfall calibration because it represents a multi-year dataset with comparatively widespread observational coverage. MOSAiC data, although of high quality, is highly localized and only available for a single year. The key issue with any one campaign like MOSAiC is that of representation of its measurements in our effort to produce a multi-decadal pan-Arctic dataset. Drifting station snowfall measurements are likewise relatively localized compared to CloudSat measurements, and we would prefer to use contemporary observations for this work.

We agree that out-of-sample testing can be improved and we are using the most recent seasons to generate tests of the current parameter settings that we will briefly summarize. The prompt to use MOSAiC data is a good idea; rather than use it for calibration, we will use it as an independent dataset for comparison and we will provide a summary analysis that will not add too much to the length of the paper. Interestingly, the comparison of NESOSIM and SnowModel-LG with MOSAiC seems to reveal as many characteristics of the MOSAiC sampling strategy as of the models themselves. This analysis provides a significant impetus for future work in the area. This will also be touched on in the discussion. A brief summary of the analysis is included below.

**Comparison of NESOSIM and SnowModel-LG to MOSAiC observations**

Below is a brief summary with key points for a comparison to observations we intend to include in our revised manuscript; more detail and discussion will be added when we prepare the revised manuscript. We compare output from NESOSIM and SnowModel-LG to snow depth and density measurements (Macfarlane et al., 2023) obtained during the 2019-2020 Multidisciplinary drifting Observatory for the Study of Arctic Climate (MOSAiC) campaign (Nicolaus et al., 2022). Snow depths were measured using magnaprobes (Itkin et al., 2021), and in previous studies have been noted to be relatively thin (Itkin et al., 2023). Bulk snow densities used in this comparison were calculated from density cutter measurements, which sample densities at varying depths within a snow pit (Macfarlane et al., 2022). Snow was sampled over a variety of conditions, including ridges and leads, and snow over first-year and multi-year ice (Macfarlane et al., 2023).

To compare with gridded snow model outputs, MOSAiC observations are collocated to the nearest model grid point, and then averaged by day for each grid point. These values are then compared to the corresponding model grid point value, excluding dates during which NESOSIM output is unavailable. Below, we present figures aggregated by month.

[Figure]

[Figure]

Figure A: (left): MOSAiC magnaprobe snow depth (Itkin et al., 2021) compared to NESOSIM snow depth, before ("E5config") and after ("MCMC") individual dataset calibration to observations; triangles indicate individually-calibrated datasets. (right): MOSAiC snow depth compared to SnowModel-LG snow depth with different snow forcings. May-August are excluded due to the absence of NESOSIM data. Error bars represent 1 standard deviation of the monthly mean (with MOSAiC data also including contributions from the daily standard deviation).

Figure A shows monthly-averaged MOSAiC snow depth measurements (Itkin et al., 2021) compared to NESOSIM and SnowModel-LG. Note that aggregated MOSAiC outputs may differ slightly between figure panels because the model output is being aggregated to different model grids. Both models show general good agreement with the observations, with some products showing slight biases. The uncalibrated NESOSIM output driven by JRA55 has a general high bias relative to the other products (and a daily mean bias of 3.2 cm relative to MOSAiC). Differences in seasonal cycles are apparent between the models. Compared to MOSAiC, several NESOSIM products are biased low in October-November 2019, and some products are biased high in March-April. SnowModel-LG (particularly when driven by MERRA-2) is conversely biased slightly high in November and December. Nevertheless, overall agreement is close, with daily root-mean-square difference not exceeding 10 cm for all products relative to MOSAiC.

[Figure]

Figure B: Monthly averages of gridded MOSAiC snow density cutter data vs. monthly averages of coincident model output data. Error bars indicate 1 standard deviation. (left): MOSAiC compared to NESOSIM, before ("E5config") and after (MCMC) the MCMC calibration. (right): MOSAiC compared to SnowModel-LG. Only months with at least 8 measurements are shown.

 Figure B shows monthly averages of MOSAiC snow density cutter measurements (Macfarlane et al., 2022) compared to NESOSIM and SnowModel-LG. Prior to gridded collocation with the models, bulk density for each measurement event was calculated from the average of sampled densities weighted by sample snow thickness. NESOSIM snow density from all products shows relatively little variation over the time period, whereas SnowModel-LG snow density shows more seasonality. Both models show a high mean bias relative to observed values, with SnowModel-LG driven by MERRA-2 having the largest daily mean bias (60 kg/m$^3$). The comparatively high variability of the observed values is also apparent.

Below are tables with daily (gridded) comparison statistics for NESOSIM and SnowModel-LG with respect to MOSAiC, including correlation, root-mean-square difference (RMSD) and mean bias error (MBE), for reference.

Table A: Daily comparison statistics for NESOSIM and SnowModel-LG comparisons to MOSAiC snow depth observations

| | NESOSIM-ERA5-MCMC | NESOSIM-JRA55-E5config | NESOSIM-JRA55-MCMC | NESOSIM-MERRA2-E5config | NESOSIM-MERRA2-MCMC | SnowModel-LG-ERA5 | SnowModel-LG-MERRA2 |
|---|---|---|---|---|---|---|---|
| Pearson Correlation to MOSAiC | 0.68 | 0.67 | 0.67 | 0.67 | 0.67 | 0.64 | 0.58 |
| RMSD (cm) | 8.5 | 9.5 | 8.6 | 8.7 | 8.9 | 9.3 | 10 |
| MBE (cm) | -2.4 | 3.2 | -1.6 | -0.24 | -2.9 | -0.26 | 1.8 |

Table B: Daily comparison statistics for NESOSIM and SnowModel-LG comparisons to MOSAiC snow density observations

| | NESOSIM-ERA5-MCMC | NESOSIM-JRA55-E5config | NESOSIM-JRA55-MCMC | NESOSIM-MERRA2-E5config | NESOSIM-MERRA2-MCMC | SnowModel-LG-ERA5 | SnowModel-LG-MERRA2 |
|---|---|---|---|---|---|---|---|
| Pearson Correlation to MOSAiC | 0.22 | 0.20 | 0.20 | 0.15 | 0.15 | 0.24 | 0.16 |
| RMSD ($kgm^{-3}$) | 79 | 80 | 80 | 80 | 79 | 80 | 93 |
| MBE ($kgm^{-3}$) | 32 | 32 | 35 | 32 | 28 | 32 | 60 |

References (abbreviated for this response, full references will be included in the article):

Itkin, P., Hendricks, S., Webster, M., et al.: Sea ice and snow characteristics from year-long transects at the MOSAiC Central Observatory, *Elementa: Science of the Anthropocene*, 11, 00048, https://doi.org/10.1525/elementa.2022.00048, 2023.

Macfarlane, A. R., Schneebeli, M., Dadic, R. et al.: A Database of Snow on Sea Ice in the Central Arctic Collected during the MOSAiC expedition, *Sci Data*, 10, 398, https://doi.org/10.1038/s41597-023-02273-1, 2023.

Nicolaus, M., Perovich, D. K., Spreen, G. et al.: Overview of the MOSAiC expedition: Snow and sea ice, *Elementa: Science of the Anthropocene*, 10, 000046, https://doi.org/10.1525/elementa.2021.000046, 2022.

Dataset references:

Itkin, Polona; Webster, Melinda; Hendricks, Stefan, et al. (2021): Magnaprobe snow and melt pond depth measurements from the 2019-2020 MOSAiC expedition [dataset]. *PANGAEA,* https://doi.org/10.1594/PANGAEA.937781

Macfarlane, Amy R; Schneebeli, Martin; Dadic, Ruzica, et al. (2022): Snowpit snow density cutter profiles measured during the MOSAiC expedition [dataset]. *PANGAEA,* https://doi.org/10.1594/PANGAEA.940214, In: Macfarlane, AR et al. (2021): Snowpit raw data collected during the MOSAiC expedition [dataset bundled publication]. *PANGAEA,* https://doi.org/10.1594/PANGAEA.935934

*Another issue with the paper is the section on trend analysis. I don't think this section adds anything to the paper. Moreover, it is not always clear from the discussion when trends have passed tests for statistical significance. An example of this is:*

> *Although the trend magnitudes and seasonal cycles for snowfall vary by region, most of the trends for most products are not statistically significant at a 95% confidence interval, likely due to high interannual variability of snowfall.*

*In trend analysis, the most common purpose of a significance test is to decide whether or not the null hyposthesis, that the trend is zero (\alpha = 0), can be rejected.  If it cannot be rejected (i.e. it is "insignificant"), the null hypothesis has to be accepted; no trend distinguishable  from zero can be detected.  Put another way, there is no trend, it is zero and it cannot be positive or negative value.  So in the example  above, trends for the regionds cannot be different if they are not statistically significant.*

*If the authors can demonstrate that this section is relevant to the paper, I would suggest only discussing statistically significant trends in the  text and only showing statistically significant trends in the figures.  Note, the grey hatching is not visible in the pdf version of the paper.  I  suggest not showing regions that do not pass the significance test.  Also the overlapping shaded regions in the line plots obscure the  information.  As the plots show trends for each month, the statistically significant trends for each product should be shown as symbols with  bars to show the 95% confidence intervals, grouped by month, because they are individual data points.*

We respectfully disagree that the section does not add anything to the paper. The motivation for this section was to examine how trends interpreted from snow-on-sea-ice products can differ depending on which product and which reanalysis input is used, and to examine if trends differ more between products or between snow forcings. There is precedent for using reanalysis products to examine trends in Arctic precipitation (e.g. Boisvert et al, 2018) and given the lack of widespread long-term continuous observations of snow on Arctic sea ice, we find it justifiable to consider the use of reanalysis-based snow-on-sea-ice reconstructions such as NESOSIM (and SnowModel-LG, for that matter) to examine trends and variability. In fact, SnowModel-LG has recently been used to examine the influence of the Arctic Oscillation on summer snow on sea ice (Webster et al. 2024). Furthermore, previous sea ice studies have found that snow can influence sea ice volume (and therefore thickness) trends (Bunzel et al., 2018), and as such, for a product with sea ice thickness applications, we believe that investigating these trends is relevant.

References:

Boisvert, L. N., Webster, M. A., Petty, A. A., Markus, T., Bromwich, D. H., and Cullather, R. I.: Intercomparison of Precipitation Estimates over the Arctic Ocean and Its Peripheral Seas from Reanalyses, *J. Climate*, 31, 8441–8462, https://doi.org/10.1175/JCLI-D-18-0125.1, 2018.

Bunzel, F., Notz, D., and Pedersen, L. T.: Retrievals of Arctic Sea-Ice Volume and Its Trend Significantly Affected by Interannual Snow Variability, *Geophysical Research Letters*, 45, 11,751-11,759, https://doi.org/10.1029/2018GL078867, 2018.

Webster, M. A., Riihelä, A., Kacimi, S., Ballinger, T. J., Blanchard-Wrigglesworth, E., Parker, C. L., and Boisvert, L.: Summer snow on Arctic sea ice modulated by the Arctic Oscillation, *Nat. Geosci.*, 17, 995–1002, https://doi.org/10.1038/s41561-024-01525-y, 2024.

As with any reanalysis-based product, caution is required when interpreting trends. Our work demonstrates that different conclusions may be drawn about basin-wide and regional trends depending on what product is used. As discussed above, differing trends in snow on Arctic sea ice products have implications for ice thickness retrieval applications. We also find it interesting to explore how snow depth trends do not necessarily reflect reanalysis snowfall trends (or lack thereof).

Regarding the phrasing around statistical significance, we agree that some of our phrasing was unclear, and we did not intend to overstate results that did not represent actual trends (i.e. were not statistically significant). We propose some substantial rephrasing of our section on trends further below to address this issue (refer to our

comments further below addressing Line 414 and onwards), and we will likewise adjust phrasing in the discussion accordingly.

Regarding trend plots, as suggested, we will remove the values that are not statistically significant. As the reviewer suggested, we propose the following modification to Fig. 10; substituting the shading for bars. Note that this is an illustrative example; we will adjust the subplot horizontal spacing and include a legend for better legibility in the next revision of the manuscript. By suggestion of another reviewer, we will also include tables of values corresponding to these plots in a supplement. Also, we have removed the row with snowfall trends and will show that in a supplement, since snowfall trends are generally absent.

[Figure]

"Figure 10. Monthly trends for regionally-averaged quantities over the 1980-2019 time period: snow depth, density and volume (from NESOSIM and SnowModel-LG), and sea ice area (from the Climate Data Record product). Error bars indicate a 95% confidence interval as given by the trend estimator; points where there are no trends (the interval overlaps the zero line) are not shown."

For reference, the regional snowfall trends over sea ice are shown below:

[Figure]

Likewise, we will fully grey-out regions with no significant trends in Fig 11 and 12, and will move Fig. 11 to a supporting figure. We show the updated Fig. 12 for reference; Fig. 11 will be updated similarly:

March snow depth trend (cm/y)

[Figure]

"Figure 12: Snow depth trend maps for March 1980-2019 from NESOSIM run with snowfall input from ERA5, MERRA-2, and JRA-55, and SnowModel-LG with snowfall input from ERA5 and MERRA-2. The snow depth is output from NESOSIM with parameters specific to each separate reanalysis product. The trend in the average of the output of the three NESOSIM runs is also plotted (Average). Regions with no trends (not significant to a 95% confidence interval) are shaded in grey. Note that SnowModel-LG is not provided within the Canadian Arctic Archipelago, so data from that region is absent in this map."

We propose modifying the text of the section on trends as follows (paragraphs labelled by line):

Line 414: "In the following discussion, trends are considered significant if the 95% confidence interval does not overlap with zero. If the confidence interval overlaps with zero, we consider there to be no trend."

Line 416: "Basin-average trends from NESOSIM for snowfall over sea ice, snow depth, snow density, snow volume, and sea ice area are shown in Fig. 9. The trends in snowfall over sea ice are not statistically significant for most products except for a  decline for MERRA-2 from November onwards and an increasing snowfall trend in October for JRA-55.  The basin-average trends in snow depth from MCMC-calibrated NESOSIM output vary in magnitude by product, but are all broadly similar in sign. MERRA-2 has the strongest trends in the basin-average overall. The trend is found to be negative (declining snow depth) in all months except September, where the trend is significantly positive for all products, and October, where the trend is not significant for the multi-product average; MERRA-2 shows a  decline but the other products show no  trend in that month. Snow density trends are generally  similar between the products, aside from October, where only JRA-55 shows a trend.  Similarity between the snow density trends is expected, since snow density in NESOSIM is less sensitive to snow input, being primarily dependent on wind speed."

We modify Figure 9 as follows:

[Figure]

"Figure 9. Basin-average monthly trends from 1980-2019 for snowfall over sea ice from reanalysis products, MCMC-calibrated NESOSIM snow depth, density, and volume, and CDR sea ice concentration, calculated using a Theil-Sen trend estimator for all products. "Average" denotes the multi-product average. Error bars indicate a 95% confidence interval as given by the trend estimator; points where there are no trends (the interval overlaps the zero line) are not shown. The grey dashed lines indicate the zero line for reference. SnowModel-LG is excluded from this plot due to differences in model domains."

Line 431: "Regional trends in  snow depth, snow density, snow volume, and sea ice area are shown in Fig. 10. Regional trends in snowfall are shown in Fig. A#; although there is regional variation in snowfall trends,  most products show no trend for most months, likely due to high interannual variability of snowfall.

and April, where a declining trend is observed.  A large and significant early-season snowfall decline is apparent in the Kara Sea region, but only for the month of September for most products."

Line 441: "The East Greenland Sea region differs noticeably  from the other regions shown with no trend for most months except for a slight increase in November for SnowModel-LG driven by MERRA-2, and declines in April only for NESOSIM products.  In the Central Arctic region,  declines are generally seen only for products driven by MERRA-2, with no trend for products driven by JRAA-55, and only a slight October decline for SnowModel-LG driven by ERA5."

Line 450: change sentence to "In most regions, the strongest declining trends are found in MERRA-2, whereas trends often tend to be smaller or absent for ERA5, for both NESOSIM and SnowModel-LG."

Line 452: "Several regions demonstrate  small or absent snow depth trends in September followed by an abrupt decline in the following months. These patterns are not reflected in the snowfall trends, and are likely to be related to sea ice decline. In the Central Arctic region, where more sea ice is present during the early months, the early-season change in the trend is less pronounced . However, in the Kara Sea region, which experienced a significant declining trend in snowfall over sea ice in September for all snowfall products , a corresponding decline in snow depth is not observed in September in either NESOSIM or SnowModel-LG. Nevertheless, significant declines are found in this region for  later months."

Line 458: "For snow density trends, inter-model differences tend to be larger than inter-product differences. Declining trends are largest  around October-November for most products and regions, except in the Barents and East Greenland Seas. Densities in SnowModel-LG tend to show large  declines relative to NESOSIM. As discussed previously, NESOSIM end-of-season density trends may be spuriously low due to NESOSIM snow densities approaching their maximum towards the end of the season, although end-of-season density trends as represented in SnowModel-LG also tend to be smaller. ."

Line 463: "Trends in snow volume closely mirror snow depth trends in several regions, though differences are nevertheless apparent. In the Central Arctic, Beaufort Sea and Chukchi Sea regions, there is a notable significant early-season decline in snow volume for all products, whereas such a decline is not necessarily found in the corresponding snow depths for these regions. . These volume declines are associated with strong early-season sea ice area declines. The inter-product spread in trends increases towards the end of the season, however. In the Kara Sea region,  snow volume trends are declining from October onward. In the Barents and East Greenland Sea regions,  later-season declines are also apparent for most products. Snow volume trends in these two regions differ considerably in seasonality from the snow depth trends, with stronger late-season declines, likely influenced by sea ice area decline in these regions. There is a large inter-model difference in trend magnitude between NESOSIM and SnowModel-LG in the East Greenland Sea region, with NESOSIM showing much larger declines overall, and SnowModel-LG driven by ERA5 not showing any declining trend until March. The largest snow volume declines are found in the Central Arctic for NESOSIM driven by MERRA-2, although this region also has a very wide inter-product spread, with NESOSIM driven by ERA5 and JRA-55 and SnowModel-LG driven by ERA5 not showing  late-season snow volume declines. Thus, the choice of reanalysis also has an impact on snow volume trends, though inter-model differences are more readily apparent in some regions."

Line 477: "Sea ice area trends vary by region, but strong declines are found for at least part of the season in all regions shown. In the Central Arctic and the Siberian sector, as well as the Beaufort Sea, the  largest declining trends are in the earlier months of the cold season. ( Larger trends may be present in months outside of the NESOSIM study period.) When sea ice in these regions attains its maximum extent, the trends

largely vanish, suggesting a persistent cold-season cover. Towards the North Atlantic (Barents, East Greenland),  larger declines are seen in later months."

Line 482 (A# is a placeholder for the number of the figure in the final manuscript): "To provide a more regional perspective on snow trends,  maps of snow depth trends in NESOSIM and SnowModel-LG output are shown in Fig. 12. Corresponding snowfall trends are shown in Supporting Fig. A#. For these plots, trends were also calculated using a Theil-Sen estimator, but only grid squares containing at least 20 years of values were included to exclude spurious trends. Consistently with results from the regional monthly trend plots,  there is a lack of snowfall trends over most of the Arctic basin, due to the high interannual variability of Arctic snowfall relative to the magnitude of the trends. The depth trends are more robust, highlighting a decline in the peripheral seas consistent with the results shown in the regional plots, as well as some slight declines around Hudson Bay and Labrador Sea. Some significant increasing depth trends north of the Beaufort Sea are found in both SnowModel-LG products, as well as in NESOSIM driven by ERA5 and JRA-55, though the products differ on the existence  of the increasing trend near the North Pole."

Line 494: "There is broad consistency, however, in the  declining trend found  in the Barents Sea region. The overall  large declining trends in depth derived from MERRA-2 are particularly apparent in Fig. 12. ERA5 and JRA-55 agree better on the spatial pattern of the snow depth trends compared to MERRA-2."

Line 499: "There are small but significant increases in snow depth in Hudson Bay and some strong declines east of Greenland that are absent from the NESOSIM output.

*The reader is left to do a lot of work to understand what was done and how the NESOSIM model works. The description of the scaling using CloudSat is very brief (Lines 211 to 215). Cabaj et al (2020) describes the scaling but I cannot find how it is applied to the four Arctic quadrants in Cabaj et al (2023). In my opinion, papers should contain sufficient information to allow a reader to understand what was done. I would recommend including a brief description of how CloudSat scaling is applied and how blowing snow and wind packing parameters are applied.*

We thank the reviewer for indicating where clarification is needed, and will make changes accordingly. It appears that we have erroneously left in some text which was supposed to be relegated to a different section; we are moving the description of the quadrants to Section 3.2 and will expand the description to clarify. We will also modify Fig. A1 to better illustrate the quadrants (including the associated interpolation points). We propose the following changes to the text below:

From line 205:

Figure 1 shows regionally-aggregated monthly-mean snowfall rates from reanalysis products and CloudSat, from 1980-2016, without and with scaling to the CloudSat monthly climatology (Cabaj et al., 2020).  The scaling entails taking the monthly reanalysis snowfall rate for each month and multiplying it by a scaling factor, which consists of the CloudSat climatological monthly mean snowfall rate divided by the reanalysis climatological monthly mean snowfall rate for each respective month. The climatological means for this scaling are taken from 2006-2016, excluding months in 2011 where CloudSat observations are absent due to instrument malfunctions. Further details of this scaling are provided in Cabaj et al. (2020). Before the scaling is applied in Fig. 1, [...]"

From line 214, text moved to line 205 above:

"As in Cabaj et al. (2020), we bias-adjust reanalysis snowfall input for NESOSIM to climatological CloudSat snowfall for 2006-2016 (excluding months in 2011 where CloudSat observations are absent due to instrument malfunctions). The adjustment uses multiplicative scaling interpolated across four Arctic quadrants, a level of aggregation that was found to be a necessary to obtain robust results (Cabaj et al., 2023). CloudSat scaling [...]"

From Line 222 revising as follows:

"To apply CloudSat scaling over the NESOSIM model domain, reanalysis snowfall rates are scaled to CloudSat measurements from 60-82°N over four quadrants, as described in Cabaj et al. (2020). The scaling coefficients for these quadrants were obtained by dividing the NESOSIM model domain north of 60N into four quadrants (illustrated in Fig. A1), calculating monthly reanalysis and CloudSat snowfall climatologies for each respective quadrant to obtain climatological scaling factors for each quadrant following the approach in Cabaj et al. (2020). Then, the scaling factors were linearly interpolated over the model domain from the outer corners of the model domain towards the centre, with one scaling factor at each corner. The CloudSat scaling was found to improve agreement in basin-averaged and regionally-averaged snow depths in NESOSIM v1.0, as was discussed in Cabaj et al. (2020). Some adjustments were made to the scaling for NESOSIM v1.1, which has a larger model domain (Petty et al., 2023), extending down to 50°N, compared to 60°N for NESOSIM v1.0 (Petty et al., 2018). The model domain of NESOSIM v1.1 is shown in the map in Fig. A1, with shading indicating the 60-82°N latitude band, and reference points for the scaling coefficient interpolation indicated. To apply CloudSat scaling over the NESOSIM model domain, reanalysis snowfall rates are scaled to CloudSat measurements from 60-82°N over four quadrants, as discussed in Cabaj et al. (2020), and are linearly interpolated over the model domain from the corners. The longitudinal boundaries of the quadrants are at longitudes 135°W, 45 °W, 45°E, and 135 °E, respectively, as illustrated in Fig. A1. For this work, the scaling is set up as follows. The same scaling coefficients as previously calculated for Cabaj et al. (2020) are used. Over a central rectangular subdomain with corners at 45°N (longitudes of 90°W, 0°E, 90°E, and 180°E), the scaling factors are linearly interpolated from the corners across the pole. For the rest of the domain, the factors In NESOSIM v1.1, the scaling factors are unchanged, but are extrapolated southward as constant values to cover the extended model domain."

Regarding describing how wind packing and blowing snow factors are applied, we are adding the following to clarify based on this comment and suggestions from other reviewers, at Line 134:

"using an MCMC process (Cabaj et al., 2023). At each model step and grid point, if the input wind speed exceeds 5 m/s, the wind packing and blowing snow processes act on the snow in NESOSIM. Wind packing transfers snow from the upper (less dense) model layer to the lower (denser) model layer, controls the amount of snow transferred between layers, impacting decreasing the snow depth and increasing the bulk snow density. The wind packing factor scales the amount of snow transferred between layers at each timestep. The blowing snow process acts only on the upper snow layer, and decreases the snow depth in the upper layer linearly with wind speed. The blowing snow term includes an atmosphere loss and an open-water loss term, which are prescribed separately in NESOSIM v1.1 (Petty et al., 2023). The open-water loss term accounts for sea ice concentration, with regions of lower sea ice concentration experiencing more open water loss. The blowing snow term is exclusively a loss term and does not include redistribution. When snow is lost from a grid cell via this process, it is removed, not redistributed to another grid cell. For the purpose of this study, the blowing snow term parameters are treated as a single term, as was done in previous work (Cabaj et al., 2023), with the atmospheric loss factor being 0.15 times the blowing snow parameter. Both the wind packing and blowing snow processes are subject to a wind action threshold of 5 m/s. This current study will extend previous parameter calibration work by investigating the impact of using different reanalysis snowfall input products in NESOSIM."

*The authors should use the correct citations for datasets and I would encourage them to cite DOIs. For example. The correct citation and title for the NOAA/NSIDC sea ice concentration is:*

*Meier, W. N., F. Fetterer, A. K. Windnagel, and S. Stewart. 2021. NOAA/NSIDC Climate Data Record of Passive Microwave Sea Ice Concentration, Version 4. [Indicate subset used]. Boulder, Colorado USA. NSIDC: National Snow and Ice Data Center https://doi.org/10.7265/efmz-2t65. [Date Accessed].*

*This allows the data to be found easily and quickly instead of digging through the paper describing the dataset. They should should check that the data citations are correct for the other datasets they use.*

We appreciate the correction and will double-check and revise our dataset citations, and include DOIs where possible. We note that some of the datasets (e.g. the CRREL buoy data) do not have DOIs (we have cited those data in the format requested by the dataset website, though we have noticed a duplicate reference there which we will clean up).

*It is not clear to me why the CRREL Buoy snow depths are treated as monthly climatologies but the OIB snow depths are daily. Why not use daily data from the buoys.*

This is somewhat of a subtle point, and relates to the spatial extent of the observations. In our previous model calibration work (Cabaj et al., 2023), we found that using daily buoy data led to the calibration being overconstrained; each buoy is localized at a single point (rather than having a comparatively large spatial extent, as OIB did). Since capturing small-scale local variations is challenging for a model with a 100-km grid, we opted to aggregate the buoy measurements for the purpose of the calibration. The OIB measurements are also aggregated, but to a daily 100 km grid. We did also test using climatological values for the OIB measurements in our 2023 publication, but we ultimately found that the daily measurements provided a good balance between data coverage and constraint.

*Specific Comments*

*Line 227: Suggest "described" instead of discussed.*

Thank you for the suggestion; we have moved this line up as mentioned in the revisions above, (starting from line 222) but have rephrased this sentence accordingly.

*Line 228: I have no idea what "interpolated from the corners" means here. Corners of the quadrants or of what?*

We intended this to mean from the corners of the quadrants towards the centre of the pole, although offset at a higher latitude. We have rephrased as described earlier in our revision (from line 222) and will update Figure A1 with the relevant regions outlined and the corner points identified to clarify.

*Line 242: Which regions and months?*

Over sea ice, the overall inter-product spread increases for Sept-Nov in the Central Arctic, October in the Beaufort Sea, Oct-November in the Chukchi Sea, all months except September in the Kara and Barents Seas, and all months except April in the East Greenland Sea. We will specify this in the revised manuscript.

*Line 243: I am trying to understand why the inter-reanalysis spread is not reduced by scaling with CloudSat. The discussion indicates that over the central Arctic, the inflation of JRA-55 may be because CloudSat does not extend above 82N. This suggests to me that an alternative scaling strategy for this region should be explored.*

Overall, the CloudSat scaling performs best in aggregate over large regions, since smaller regions may have high variability which may not necessarily be accounted for by the scaling. When the analysis is relegated to smaller regions and restricted to include only sea-ice-covered regions (which also decreases the size of regions analyzed), we find that the scaling performs worse. We considered alternate scaling approaches, but there is a tradeoff with using smaller reference regions for calculating scaling factors, since then, spatial undersampling from CloudSat may be encountered. In particular, one approach we attempted was calculating CloudSat scaling factors using only reanalysis and CloudSat snowfall over ice-covered grid cells as determined by a sea ice area climatology. This did improve the agreement even in smaller regions, but as we discussed in the article, we encountered

sampling issues as shown in Figure A3: Due to limited coincident CloudSat measurements coincident with the (limited-extent) sea ice in the Greenland/Norwegian Sea quadrant, CloudSat failed to reproduce the monthly climatology in that region, and this biased the precipitation excessively low. We do agree that exploration of additional scaling strategies would be of interest, but we leave this as a consideration for future work.

*Line 250: How well does the CloudSat snowfall rate algorithm perform over sea ice?*

The primary factor influencing CloudSat retrieval performance over sea ice is enhanced ground clutter due to the higher elevation of sea ice (relative to open ocean). The CloudSat retrieval algorithm estimates surface snowfall from vertical radar profiles by selecting a designated near-surface bin. To mitigate adverse impacts from sea ice, for locations over a climatological sea ice mask, the retrieval algorithm selects snowfall from the $5^{th}$ radar bin above the (retrieval-designated) surface rather than the $3^{rd}$ radar bin from the surface (as is usually the case over ocean). We would expect some biases to be introduced here if sea ice were present in regions that did not coincide with the climatological mask. In an effort to mitigate this, we filter data based on retrieval quality flags as was done in Cabaj et al. (2020) but we are aware that some biases may still be present. We will add mention of this to Section 2.2.

*Line 267: It is not clear if the first iterations refer to the burn-in period or to the first iterations after the burn in period. Please clarify in the text.*

We will rephrase as follows: "the optimal posterior parameters values did not differ significantly between the first (burn-in) and subsequent (after burn-in) sets of iterations,"

*Figure 3: It is difficult to see the individual marginal distributions. I would suggest using lines rather than bars.*

We agree, and will change the marginal distributions to be shown with lines for the next revision.

*Line 277: I suggest avoiding the "a(b) for c(d)" pattern and write this out in full. It is much easier to read. E.g. "Coefficients of variation for the wind packing parameters are 15% for ERA5, 42% for JRA-55... Coefficients of variation for blowing snow parameters are 13% for ERA5, 38% for JRA-55..." Or better yet, use a table.*

We note that we do already use a table—these values are shown in Table 2—but we will rephrase the lines in the text for clarity as suggested: "The respective coefficients of variation for the wind packing parameters, as indicated in Table 2, are 15% for ERA5, 42% for JRA-55, and 21% for MERRA-2. The coefficients of variation for the blowing snow parameters are 13% for ERA5, 38% for JRA-55, and 19% for MERRA-2."

*Line 325: Aren't spread and uncertainty the same thing?*

We acknowledge that the phrasing was ambiguous, and will clarify by replacing mentions of "spread" here with "inter-product differences". The point being discussed here was the impact of considering uncertainties from multiple products in aggregate.

*Line 411: Suggest "following" instead of "consistent with".*

Agreed, we will make the change at Line 411 as suggested.

---

## Author Response (AR2)

Authors' response to minor revisions for "Investigating the impact of reanalysis snow input on an observationally calibrated snow-on-sea-ice reconstruction", Cabaj et al., egusphere-2024-2562

We thank the reviewers for their helpful feedback and have made the minor revisions as suggested. Reviewer comments pertaining to these revisions are presented italicised in blue and our responses are in plain text. Other changes we have made include other minor phrasing adjustments for clarity/correctness, and fixing incorrect legend markers in Fig 11.

*L261: (retrieved ...; Liston et al. (2021)) >> (retrieved ...; Liston et al., 2021)* Changed as suggested

*L482: (with ... Meier and Stewart (2023)) >>(with ... Meier and Stewart, 2023)* Changed as suggested

*L511: ("MCMC"))> ("MCMC")* There are two parentheses here because the parentheses are nested (refer to the open parenthesis near the beginning of the sentence) so we leave this unchanged. We have, however, checked through the document for other instances of double parentheses just in case.

*L807: sub-gridscale >> sub-grid scale* Changed as suggested

*L808: may not be coincident with>> may not align with?* Changed to "may not align with" as suggested

*L816: on differing sides of the threshold>> opposite sides?* Changed to "opposite sides" as suggested

*L826: This highlights the value of accounting for uncertainties due to >>*
*This highlights the importance of accounting for uncertainties arising from* Changed as suggested

*L827: spans a more reasonable 8-18% range >> ranges from 8% to 18%, which is more reasonable*
Changed as suggested

*L830: due to the limited density range represented by the model >> because of the model's limited density range* Changed as suggested

*L832: This calls into question >> This challenges* Changed as suggested

*L834: An analogous >> A similar* Changed as suggested

*L853: The term "climate sensitivity" has a specific meaning in climate science. Revise to the sensitivity of sea ice variables to changes in snow depth?* Thank you for catching this, we are familiar with the specific technical meaning of this phrasing and used it unintentionally. Revised to "[...] impacts on estimates of the sensitivity of sea ice variables to changes in snow depth."

*L890: I find the connection between the two sentences unclear. The first sentence states that snow depth trends may align more closely between models even in regions where climatologies (e.g., average snow depth) disagree, suggesting that systematic biases persist but trends are less sensitive to model differences. However, the next sentence emphasizes that the choice of reanalysis snow input greatly impacts the magnitude and statistical significance of snow depth trends, implying that trends are indeed sensitive to reanalysis inputs. These statements appear contradictory. Could the authors clarify this? Or remove.* We meant to state that trends can agree in regions where climatologies disagree (i.e. disagreement between climatologies does not necessarily imply disagreement in trends), but nevertheless, trends do not agree everywhere (and hence must be interpreted with caution). As suggested, we have rephrased to clarify as follows: "In regions where climatologies disagree between models, snow depth trends between models can sometimes show more agreement. In other cases, the choice of reanalysis snow input can still greatly impact the magnitude and statistical significance of snow depth trends, and thus, trends derived from reanalysis-based reconstructions of snow on sea ice must be treated with caution."

*L956: (Tschudi et al., 2019)) >>(Tschudi et al., 2019)* This was a closing parenthesis corresponding to the open parenthesis at L954; have removed the parentheses and rephrased slightly for added clarity